# Holliday junction–ZMM protein feedback enables meiotic crossover assurance

Adrian Henggeler[1,2,3], Lucija Orlić[1,2,4,5], Daniel Velikov[1,2,4,5] & Joao Matos[1,2✉]

Holliday junctions (HJs) are branched four-way DNA structures that link recombining chromosomes during double-strand break repair[1]. Despite posing a risk to chromosome segregation, HJs accumulate during meiotic prophase I as intermediates in the process of crossing-over[2,3]. Whether HJs have additional regulatory functions remains unclear. Here we establish an experimental system in budding yeast that enables conditional nucleolytic resolution of HJs after the establishment of meiotic chromosome synapsis. We find that HJ resolution triggers complete disassembly of the synaptonemal complex without disrupting the axis–loop organization of chromosomes. Mechanistically, HJs mediate the continued association of ZMM proteins with recombination nodules that form at the axes interface of homologous chromosome pairs. ZMM proteins, in turn, promote polymerization of the synaptonemal complex while simultaneously protecting HJs from processing by non-crossover pathways. Thus, reciprocal feedback between ZMMs, which stabilize HJs, and HJs, which retain ZMM proteins at future crossover sites, maintains chromosome synapsis until HJ-resolving enzymes are activated during exit from prophase I. Notably, by polymerizing and maintaining the synaptonemal complex structure, the HJ–ZMM interplay suppresses de novo double-strand break formation and recombination reinitiation. In doing so, this interplay suppresses the DNA damage response, enabling meiotic progression without unrepaired breaks and supporting crossover assurance.

During meiosis, an evolutionarily conserved proteinaceous structure—the synaptonemal complex (SC)—mediates the synapsis of homologous chromosome pairs to regulate genetic exchange and crossing-over[4]. The SC comprises filamentous lateral elements that organize chromatin into linear arrays of loops together with cohesin complexes, and a central region that connects the two homologue axes[5]. The central region of the SC, but not the axes, is thought to have liquid-crystalline properties, where weakly bonded proteins can undergo internal rearrangement and exchange[6–10]. SC assembly takes place gradually, starting in the zygotene stage and eventually connecting the paired homologue axes along their entire length during the pachytene stage, a process that takes around 2.5 h in budding yeast and several days in mice[4,11]. How SC formation is initiated and how the SC structure is subsequently maintained to ensure the stable end-to-end engagement of all homologue pairs remain subjects of intense research.

In several organisms, including budding yeast, SC assembly is functionally linked to the repair of developmentally programmed DNA double-strand breaks (DSBs) through homologous recombination[12]. SC polymerization initiates at a subset of recombination sites where interhomologue DNA joint molecules are progressively stabilized and mature into intermediates containing two four-armed HJs (double HJs, dHJs)[2,11,13–15]. A link between synapsis and recombination is provided by a group of meiosis-specific proteins, collectively known as ZMMs (Zip1–4, Msh4–5, Mer3, Spo16), which bind to and stabilize nascent repair intermediates and coordinate dHJ maturation with SC assembly[14–23]. By promoting SC assembly between homologous chromosomes, ZMMs contribute to the downregulation of DSB formation[24–29]. Whether ZMMs are continuously required for the stability of dHJs, for the maintenance of chromosome synapsis and, as such, for suppressing the formation of new DSBs is unclear.

Whereas SC assembly takes place gradually, SC disassembly is linked to a sharp cell cycle transition that drives exit from prophase I and entry into the first meiotic division[30]. Notably, SC disassembly and dHJ resolution are temporally coordinated by Ndt80-mediated expression of polo kinases, which activate HJ resolvases while also driving the disassembly of SC components[31–33]. Such a tight spatio-temporal relationship between the stabilization and resolution of recombination intermediates and the assembly and disassembly of the SC led us to examine whether recombination intermediates have a previously unknown role in the maintenance of chromosome synapsis. We identified a reciprocal functional interplay between dHJs and ZMM proteins that is crucial for maintenance of the SC structure. We propose that, by supporting chromosome synapsis and, thereby, contributing to the suppression of de novo DSB formation, this dHJ–ZMM interplay coordinates meiotic progression with crossover assurance.

[1]Max Perutz Labs, Vienna Biocenter Campus (VBC), Vienna, Austria. [2]University of Vienna, Max Perutz Labs, Department of Chromosome Biology, Vienna, Austria. [3]Institute of Biochemistry, ETH Zürich, Zurich, Switzerland. [4]Vienna BioCenter PhD Program, Doctoral School of the University of Vienna and Medical University of Vienna, Vienna, Austria. [5]These authors contributed equally: Lucija Orlić, Daniel Velikov. ✉e-mail: joao.matos@maxperutzlabs.ac.at

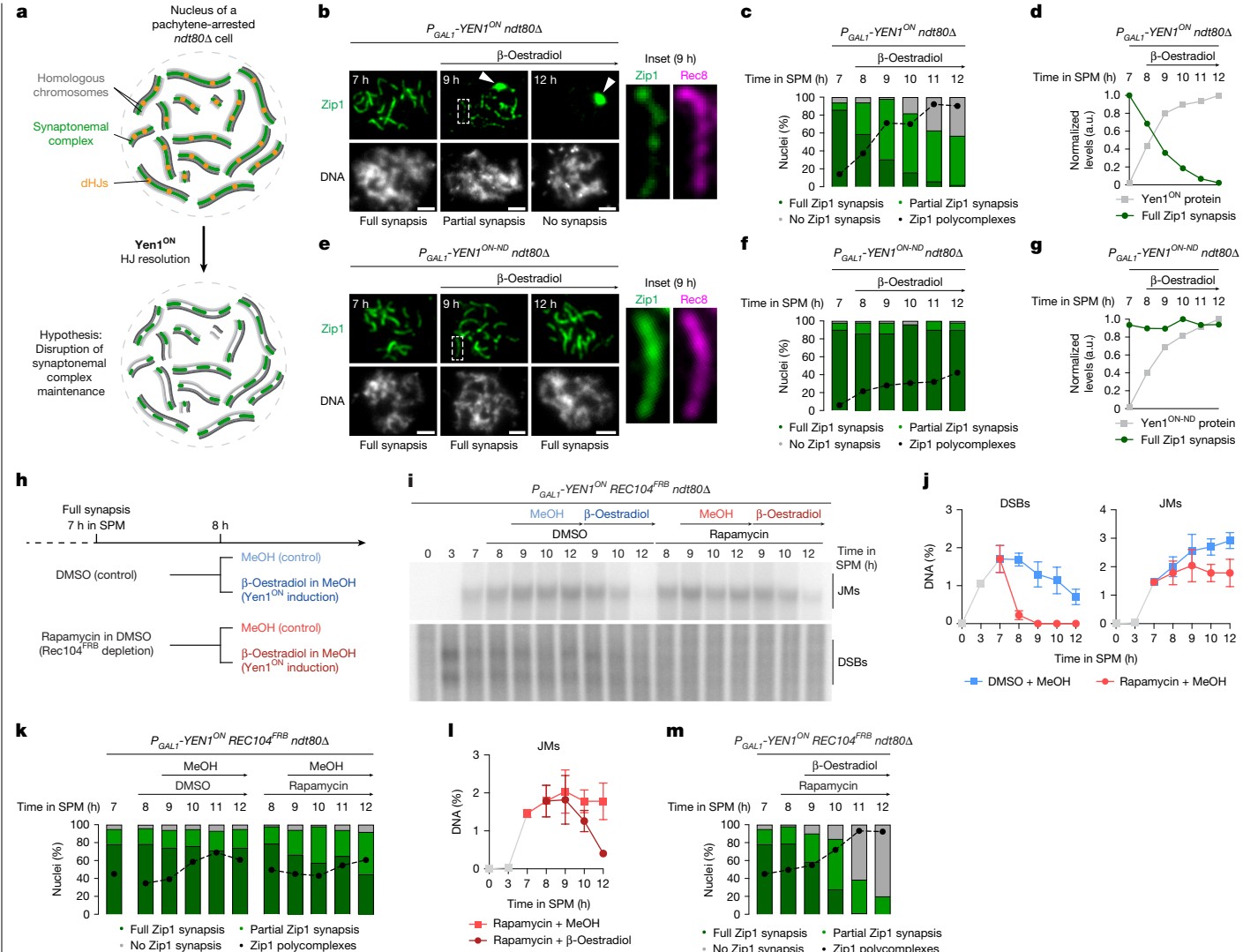

**Fig. 1 | HJs stabilize the SC during meiotic pachytene. a**, The experimental set-up for conditional nucleolytic resolution of HJs after meiotic chromosome synapsis is established. **b**, Representative images of meiotic chromosome spreads at the indicated times in SPM, immunostained for Zip1 (green); DNA was stained with 4′,6-diamidino-2-phenylindole (DAPI; grey). Yen1[ON] was induced by addition of β-oestradiol (or methanol (MeOH) as a control) at 7 h. The insets show a magnification of the region indicated by a dashed box; Rec8 (magenta) marks chromosome axes. The arrowheads indicate Zip1 polycomplexes. **c**, Quantification of Zip1 synapsis and polycomplexes from **b**. *n* = 50 nuclei per timepoint, representative of three biological replicates. **d**, Quantification of Yen1[ON] protein levels from Extended Data Fig. 1b and full Zip1 synapsis from **c**, normalized to peak values. **e**, Representative images of meiotic chromosome spreads at the indicated times in SPM as in **b**, but for $P_{GAL1}$-*YEN1*[ON-ND] cells. **f**, Quantification of Zip1 synapsis and polycomplexes (as in **c**) from the experiment in **e**. **g**, Quantification of Yen1[ON-ND] protein levels (as in **d**) from the experiment in Extended Data Fig. 1d and full Zip1 synapsis from the experiment in **f**. **h**, The

experimental set-up to inhibit DSB formation by Rec104[FRB] nuclear depletion and resolve HJs by Yen1[ON] expression in pachytene-arrested *ndt80Δ* cells. **i**, Southern blot analysis of DSBs and DNA joint molecules (JMs) at the *HIS4::LEU2* recombination hotspot from **h**. Gel source and biological replicate data are provided in Extended Data Fig. 3c. **j**, Quantification of DSBs and joint molecules from the experiment in **i** and a biological replicate, with or without Rec104[FRB] nuclear depletion. Data are the mean and range of the percentage of total DNA. **k**, Quantification of Zip1 synapsis and polycomplex formation from the experiment in **i**, with or without Rec104[FRB] nuclear depletion, including an additional timepoint (11 h in SPM). *n* = 50 nuclei per timepoint, representative of two biological replicates. **l**, Quantification of joint molecules as in **j**, after Rec104[FRB] nuclear depletion and with or without Yen1[ON] induction. **m**, Quantification of Zip1 synapsis and polycomplex formation as in **k**, after Rec104[FRB] nuclear depletion and Yen1[ON] induction. *n* = 50 nuclei per timepoint, representative of two biological replicates. For **b** and **e**, scale bars, 2 µm.

## dHJs stabilize the SC during pachytene

To assess whether recombination intermediates are required for the maintenance of chromosome synapsis, we used *ndt80*-deletion (*ndt80Δ*) mutants in combination with conditional expression of an engineered structure-selective endonuclease, Yen1[ON], which can process a broad range of branched DNA repair intermediates[34] (Fig. 1a). *ndt80Δ* mutants arrest meiotic progression with synapsed homologues and unresolved dHJs[3,30]. Conversely, ectopic expression of Yen1[ON] is sufficient to drive nucleolytic resolution of HJ-containing recombination

intermediates during prophase I[35]. Yen1[ON] expression was initiated by the addition of β-oestradiol to cultures 7 h after induction of meiosis, at which point more than 85% of cells displayed fully synapsed chromosomes as assessed by immunostaining the transverse filament protein Zip1 (ref. 36) on chromosome spreads (Fig. 1b,c (7 h in sporulation medium, SPM) and Extended Data Fig. 1a). Notably, expression of Yen1[ON] resulted in a progressive loss of chromosome-associated Zip1 (Fig. 1b,c). This was accompanied by a sharp increase in nuclei with Zip1 aggregates, known as polycomplexes, which are known to form as a consequence of impaired SC assembly[36] (Fig. 1b (white

arrowheads) and 1c (dashed line)). Despite the altered localization, Zip1 protein levels were unaffected by Yen1[ON] expression, suggesting that the loss of Zip1 from chromosomes occurred independently of changes in protein expression or stability (Extended Data Fig. 1b,c). Supporting a causal link between the nucleolytic resolution of recombination intermediates and chromosome synapsis defects, the increase in Yen1[ON] protein levels mirrored the decreasing proportion of nuclei with fully synapsed chromosomes (Fig. 1d). Moreover, Zip1 loading onto chromosomes remained unchanged in cells expressing a nuclease-deficient Yen1[ON] variant[35], Yen1[ON-ND] (Fig. 1e–g and Extended Data Fig. 1d,e), or in control cultures in which Yen1[ON] or Yen1[ON-ND] were not induced (Extended Data Fig. 1f,g). Similar findings were obtained by following SC dynamics in living cells using Zip1[GFP], where expression of Yen1[ON], but not Yen1[ON-ND], promoted the stepwise loss of chromosome synapsis (Extended Data Fig. 1h–k and Supplementary Videos 1 and 2).

We next expanded our analysis to other components of the SC. Immunostaining of the central element protein complex Ecm11–Gmc2 (refs. 37,38), as well as SC-associated Smt3 (also known as SUMO)[38], revealed analogous outcomes to Zip1, with progressive loss of chromosome association and accumulation into large aggregates linked to Yen1[ON] expression (Extended Data Fig. 2a–c). In budding yeast, disassembly of the SC central element during pachytene exit coincides with the loss of chromosome axis components and partial release of Rec8-containing cohesin, a process driven by Ndt80-dependent expression of polo kinase Cdc5 (refs. 31,32,39). After Yen1[ON] expression, Rec8 remained localized to chromatin, retaining a linear pattern of accumulation even in regions in which chromosome synapsis was lost (Extended Data Fig. 2d,e). However, we did notice that a substantial fraction of nuclei exhibited partially disorganized Rec8 threads, possibly due to the local separation of homologous chromosomes (Extended Data Fig. 2e (light pink)). In agreement with this view, super-resolution stimulated emission depletion (STED) microscopy revealed frequent chromosome axis splitting, which correlated with Zip1 loss (Extended Data Fig. 2f). These findings suggest that resolution of recombination intermediates leads to the loss of chromosome synapsis without significantly disrupting the axis–loop organization of meiotic prophase I chromosomes.

Meiotic DSBs continue to form in pachytene-arrested *ndt80Δ* mutants, albeit at low levels compared with peak DSB formation during the leptotene stage[3,25,30] (see also Fig. 1i,j, with data at the *HIS4::LEU2* recombination hotspot[13]). Thus, downstream formation of nascent recombination intermediates could have a role in maintaining the SC structure. To examine this possibility, we performed conditional nuclear depletion of Rec104, an essential factor in meiotic DSB formation, using the rapamycin-dependent anchor-away system[40] (Fig. 1h). Addition of rapamycin to the pachytene-arrested cultures strongly reduced DSB levels and halted the accumulation of DNA joint molecules (Fig. 1i,j (light red) and Extended Data Fig. 3a–c). Moreover, analysis of DNA joint molecules in two-dimensional gels showed stabilization of four-armed recombination intermediates containing dHJs, as well as three-armed single-end intermediates that are likewise crossover-designated[2,13] (Extended Data Fig. 3d,e (light red)). Importantly, inhibition of DSB formation did not severely interfere with chromosome synapsis, even though we observed that a slightly higher proportion of nuclei contained partial Zip1 synapsis (Fig. 1k (light green)). Expression of Yen1[ON] after DSB inhibition by Rec104[FRB] depletion resulted in the resolution of all remaining recombination intermediates at *HIS4::LEU2* and a complete loss of Zip1 from chromosomes (Fig. 1l,m (dark red) and Extended Data Fig. 3d–g (dark red)). These data suggest that DSB formation, although initially required for SC assembly, is largely dispensable for SC maintenance. Instead, SC maintenance relies on long-lived recombination intermediates, most likely containing dHJs.

## dHJ–ZMM interplay maintains the SC

As some ZMM proteins are known to preferentially bind to HJ DNA in vitro, and all localize to future crossover sites in vivo[15–21,23], we posited that Yen1[ON]-mediated cleavage of dHJs could lead to loss of ZMMs from chromosomes. In agreement, we observed a significant loss of chromosome-associated Zip3, Msh5 and Zip4 foci, which accumulated in polycomplexes after Yen1[ON] expression (Fig. 2a,b and Extended Data Fig. 4a–e). The three selected ZMMs represent functional subgroups: Msh5 forms with Msh4 the MutSγ complex, Zip4 forms with Zip2 and Spo16 a functional ZMM subcomplex called ZZS, and Zip3 is an E3 ligase[23]. It is therefore likely that all ZMM proteins directly or indirectly require recombination intermediates for continued chromosome association during pachytene.

We next sought to test whether ZMMs are continuously required for the maintenance of chromosome synapsis. To this end, we generated auxin-inducible degron (AID) alleles of *ZIP3*, *MSH4* and *ZIP4* (Fig. 2c). Addition of 5-Ph-IAA to pachytene-arrested cultures triggered rapid degradation of Zip3[AID], Msh4[AID] and Zip4[AID], resulting in their efficient depletion from chromosomes (Fig. 2d,e and Extended Data Fig. 4f–i). Notably, this resulted in a complete loss of chromatin-associated Zip1, as well as a more disorganized pattern of Rec8 accumulation, indicative of synapsis defects and homologue separation (Fig. 2e,f and Extended Data Fig. 4h–o). These results show that, in addition to their previously described roles in promoting SC assembly, ZMMs are continuously required for SC maintenance.

## ZMMs prevent dHJ dissolution by STR

During meiosis, ZMMs have been shown to protect nascent recombination intermediates from disassembly by the Sgs1–Top3–Rmi1 (STR) complex[41], but it is unclear whether their function is similarly required to suppress STR-mediated dHJ dissolution[42]. Physical analysis of recombination revealed a rapid loss of all DNA joint molecules at the *HIS4::LEU2* recombination hotspot after Zip3[AID] or Msh4[AID] depletion (Fig. 2g,h and Extended Data Fig. 5a–c). Consistent with STR-mediated dHJ dissolution, depletion of Zip3[AID] or Msh4[AID] resulted in a specific increase in non-crossover products (Fig. 2h and Extended Data Fig. 5c), and simultaneous depletion of Zip3[AID] or Msh4[AID] and Sgs1[AID] resulted in the stabilization and accumulation of DNA joint molecules (Fig. 2i–l and Extended Data Fig. 5d–f). Note that depletion of Sgs1[AID] also led to a small reduction in the accumulation of crossovers in *ndt80Δ* mutants (Extended Data Fig. 5g–i). This observation is consistent with previous research showing that *ndt80Δ* mutants accumulate a small proportion of Sgs1-dependent crossovers[3,43,44]. The precise origin of these crossovers remains unclear. Overall, our findings demonstrate that ZMMs are not only required for the formation of crossover-designated recombination intermediates but are also crucial for their continued protection from dissolution into non-crossovers by the STR complex.

To determine whether recombination intermediates contribute to SC maintenance independently of ZMMs, we monitored Zip1 after combined depletion of Zip3[AID] and Sgs1[AID]. Cells depleted of Sgs1[AID] and Zip3[AID] showed loss of Zip1 from chromosomes, but with a significant delay compared to Zip3[AID] depletion alone (Fig. 2m,n; compare with Fig. 2e,f). Similar results were obtained by combining conditional depletion of Msh4[AID] and Sgs1[AID] (Extended Data Fig. 5j,k). Furthermore, live-cell imaging confirmed these observations, with the structured Zip1[GFP] signal persisting significantly longer after combined depletion of Zip3[AID] and Sgs1[AID] (Extended Data Fig. 5l–n and Supplementary Videos 3 and 4). We then hypothesized that the delayed loss of chromosome synapsis might be due to partial retention of the ZMM subcomplex ZZS on chromosomes, as it can directly bind to recombination

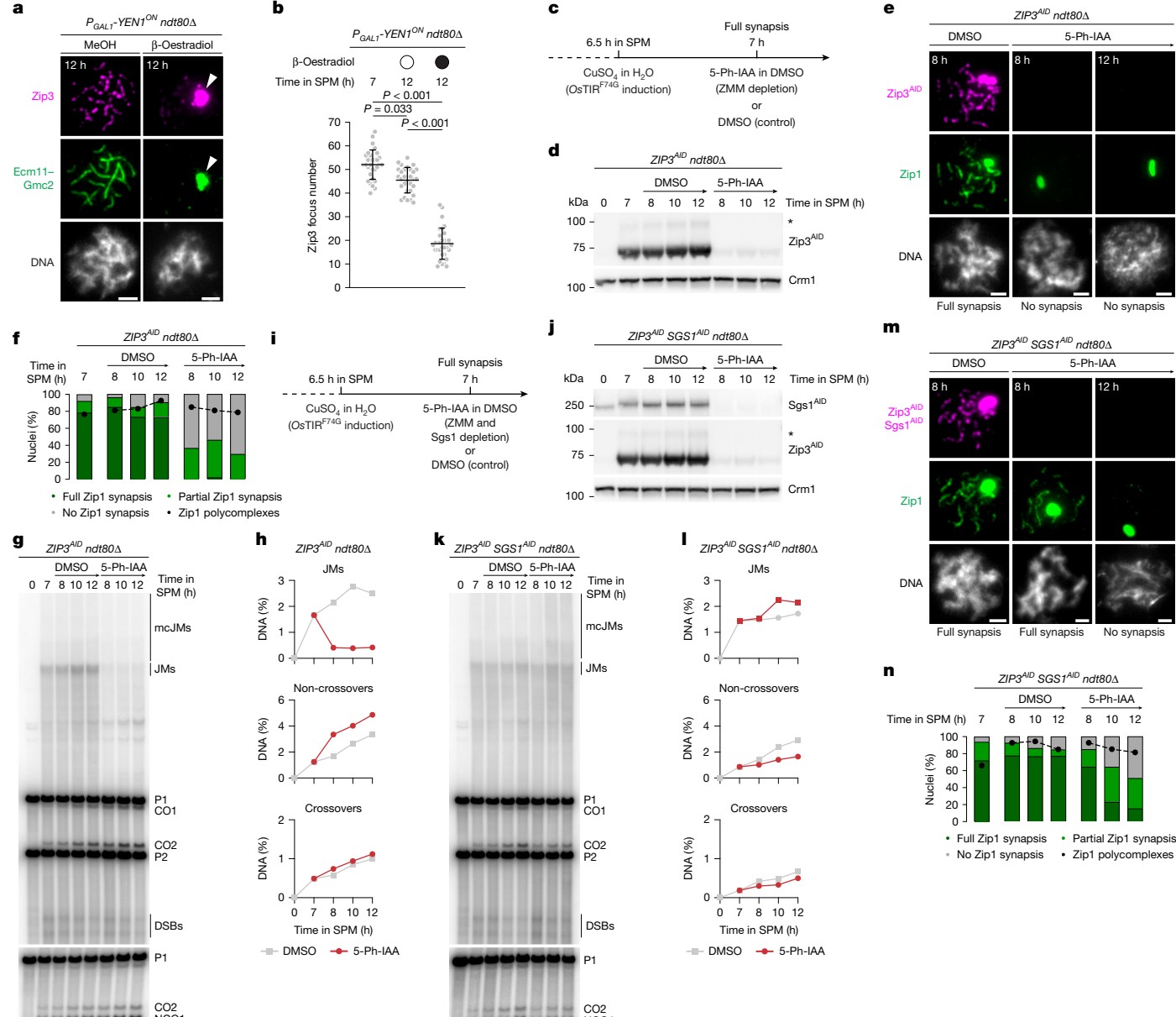

**Fig. 2 | dHJ–ZMM protein interplay maintains chromosome synapsis and protects crossover precursors. a**, Representative chromosome spreads at the indicated times in SPM, after Yen1[ON] induction by β-oestradiol addition (or methanol control) at 7 h, immunostained for Ecm11–Gmc2 (green) and Zip3 (magenta). The arrowheads mark Zip3 localizing to the Ecm11–Gmc2 polycomplex. **b**, Quantification of Zip3 focus number in **a**. Data are mean ± s.d. $n = 30$ nuclei per timepoint. Statistical analysis was performed using Kruskal–Wallis tests with Dunn's multiple-comparison test ($P \leq 0.001$). Representative of two biological replicates. **c**, The experimental set-up for conditional depletion of a ZMM protein in pachytene-arrested $ndt80\Delta$ cells. *Os*, *Oryza sativa*. **d**, Western blot analysis of Zip3[AID] levels in cells at the indicated times in SPM, treated as described in **c**. Crm1 was used as the protein loading control. The asterisks indicate putative SUMOylated Zip3. Representative of two biological replicates. **e**, Representative images of chromosome spreads from **d**, immunostained for Zip1 (green) and Zip3[AID] (magenta). *ZIP3[AID]* cells exhibit frequent polycomplex formation but show normal spore viability (Supplementary Table 2). **f**, Quantification of Zip1 synapsis and polycomplexes from **e**. $n = 50$ nuclei per timepoint, representative of two biological replicates. **g,h**, Southern blot (**g**) and quantification of joint molecules and non-crossover (NCO1) and crossover (CO2) products (**h**) at the *HIS4::LEU2* recombination hotspot from **d**. mcJM, multichromatid DNA joint molecule; P1, parental 1; P2, parental 2. **i**, The experimental set-up for conditional depletion of a ZMM protein and Sgs1 in pachytene-arrested $ndt80\Delta$ cells. **j**, Western blot analysis of Zip3[AID] and Sgs1[AID] levels as in **d** for *ZIP3[AID]SGS1[AID]* cells treated as described in **i**. Representative of two biological replicates. **k,l**, Southern blot (**k**) and quantification of joint molecules and non-crossover and crossover products (**l**) as in **g** and **h**, respectively, for the experiment in **j**. **m**, Representative images of chromosome spreads as in **e**, for the experiment in **j**. Both Zip3[AID] and Sgs1[AID] carry the Myc epitope tag. **n**, Quantification of Zip1 synapsis and polycomplexes as in **f**, for the experiment in **m**. $n = 50$ nuclei per timepoint, representative of two biological replicates. For **a**, **e** and **m**, scale bars, 2 μm.

intermediates[20,21] and has been directly implicated in synapsis initiation[22]. In support of this model, combined depletion of Zip4[AID] and Sgs1[AID] resulted in the rapid loss of Zip1 without delay (Extended Data Fig. 5o–q; compare with Extended Data Fig. 5j,k).

Taken together, these observations suggest that dHJs are required for the continuous association of ZMMs with chromosomes. In turn, ZMMs prevent dHJ dissolution by non-crossover pathways, while also promoting SC maintenance.

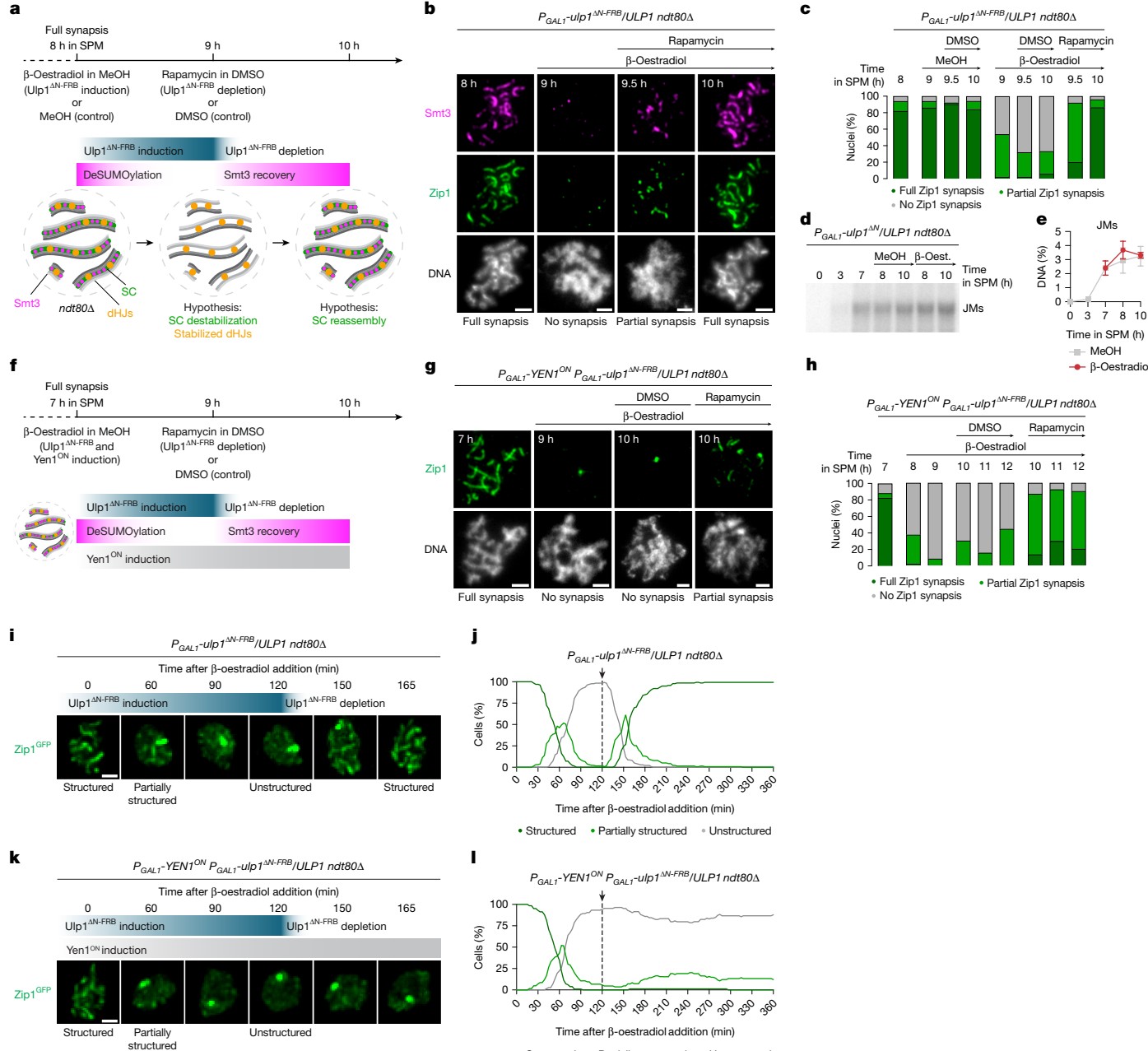

**Fig. 3 | dHJs enable reversible SC disassembly. a**, The experimental set-up for reversible SC disassembly in pachytene-arrested *ndt80Δ* cells through conditional protein deSUMOylation using the Smt3 isopeptidase mutant *ulp1^ΔN-FRB^*. **b**, Representative images of meiotic chromosome spreads from the experiment in **a** at the indicated times in SPM, immunostained for Zip1 (green) and Smt3 (magenta). **c**, Quantification of Zip1 synapsis from **b**. *n* = 50 nuclei per timepoint, representative of two biological replicates. **d,e**, Southern blot (**d**) and quantification of joint molecules (**e**) at the *HIS4::LEU2* recombination hotspot. Ulp1^ΔN^ was induced by β-oestradiol (β-oest.) addition (or methanol as a control) at 7 h in SPM. Data are the mean and range of two biological replicates. Gel source and biological replicate data are provided in Extended Data Fig. 6e. **f**, The experimental set-up to investigate whether dHJs are required for SC reassembly. **g**, Representative images of meiotic chromosome spreads from

the experiment in **f** at the indicated times in SPM, immunostained for Zip1 (green). **h**, Quantification of Zip1 synapsis from **g** (*n* = 50 nuclei per timepoint). **i**, Time-lapse image montage of Zip1^GFP^ in a cell nucleus after β-oestradiol-induced Ulp1^ΔN-FRB^ expression (*t* = 0 min; -7 h in SPM) and rapamycin-induced nuclear depletion (*t* = 120 min). Corresponds to Supplementary Video 6. **j**, Quantification of structured Zip1^GFP^ signal from **i**. The arrow and dashed line indicate the timepoint of rapamycin addition. Data are the mean of two biological replicates. *n* = 40 cells each. **k**, Time-lapse image montage as in **i**, for the simultaneous expression of Ulp1^ΔN-FRB^ and Yen1^ON^ by β-oestradiol addition. Corresponds to Supplementary Video 7. **l**, Quantification of structured Zip1^GFP^ signal as in **j**, for the experiment in **k**. Data are the mean of two biological replicates. *n* = 40 cells each. Scale bars, 2 μm (**b** and **g**) and 1 μm (**i** and **k**).

## dHJs enable reversible SC disassembly

In budding yeast, paired chromosome axes retain the ability to assemble the SC structure even in late prophase I[6]. To test whether dHJs function to stabilize sites of synapsis initiation to enable continuous reinitiation

of SC polymerization and maintenance of the SC structure, we developed a molecular tool to reversibly disassemble the SC without loss of dHJs (Fig. 3a). Conceptually, this tool takes advantage of Smt3 constituting a reversible post-translational protein modifier with essential roles in SC assembly[37,38,45] as well as in dHJ processing[46,47]. In the first of

two steps, we examined whether conditional protein deSUMOylation is sufficient to disassemble the SC in pachytene-arrested cells. To address this, we generated $ndt80\Delta$ strains conditionally expressing a truncated version of the Smt3 isopeptidase Ulp1 (ref. 48) (Ulp1$^{\Delta N}$; Extended Data Fig. 6a). In contrast to Ulp1, Ulp1$^{\Delta N}$ is unable to associate with the nuclear pore complex, thereby extending its activity beyond the nuclear periphery. In addition to placing $ulp1^{\Delta N}$ under an β-oestradiol-inducible promoter, we also fused $ulp1^{\Delta N}$ to the FKBP12–rapamycin-binding (FRB) domain ($ulp1^{\Delta N\text{-}FRB}$), which was necessary in a subsequent second step to enable nuclear depletion using the anchor-away system (Extended Data Fig. 6b). Induction of Ulp1$^{\Delta N\text{-}FRB}$ expression using β-oestradiol resulted in a rapid decrease in global Smt3-modified protein levels, including polySUMOylated forms of Ecm11 (Extended Data Fig. 6c)—one of the most prominent substrates for polySUMOylation during budding yeast meiosis[37,45]. As a result, we observed complete loss of Smt3 and Zip1 from chromosomes within 1 h of Ulp1$^{\Delta N\text{-}FRB}$ induction (Fig. 3b,c and Extended Data Fig. 6d). Central to our approach, we found that expression of Ulp1$^{\Delta N}$ did not alter the level of DNA joint molecules at $HIS4::LEU2$ (Fig. 3d,e and Extended Data Fig. 6e) and the co-alignment of homologue axes was largely maintained (Extended Data Fig. 6f,g). In a second step, we tested whether the subsequent nuclear depletion of Ulp1$^{\Delta N\text{-}FRB}$ would enable the re-establishment of the SC. Notably, addition of rapamycin to the cultures resulted in the recovery of protein SUMOylation and reassembly of Zip1 along chromosomes within 30 min (9.5 h in SPM), and complete restoration of the SC structure within 1 h (10 h in SPM) (Fig. 3b,c and Extended Data Fig. 6c,d). Ulp1$^{\Delta N\text{-}FRB}$ expression did not noticeably alter DSB levels (Extended Data Fig. 6e), and SC reassembly did not depend on newly arising DSBs, as experimentally confirmed by monitoring Zip1 synapsis after Rec104$^{AID}$ depletion (Extended Data Fig. 6h–j).

To experimentally test whether dHJs are required for SC reassembly, Yen1$^{ON}$ was expressed simultaneously with Ulp1$^{\Delta N\text{-}FRB}$ (Fig. 3f). Under these conditions, most nuclei failed to re-establish full Zip1 synapsis, even 4 h after nuclear depletion of Ulp1$^{\Delta N\text{-}FRB}$ (12 h in SPM) (Fig. 3g,h and Extended Data Fig. 7a–c). We further confirmed these findings by visualizing SC disassembly and reassembly in single cells. Live-cell imaging of Zip1$^{GFP}$ showed reversible SC disassembly by Ulp1$^{\Delta N\text{-}FRB}$ expression/nuclear depletion in all analysed cells (Fig. 3i,j, Extended Data Fig. 7d,e and Supplementary Videos 5 and 6). Moreover, SC reassembly was severely disrupted when Yen1$^{ON}$ was co-expressed with Ulp1$^{\Delta N\text{-}FRB}$ (Fig. 3k,l and Supplementary Video 7). Overall, these data support a model in which dHJs are sufficient to maintain interhomologue connections in the absence of the central region of the SC. These connections enable the complete re-establishment of chromosome synapsis, presumably by providing a platform that positions ZMM proteins at the axis interface of homologue pairs. In support of this interpretation, we found that, while many nuclei contained Zip3 aggregates after Ulp1$^{\Delta N\text{-}FRB}$ expression, the number of Zip3 foci remained largely unchanged (Extended Data Fig. 7f–h). Furthermore, Yen1$^{ON}$ expression resulted in a marked reduction in the number of Zip3 foci (Extended Data Fig. 7i–k), indicating that Zip3 associates with dHJs and that its retention—and, by extension, the retention of other ZMMs—underlies the rapid SC reassembly after Ulp1$^{\Delta N\text{-}FRB}$ depletion.

## dHJ–ZMM feedback limits DSB formation

In organisms in which DSB formation is required for chromosome synapsis, such as budding yeast and mice, SC assembly is thought to suppress the formation of new DSBs on chromosomes that have already successfully engaged in crossover repair[24–29]. This feedback control involves the displacement of HORMAD proteins from chromosome axes, leading to the loss of a DSB-competent state[24,26]. We therefore hypothesized that SC maintenance through the functional interplay between dHJs and ZMMs might be important for continued downregulation of DSB formation. Indeed, Yen1$^{ON}$ expression in

pachytene-arrested cultures led to reaccumulation of Hop1 (also known as HORMAD) on meiotic chromosomes that had lost Zip1 (Fig. 4a,b). This was accompanied by increased Hop1 phosphorylation at Thr318 by the DNA-damage response kinases Mec1 (ATR) and Tel1 (ATM), which occurs in response to meiotic DSB formation[49] (Fig. 4c). We confirmed this inference by directly monitoring DSBs at two recombination hotspots ($CCT6$ and $ERG1$) using Southern blotting, with DSB levels increasing by around 3–4-fold after Yen1$^{ON}$ induction (Fig. 4d,e and Extended Data Fig. 8a–d). We predicted that the loss of chromosome synapsis after conditional ZMM depletion during pachytene should also result in the re-establishment of a competent state for DSB formation. Indeed, we observed an increase in Hop1 Thr318 phosphorylation after auxin-mediated depletion of Zip3$^{AID}$, Msh4$^{AID}$ and Zip4$^{AID}$, and an approximately 3–6-fold increase in DSB levels after Msh4$^{AID}$ depletion (Fig. 4f–h and Extended Data Fig. 8e–h). These findings suggest that dHJs and ZMM proteins have a key role in maintaining a suppressive state for DSB formation by contributing to the maintenance of chromosome synapsis.

## Premature loss of dHJs impairs meiosis I

To determine whether recombination intermediates are required for SC maintenance during meiotic progression in wild-type ($NDT80$) cells, we combined timed Yen1$^{ON}$ expression with live-cell imaging of Zip1$^{GFP}$. We chose to initiate Yen1$^{ON}$ expression around 5 h after induction of meiosis as DNA replication was largely completed (Extended Data Fig. 9a) and cells in different substages of prophase I could be identified on the basis of the Zip1$^{GFP}$ signal: pre-leptotene/leptotene cells lacking structured Zip1$^{GFP}$; early zygotene cells with a partially structured Zip1$^{GFP}$ signal; and late zygotene/pachytene cells with long Zip1$^{GFP}$ threads throughout the nucleus. Under our imaging conditions, zygotene lasted about 55 min, pachytene around 115 min and anaphase I was initiated around 45 min after exit from pachytene (Extended Data Fig. 9b and Supplementary Video 8). As robust accumulation of Yen1$^{ON}$ requires 45–60 min (Extended Data Fig. 9c,d), and significant processing of DNA joint molecules by Yen1$^{ON}$ becomes detectable only after 1–2 h (Extended Data Fig. 3f,g), we decided to follow the subpopulation initially in early to mid-zygotene at the time of Yen1$^{ON}$ induction to determine the impact of Yen1$^{ON}$ specifically during pachytene. As predicted, cells induced to express Yen1$^{ON}$ in early to mid-zygotene successfully entered pachytene and reached full chromosome synapsis (Fig. 4i,j and Supplementary Video 9). However, 90% of cells reverted from full chromosome synapsis to a state of partial synapsis, with conspicuous Zip1$^{GFP}$ aggregates (Fig. 4i,j). Notably, we also found that cells that reverted to a state of partial synapsis showed a long delay in completing prophase I and eventually failed to undergo the first meiotic division and form spores (Fig. 4j–l and Supplementary Video 9). By contrast, control methanol-treated cultures disassembled the SC, completed meiosis I nuclear division and formed spores efficiently (Fig. 4k,l, Extended Data Fig. 9e,f and Supplementary Video 10). Importantly, Yen1$^{ON}$ expression did not interfere with SC assembly and disassembly, nor with nuclear division, in cells that were in pachytene at the time of induction (Extended Data Fig. 9g–i and Supplementary Video 11). However, Yen1$^{ON}$ expression prevented almost all pre-leptotene/leptotene cells from reaching a state of full synapsis and completely blocked meiosis I nuclear division (Extended Data Fig. 9j–l and Supplementary Video 12). To confirm these findings, we induced Yen1$^{ON}$ expression from the strong copper-inducible $P_{CUP1}$ promoter in pachytene-arrested $ndt80\Delta$ mutants and subsequently triggered pachytene exit by expressing Ndt80 from the $P_{GAL1}$ promoter. Cells expressing Yen1$^{ON}$ before Ndt80 induction disassembled Zip1$^{GFP}$ but retained a large polycomplex for a prolonged time (Extended Data Fig. 9m–p and Supplementary Videos 13 and 14). Eventually, the Zip1$^{GFP}$ signal was lost and cells attempted to undergo meiosis I nuclear division, as indicated by HTB1$^{mCherry}$ labelling of chromatin. However, most failed, while those that underwent anaphase I exhibited DNA bridges

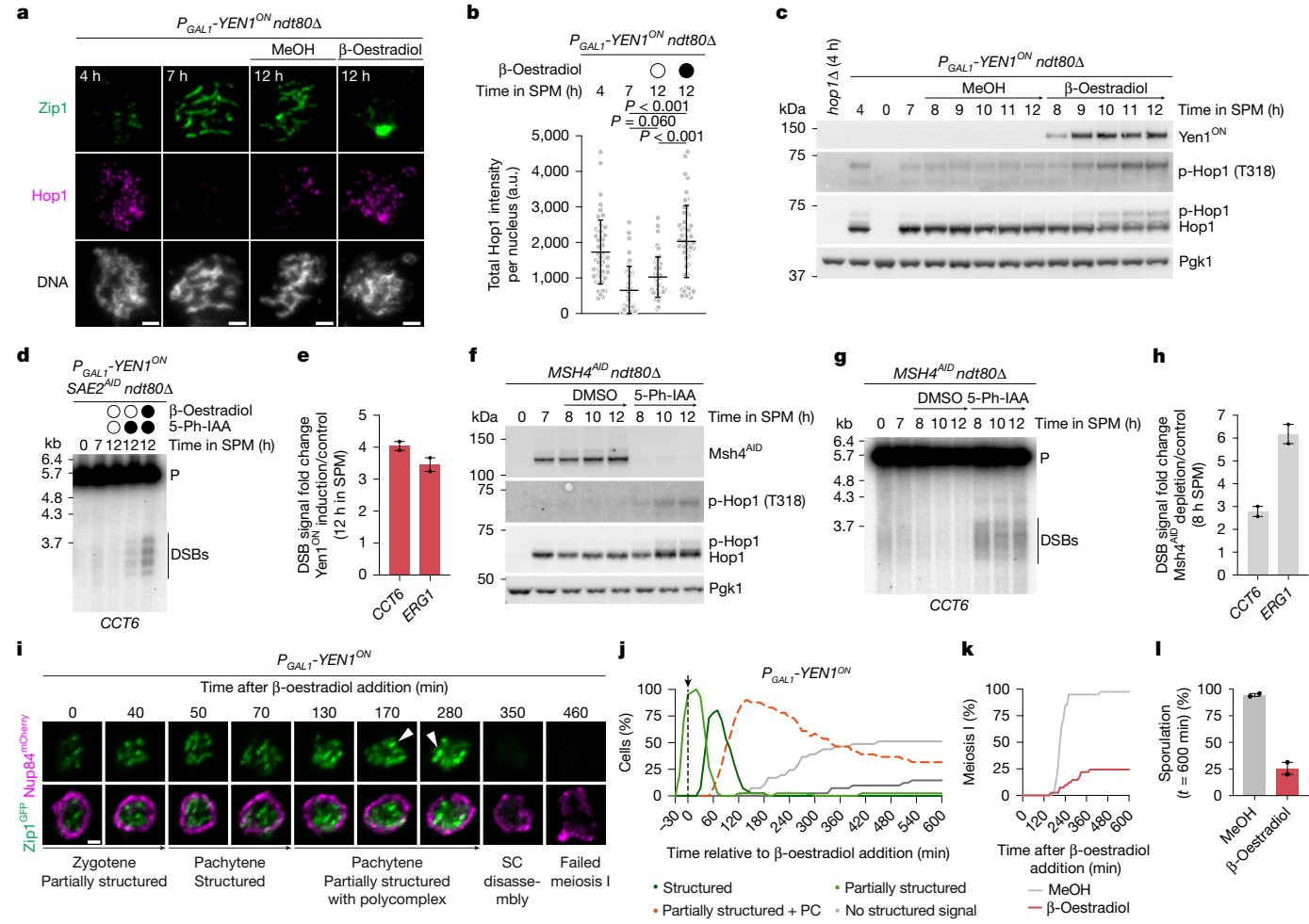

**Fig. 4 | dHJ–ZMM protein interplay suppresses DSB formation and promotes meiotic progression. a**, Representative images of chromosome spreads at peak DSB formation (4 h), and before (7 h) and after (12 h) Yen1[ON] induction by addition of β-oestradiol (or methanol control), immunostained for Zip1 (green) and Hop1 (magenta). **b**, The total Hop1 signal intensity per nucleus from **a**. Data are mean ± s.d. *n* = 50 nuclei per timepoint. Statistical analysis was performed using Kruskal–Wallis tests with Dunn's multiple-comparison test (*P* ≤ 0.001). **c**, Western blot analysis of the levels of Hop1 and Hop1 phosphorylated at Thr318 from **a**. p-Hop1, phosphorylated Hop1. Pgk1 was used as the protein loading control. **d**,**e**, Southern blot analysis of DSBs at the *CCT6* locus (**d**) and quantification at *CCT6* and *ERG1* (**e**). Yen1[ON] was induced by β-oestradiol addition (or methanol control) at 7 h in SPM. Sae2[AID] was simultaneously depleted by addition of 5-Ph-IAA (or DMSO control) to prevent DSB repair. Data are the mean and range of fold changes relative to the Sae2[AID] depletion control from

two biological replicates. Gel source and replicate data are provided in Extended Data Fig. 8c. P, parental. **f**, Western blot analysis as in **c** for Msh4[AID] depletion at 7 h in SPM. Representative of two biological replicates. **g**,**h**, Southern blot analysis of DSBs (**g**) and quantification of *CCT6* and *ERG1* (**h**), as in **d** and **e**, for the experiment in **f**. Gel source and replicate data are provided in Extended Data Fig. 8e. **i**, Time-lapse montage of Zip1[GFP] and Nup84[mCherry] after Yen1[ON] induction in early/mid-zygotene (*t* = 0 min, -5 h in SPM). The arrowheads mark Zip1[GFP] aggregates. Nup84[mCherry] brightness was adjusted for signal visibility. Corresponds to Supplementary Video 9. **j**–**l**, Quantification of structured Zip1[GFP] signal (**j**), meiosis I (**k**) and sporulation (**l**) from the experiment in **i**. The arrow and dashed line indicate β-oestradiol addition. PC, polycomplex. Data are the mean and range (error bars) of two biological replicates, *n* = 20 cells each. Scale bars, 2 μm (**a**) and 1 μm (**i**).

or aberrant nuclear mass distribution (Extended Data Fig. 9q). Overall, these findings suggest that premature processing of recombination intermediates during wild-type meiosis leads to disruption of chromosome synapsis. As a consequence, cell cycle progression is disrupted, most likely due to the formation of de novo DSBs and their repair intermediates, which trigger DNA damage signalling[50,51], as observed after Yen1[ON] expression or ZMM depletion in *ndt80Δ* mutants (Fig. 4a–h and Extended Data Fig. 8).

## dHJ resolution promotes SC disassembly

SC disassembly during exit from pachytene is temporally coordinated with HJ resolution through Ndt80-mediated expression of polo kinase Cdc5 (also known as PLK)[31,32]. However, it is unclear whether HJ processing contributes to SC disassembly. To test this possibility, we analysed

meiotic cells lacking all four HJ resolvases: *mlh3Δ mms4[mn] slx1Δ yen1Δ*, hereafter, quadruple-resolvase mutant. First, we confirmed that ectopic expression of Cdc5 was sufficient to induce Zip1 loss from the chromosomes of pachytene-arrested *ndt80Δ* cells, as previously reported[32] (Fig. 5a,b (left) and Extended Data Fig. 10a (left)). Notably, quadruple-resolvase mutants retained chromosome-associated Zip1 for a prolonged period (Fig. 5a,b (right) and Extended Data Fig. 10a (right)), despite a significant reduction in Zip1 protein levels as observed in the control cells (Extended Data Fig. 10b,c). Live-cell imaging of Zip1[GFP] confirmed the delayed loss of chromosome synapsis in quadruple-resolvase mutant cells after Cdc5 expression (Fig. 5c–e and Supplementary Videos 15 and 16). As DNA joint molecules in the quadruple-resolvase mutant may be processed alternatively by the STR pathway, we combined a meiotic-null allele of *SGS1* (*sgs1[mn]*)[41] with the quadruple-resolvase mutant. In this quintuple mutant, in which DNA

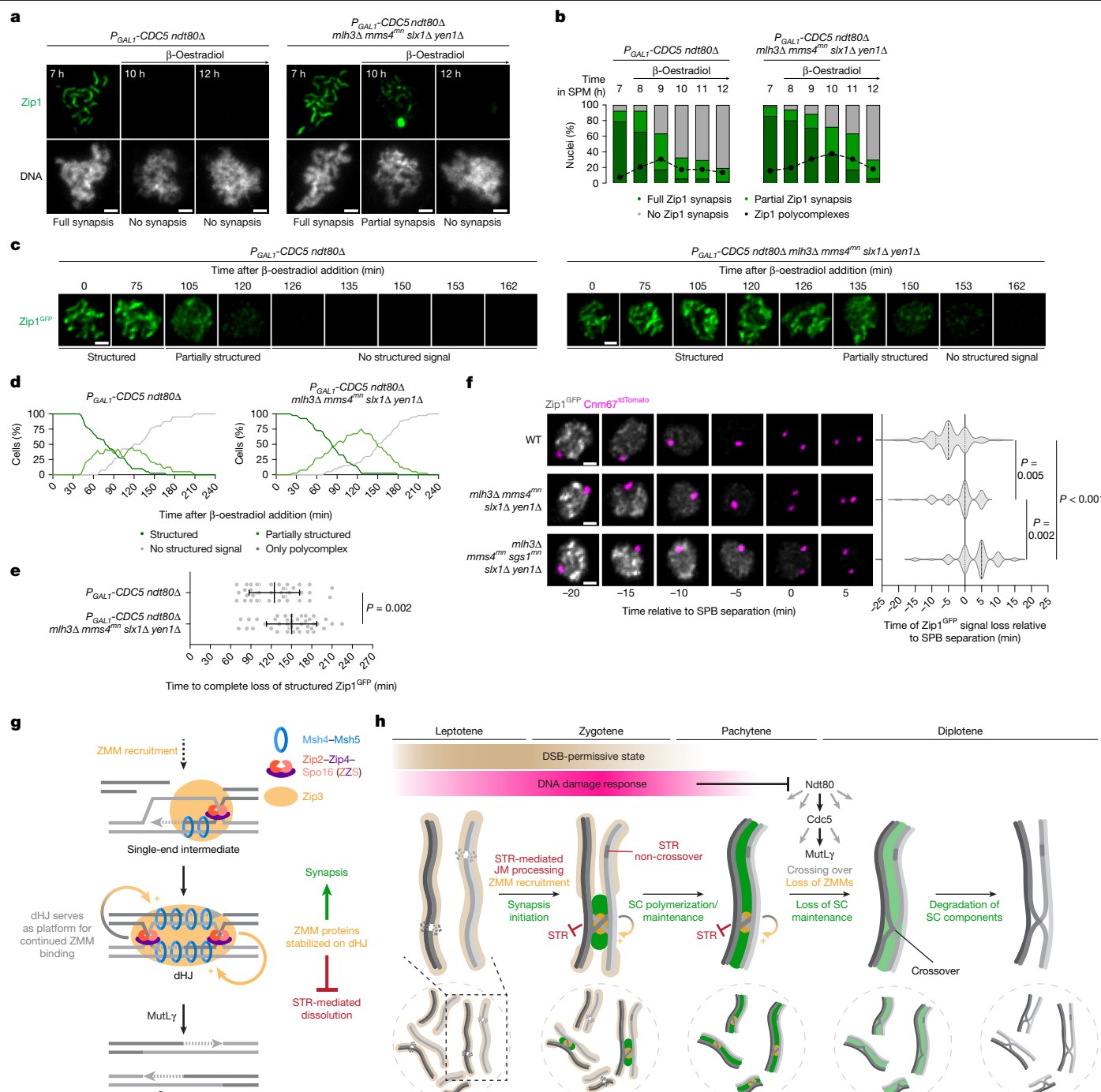

**Fig. 5 | Cdc5 promotes SC disassembly partly through the activation of HJ resolvases. a**, Representative chromosome spreads at the indicated times in SPM, with immunostaining for Zip1 (green). Cdc5 was induced by β-oestradiol addition at 7 h. **b**, Quantification of Zip1 synapsis and polycomplex formation from **a**. $n = 50$ nuclei per timepoint. **c,d**, Time-lapse montage of Zip1$^{GFP}$ (**c**) and quantification of structured Zip1$^{GFP}$ (**d**) in cells of the indicated genotypes after Cdc5 induction by β-oestradiol addition at around 7 h in SPM ($t = 0$ min). $n = 40$ cells. See Supplementary Videos 15 and 16. **e**, The time to complete loss of structured Zip1$^{GFP}$ signal from **d**. Data are mean ± s.d. $n = 40$ cells per genotype. Statistical analysis was performed using two-tailed unpaired Mann–Whitney $U$-tests. **f**, Time-lapse montage of Zip1$^{GFP}$ and Cnm67$^{tdTomato}$ in cells of the indicated genotypes (left) and quantification of Zip1$^{GFP}$ signal loss relative to SPB separation (right). Data are the median (dashed line), and first and third

quartiles (dotted lines). $n = 60$ cells per genotype. Statistical analysis was performed using Kruskal–Wallis tests with Dunn's multiple-comparison test ($P \leq 0.001$). **g,h**, dHJ–ZMM protein interplay coordinates meiotic progression with crossover assurance. **g**, ZMMs promote the formation of dHJs, which maintain ZMM association at homologue axis interfaces. In turn, ZMMs protect dHJs from STR-mediated dissolution and promote SC assembly and maintenance. MutLγ resolves dHJs into crossovers, contributing to ZMM displacement and loss of SC maintenance. **h**, SC assembly induces chromosome-autonomous and, eventually, nucleus-wide DSB downregulation. DSB repair completion silences the DNA damage response, enabling Ndt80-mediated pachytene exit without unrepaired DSBs and at least one dHJ per homologue pair. Ndt80-induced Cdc5 activates HJ resolvases, displacing ZMMs and destabilizing the SC to promote rapid SC disassembly. Scale bars, 2 μm (**a**) and 1 μm (**c** and **f**).

joint molecules remain unprocessed and accumulate to high levels[44], SC disassembly was further delayed compared with the control and quadruple-resolvase mutant but still completed in about 50% of cells within the experimental timeframe (Extended Data Fig. 10d–j and Supplementary Video 17). Next, we examined whether HJ processing contributes to SC disassembly at the prophase-I-to-metaphase-I transition in *NDT80* cells. Notably, SC disassembly was delayed relative to spindle pole body (SPB) separation in both the quadruple and the quintuple mutants (Fig. 5f). Nonetheless, the Zip1[GFP] signal was eventually lost in almost all nuclei analysed (Fig. 5f). The SC disassembly delay was confirmed by the analysis of the Zip1 signal relative to SPB separation on chromosome spreads. At the onset of metaphase I, around 57% of quadruple-resolvase mutants retained Zip1 stretches compared with around 28% of controls (Extended Data Fig. 10k). This effect was more prominent in quintuple mutants (~79%), with approximately 55% still showing Zip1 stretches in nuclei with fully separated SPBs. In summary, these findings suggest that Cdc5 promotes SC disassembly through two distinct mechanisms. One is linked to the processing of HJs, which disrupts SC maintenance, whereas the other probably involves Cdc5-mediated phosphorylation of SC components leading to their degradation, as previously suggested[52,53].

## Discussion

Our study and the accompanying work by Tang et al.[54] support a model in which reciprocal feedback between recombination intermediates and ZMM proteins stabilizes both DNA-based and protein-based inter-homologue connections throughout the extended meiotic prophase I (Fig. 5g). Mechanistically, ZMM proteins protect a subset of recombination intermediates from STR-mediated dissolution into non-crossovers. Conversely, these long-lived recombination intermediates function as platforms for the continued association of ZMM proteins with chromatin. Importantly, by stabilizing and precisely positioning ZMMs at the axes interface of recombining homologue pairs, recombination intermediates have a key role in the establishment and maintenance of chromosome synapsis, a process that we postulate involves continuous polymerization of the SC. Our study also suggests that dHJ–ZMM-mediated SC maintenance contributes to the stable suppression of DSB formation on recombining chromosomes (Fig. 5h). Consistent with this view, premature resolution of recombination intermediates during pachytene triggers SC disassembly, de novo DSB formation and DNA damage signalling, ultimately disrupting entry into the first meiotic division.

More broadly, we propose that the reciprocal interplay between dHJs and ZMM proteins constitutes a regulatory circuit that coordinates exit from prophase I with crossover assurance (Fig. 5h). In this model, dHJ-dependent maintenance of chromosome synapsis provides a straightforward mechanism to couple meiotic cell cycle progression with the accumulation of crossover precursors. Only when dHJs accumulate successfully will the SC stably inhibit de novo DSB formation in a chromosome-autonomous manner. This inhibition is in turn necessary to suppress the DNA damage response throughout the genome, which would otherwise delay meiotic progression as long as DSBs or their repair intermediates are present[50,51]. dHJs possess a unique property that is central to this model: in contrast to DSBs and early DSB repair intermediates, which typically contain extensive regions of single-stranded DNA, late recombination intermediates containing dHJs are unlikely to activate the DNA damage response. This notion is supported by studies showing that HJ resolvase mutants exit prophase I in the presence of high levels of dHJs and undergo catastrophic chromosome segregation[33,43,44]. Thus, once ZMM-stabilized dHJs accumulate on all chromosomes and synapsis permanently downregulates DSB formation, cells can proceed to the first meiotic division assured of no unrepaired breaks and at least one crossover precursor per homologue pair. A crossover outcome is further enforced by dHJ-association of

specific ZMMs, in particular MutSγ, which is thought to mediate the orientation-specific loading and activation of MutLγ (Mlh1–Mlh3) to resolve dHJs in a crossover-specific manner[44,55,56]. It is interesting to consider that, by enabling cells to silence the DNA damage response, dHJs promote the cell cycle transition that leads to the activation of HJ resolvases, triggering their own turnover. For these reasons, we propose that dHJs are more than passive intermediate structures that form in the process of crossing-over. They have a central regulatory function in coordinating meiotic progression with crossover assurance. Our work also demonstrates that HJs can mediate reversible disassembly of the SC. This reversibility may facilitate the resolution of chromosomal interlocks and contribute to the destabilization of interactions between non-allelic sequences.

We envision that related principles may operate in organisms such as plants or mice, in which DSB formation precedes chromosome synapsis[12]. In other organisms, such as worms and flies, in which SC assembly occurs independently of DSB formation[57,58], changes in SC dynamics have been reported to occur in response to the stabilization of crossover precursors[7–9,59–61]. Although dHJs may not be required for SC maintenance in these organisms, they may have a similar key regulatory role in coordinating the exit from prophase I with crossover assurance.

Finally, our findings suggest that the resolution of recombination intermediates is part of the SC-disassembly process (Fig. 5h). Although further work is required to dissect the events leading to SC disassembly, we envision that the trigger for this substantial change in chromosome organization involves at least two distinct processes driven by polo kinase Cdc5: (1) dHJ resolution, which destabilizes ZMM association and, as such, eliminates synapsis (re)initiation sites; and (2) phosphorylation-dependent degradation of SC components, which probably destabilizes the SC more broadly[52,53].

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

## Methods

### Strain construction

*Saccharomyces cerevisiae* strains used in this study were SK1 derivatives, as described in Supplementary Table 1. The following alleles have been previously described: *ndt80Δ*[62], *YEN1ON* (ref. 34), *YEN1ON-ND* (ref. 35), *P_{GAL1}-YEN1ON-3FLAG-2TEV-10HIS*[35], *P_{GPD1}-GAL4(848).ER*[63], *HIS4::LEU2* alleles for physical analysis of recombination[13], *REC104-FRB-3HA*[40], *fpr1Δ*[64], *RPL13A-FKBP12* (ref. 64), *P_{CUP1}-OsTIR1^{F74G}* (refs. 65,66), *P_{CLB2}-3HA-MMS4* (ref. 33), *P_{CLB2}-3HA-SGS1* (ref. 41), *yen1Δ*[35], *mlh3Δ*[35], *P_{GAL1}-CDC5-3HA*[62], *P_{GAL1}-NDT80* (ref. 63) *CNM67-tdTomato*[62], *HTB1-mCherry*[62] and *ZIP1::GFP^{700}* (marked with *HphMX4* 170 bp after the stop codon)[62,67]. *YEN1ON-ND* is a nuclease-dead version of *YEN1ON* (ref. 35), carrying the E193A and E195A mutations[34]. To generate *P_{GAL1}-YEN1ON* and *P_{GAL1}-YEN1ON-ND*, the endogenous promoter of C-terminally *18myc*-tagged *YEN1ON* or *YEN1ON-ND*[35] was replaced with the *P_{GAL1}* cassette, PCR-amplified from plasmid pYM-N22 (ref. 68). Similarly, *P_{CUP1}-YEN1ON* was constructed by replacing the endogenous promoter of C-terminally *18myc*-tagged *YEN1ON* with the *P_{CUP1}* cassette, PCR-amplified from plasmid pYM-N1 (ref. 68). Nup84 was C-terminally tagged with mCherry (*NUP84^{mCherry}*) by integrating the *3xmCherry* cassette, PCR-amplified from plasmid pFA6a-3xmCherry-KanMX4 (ref. 69). C-terminal tagging of *ZIP3*, *MSH4*, *ZIP4*, *SGS1*, *REC104* and *SAE2* with the AID tag was performed by integration of the *AID*–9myc* cassette (here referred to as *AID*; AID* comprises IAA17 (amino acids 71–114)), PCR-amplified from plasmid pHyg-AID*–9myc[70]. *ZIP4* was *9myc*-tagged at the C terminus (*ZIP4^{9myc}*) by integration of the *9myc-KITRP1* cassette, PCR-amplified from plasmid pWZV86-9myc-KITRP1 (pWZV87) as described previously[69]. The functionality of the tagged proteins was validated by tetrad microdissection and assessment of spore viability, as described in Supplementary Table 2. *ulp1^{ΔN}* was first published as *ΔNulp1-GFP*, in which part of the nuclear pore complex anchoring domain (amino acids 172 to 340) was replaced by GFP[48]. C-terminal tagging of *ulp1^{ΔN}* with the FKBP12–rapamycin-binding domain (*FRB*) was performed by integration of a *FRB* cassette, PCR-amplified from the plasmid pFA6a-FRB-KanMX6 (ref. 64). *ULP1* is an essential gene, and therefore strains carrying β-oestradiol-inducible *ulp1^{ΔN}* or *ulp1^{ΔN-FRB}* (*P_{GAL1}-ulp1^{ΔN}*, *P_{GAL1}-ulp1^{ΔN-FRB}*) were generated by replacing the endogenous promoter in a heterozygous strain with the *P_{GAL1}* cassette, PCR-amplified from plasmid pYM-N23 (ref. 68). In meiotic experiments using the rapamycin-dependent anchor-away system[64], the *TOR1* mutation *tor1-1* to enhance rapamycin resistance is omitted from the strains. *slx1Δ* was generated by replacing the *SLX1* coding region with the selectable marker *hphMX6*, PCR-amplified from plasmid pFA6a-hphMX6 (ref. 71). Further details on strain construction and primer sequences used are available on request.

### Meiotic time courses

Meiotic time courses were performed with diploid SK1 strains as previously described[72]. In brief, cells selected on YP-glycerol plates (20 g l$^{-1}$ bactopeptone, 10 g l$^{-1}$ yeast extract, 2% (v/v) glycerol, 20 g l$^{-1}$ agar) for 48 h at 30 °C were plated in small patches on YPD plates (20 g l$^{-1}$ bactopeptone, 10 g l$^{-1}$ yeast extract, 20 g l$^{-1}$ dextrose, 20 g l$^{-1}$ agar) and grown for about 24 h at 30 °C. Cells were further expanded on YPD plates to form a lawn covering the entire plate, grown for around 24 h at 30 °C and inoculated into pre-sporulation medium (YPA; 20 g l$^{-1}$ bactopeptone, 10 g l$^{-1}$ yeast extract, 2% (w/v) potassium acetate) to an optical density at 600 nm (OD$_{600}$) of about 0.3. Cells were grown and arrested in G1 for 14 h at 25 °C, washed with prewarmed sporulation medium (SPM; 2% (w/v) potassium acetate) and inoculated into SPM to an OD$_{600}$ of about 3.5. This time defines 0 h in all meiotic time-course experiments. Large meiotic cultures were prepared after scaling up the above protocol and using a 10 l fermenter system as previously described[72–74]. Progression through pre-meiotic DNA replication was followed by fluorescence-activated cell sorting (FACS) analysis of DNA content.

Pachytene arrest in *ndt80Δ* strains was defined as 7 h after induction of meiosis in SPM, unless otherwise indicated. Expression from *P_{GAL1}* through *GAL4(848).ER* was induced by addition of 2 μM β-oestradiol (dissolved in methanol), except for *P_{GAL1}-NDT80*, which was induced with 1 μM. Expression of Yen1ON from *P_{CUP1}* was induced by the addition of 2 μM CuSO$_4$ (dissolved in H$_2$O). To deplete AID-tagged proteins, *P_{CUP1}-OsTIR1^{F74G}* was induced by the addition of 50 μM CuSO$_4$ approximately 30 min before the addition of 100 μM 5-Ph-IAA (dissolved in DMSO). Anchor-away experiments were performed by addition of 1 μg ml$^{-1}$ rapamycin (dissolved in DMSO).

### FACS analysis of DNA content

Cellular DNA content was determined to monitor release from G1 arrest and entry into the pre-meiotic S phase. In brief, 1 ml of meiotic culture was collected and fixed in 70% (v/v) ice-cold ethanol. Cells were washed once and resuspended in 50 mM Tris-HCl pH 7.5. RNA was digested for 2–4 h at 37 °C by the addition of 2 μl RNase (100 mg ml$^{-1}$). Cells were washed once in FACS buffer (200 mM Tris-HCl pH 7.5, 211 mM NaCl, 78 mM MgCl$_2$), resuspended in FACS buffer containing 50 μg ml$^{-1}$ propidium iodide and sonicated briefly. An aliquot was diluted 10–20 times in 1 ml 50 mM Tris-HCl pH 7.5 and DNA content was measured using the FACSCalibur cytometer (BD Biosciences) controlled by the CellQuest software (BD Biosciences). Cytometer data were analysed using FlowJo software (BD Biosciences).

### Meiotic chromosome spreads

Yeast chromosome surface spreads were prepared as described in refs. 75,76, with some modifications. In brief, 1 ml of meiotic culture collected at the indicated timepoint was centrifuged (4 min at 700 rcf) and cells were resuspended in 200 μl spheroplasting solution (2% (w/v) potassium acetate, 0.8 M sorbitol). Cells were incubated with 10 mM DTT for 15 min at 30 °C and then digested with 5 μl Zymolyase 20T solution (10 mg ml$^{-1}$) for around 10 min. Spheroplasting efficiency was monitored by mixing an aliquot of 2 μl with 2 μl of 2% (w/v) sarcosyl, which should lyse the spheroplasted cells. The digestion was stopped by adding 400 μl of ice-cold stop solution (0.1 M MES, 1 mM EDTA, 0.5 mM MgCl$_2$, 1 M sorbitol, pH 6.4) and the spheroplasts were centrifuged (4 min at 700 rcf) and resuspended in 100 μl of stop solution. Immediately after, 20 μl of spheroplast suspension was processed on a clean glass slide by sequentially adding 40 μl of fixative (4% (w/v) paraformaldehyde, 3.4% (w/v) sucrose) for pre-fixation, 80 μl of 1% (v/v) Lipsol to initiate lysis and 80 μl of fixative for final fixation. The slides were dried overnight in a fume hood before immunostaining.

For immunostaining, slides were washed once for 15 min in 1× PBS and blocked for 20 min with 200 μl blocking buffer (1% (w/v) BSA and 0.2% (w/v) gelatin in 1× PBS). The slides were then incubated with 50–80 μl diluted primary antibodies (in blocking buffer) covered with a coverslip in a humidity chamber for either 4 h at room temperature or overnight at 4 °C. The slides were then washed three times in 1× PBS for 5 min and incubated with diluted secondary antibodies covered with a coverslip in a humidity chamber for 2–4 h at room temperature. Finally, the slides were washed three times in 1× PBS for 5 min and mounted with ProLong Diamond Antifade Mountant (Invitrogen) containing DAPI to visualize DNA.

The following primary antibodies were used: rabbit anti-Zip1 (1:500)[77], guinea pig anti-Rec8 (1:1000)[78], rabbit anti-Zip3 (1:1,000)[79], rabbit anti-Msh5 (1:500)[79], mouse anti-Myc (1:300, 9E10, Cancer Research UK), guinea pig anti-Ecm11-Gmc2 (1:800)[80], mouse anti-Smt3 (1:500, 4F2.F5.G2, Rockland Immunochemicals), guinea pig anti-Hop1 (1:500)[81] and mouse anti-γ-tubulin/Tub4 (1:200, MPI-CBG A81)[62]. Secondary antibodies raised in goat or donkey and conjugated to Alexa Fluor 488, Alexa Fluor 555 and Alexa Fluor 647 (1:500, Invitrogen) were used for detection. For STED, chromosome spreads were prepared as described above and immunostained using goat anti-rabbit STAR ORANGE and goat anti-guinea pig STAR RED secondary antibodies

(1:100, Abberior) for detection and slides mounted with ProLong Glass Antifade Mountant (Invitrogen). Widefield fluorescence microscopy images were acquired on a DeltaVision Ultra epifluorescence microscope (GE Healthcare) with a 100× oil-immersion UPlanSApo objective (1.4 NA, working distance 0.13 mm) and a sCMOS camera controlled by AcquireUltra software (v.1.2.3) running on Linux. STED images were acquired using an Abberior STEDYCON mounted on a Zeiss Axio Imager A2 and a 100× oil-immersion alpha Plan-Apochromat objective (1.46 NA, working distance 0.11 mm) and deconvolved using Huygens Professional (SVI) STED deconvolution software. Images were processed and analysed using Fiji[82] and ImageJ macros to aid intensity analysis and focus counting. Unless otherwise noted, single $z$ slice images are shown modified for display using linear brightness and contrast adjustments.

The following criteria were used to assess chromosome synapsis based on Zip1, Ecm11–Gmc2 and Smt3 immunostaining. Using the chromosome axes (Rec8) and chromatin (DAPI) as references, three categories of nuclei were defined: 'full synapsis' includes nuclei in which ≥75% of chromosomes show complete, continuous synapsis; 'partial synapsis' includes nuclei with short stretches of polymerized Zip1, with ≤75% of the chromosomes synapsed; 'no synapsis' includes nuclei with no or few foci of Zip1 staining. Nuclei with extrachromosomal Zip1, Ecm11–Gmc2 or Smt3 polycomplexes were also scored, and polycomplexes were quantified separately by monitoring for absence of chromatin association.

### Live-cell imaging
All live imaging experiments used conditioned sporulation medium (filter-sterilized SPM from the respective cultures). In brief, 8-well Lab-Tek II chambered coverglasses (Nunc, 155409) were prepared by coating the bottom of each well with 2 mg ml$^{-1}$ concanavalin A (dissolved in 1× PBS containing 50 mM $CaCl_2$ and 50 mM $MnCl_2$) and incubating at 30 °C for 10 min. Excess concanavalin A was removed by aspiration and each well was washed twice with conditioned SPM. For imaging, cells were collected from the meiotic cultures at the desired timepoints after meiotic induction in SPM and diluted with conditioned SPM to $OD_{600}$ = 1.8. Subsequently, 0.1 ml aliquots were added to concanavalin-A-coated wells and the cells were allowed to settle at 30 °C for 3 min. The supernatant was removed and the wells were carefully washed once with conditioned SPM before 200 μl of conditioned SPM was added to each well. Cells were imaged using a DeltaVision Ultra epifluorescence microscope (GE Healthcare) with an environmental chamber heated to 30 °C. For experiments involving the addition of β-oestradiol, 5-Ph-IAA or rapamycin, the required chemicals were diluted to twice the final concentration in 200 μl of conditioned SPO. The diluted mixture was then carefully added directly to the respective wells.

Images were acquired using a 60× oil-immersion UPlanXApo objective (1.42 NA, working distance 0.15 mm) and a sCMOS camera controlled by AcquireUltra software (v.1.2.3) running on Linux. $z$-stack images (8 sections, 1 μm apart) were acquired every 3, 5 or 10 min for 12–15 h. Images were deconvolved using Huygens Professional (SVI) widefield deconvolution software and maximum intensity $z$-projected over the range of acquisition in Fiji. Maximum-intensity $z$-projections are shown in time-lapse montages and videos, modified for presentation using linear brightness and contrast adjustments. Images in videos are additionally smoothed using mean filtering in Fiji and aligned using the HyperStackReg ImageJ plugin[83]. Montages in videos were generated using the Multi Stack Montage ImageJ plugin (BIOP, EPFL).

### In situ immunofluorescence analysis
The efficiency of Rec104[FRB] nuclear depletion using the anchor-away technique was evaluated by in situ immunostaining according to a previously described protocol[84], with some modifications. In brief, 1 ml of meiotic culture was fixed with 3.7% formaldehyde overnight. Cells were washed three times with 0.1 M KPi buffer pH 6.4, once with

spheroplasting buffer (0.1 M KPi pH 7.4, 1.2 M sorbitol, 0.5 mM $MgCl_2$) and resuspended in 200 μl spheroplasting buffer. Cells were incubated with 10 mM DTT for 15 min at 30 °C and then digested with 10 μl of Zymolyase 100T solution (1 mg ml$^{-1}$), the efficiency of which was monitored by cell lysis in the presence of 2% (w/v) sarcosyl. Spheroplasts were washed once with spheroplasting buffer, loaded onto poly-L-lysine-coated microscope slides and fixed in ice-cold methanol for 3 min and in ice-cold acetone for 10 s. Cells were blocked in 1% (w/v) BSA in 1× PBS and then stained with a primary mouse anti-HA.11 antibody (1:200, 16B12, BioLegend) and a secondary Alexa Fluor 488-conjugated anti-mouse antibody (1:300, Invitrogen). Slides were mounted with ProLong Diamond Antifade Mountant (Invitrogen) containing DAPI to visualize DNA. Images were acquired on the Zeiss Axio Imager M2 equipped with a 63× oil-immersion Plan-Apochromat objective (1.4 NA, working distance 0.19 mm) and a CoolSNAP HQ2 camera under the control of Zeiss ZEN blue 3.3, and image analysis was performed using Fiji. Approximately 100 cells were analysed per experiment.

### Protein analysis by western blotting
Protein extracts were prepared as previously described[62]. In brief, 10 ml of meiotic culture was collected by centrifugation (3 min at 800 rcf) and the cells were opened in 10% trichloroacetic acid using glass beads and a FastPrep-24 5G instrument (MP Biomedicals) running three cycles of 40 s (6 m s$^{-1}$). Protein precipitates were collected by centrifugation at 4 °C (10 min at 800 rcf), resuspended in 2× NuPAGE sample buffer (Invitrogen) supplemented with 200 mM DTT and neutralized with 1 M Tris base at a 2:1 (v/v) ratio. The samples were boiled at 95 °C for 5 min and cleared by centrifugation (10 min at 21,300 rcf). The relative protein concentration was measured using the Bio-Rad protein assay. Protein samples were separated on NuPAGE 3–8% Tris-Acetate gels or 4–12% Bis-Tris gels (Invitrogen) using NuPAGE Tris-Acetate or MES SDS running buffer (Invitrogen) and transferred onto Amersham Hybond 0.45 μm PVDF membranes (Sigma-Aldrich).

The following primary antibodies were used for immunoblotting: rabbit anti-Myc conjugated to HRP (1:15,000, ab1326, Abcam), rabbit anti-Zip1 (1:5,000)[77], rabbit anti-Crm1 (1:5,000, a gift from K. Weis), mouse anti-Myc (1:5,000, 9E10, Cancer Research UK), mouse anti-GFP (1:2,000, 7.1/13.1, Roche), rabbit anti-Ecm11 (1:5,000, a gift from A. Pichler), rabbit anti-Smt3 (1:5,000, a gift from A. Pichler), mouse anti-Pgk1 (1:10,000, 22C5D8, Invitrogen), guinea pig anti-Hop1 (1:5,000)[81], rabbit anti-Hop1-pT318 (1:5,000)[81] and mouse anti-HA.11 (1:2,500, 16B12, BioLegend).

The following secondary antibodies were used in Extended Data Figs. 6c and 7b,d: goat anti-mouse immunoglobulins conjugated to HRP (1:10,000, P0447, Agilent) and swine anti-rabbit immunoglobulins conjugated to HRP (1:10,000, P0399, Agilent). For all other experiments, the following fluorescent secondary antibodies were used for detection and quantification of protein levels: goat anti-rabbit conjugated to IRDye 800CW (1:15,000, 926-32211, LI-COR Biosciences), goat anti-mouse IgG conjugated to Alexa Fluor 680 (1:15,000, A21057, Invitrogen) and goat anti-guinea pig IgG conjugated to Alexa Fluor 647 (1:15,000, A21450, Invitrogen). After washing with PBS-T, the blots were imaged using the ChemiDoc MP Imaging System (Bio-Rad) and image analysis and quantification were performed in Fiji. Images were processed using Fiji and Adobe Photoshop, with only linear adjustments to brightness and contrast applied for presentation purposes.

### Physical analysis of recombination at *HIS4::LEU2* by Southern blotting
Genomic DNA preparation and physical analysis of recombination intermediates at the *HIS4::LEU2* recombination hotspot by Southern blotting were performed as previously described[2,13,33,85,86]. In brief, 50–100 ml of cells was collected from the cultures and resuspended in 0.1 mg ml$^{-1}$ trioxsalen. DNA was crosslinked at 3,600 mJ cm$^{-2}$ using

a UVP Crosslinker CL-3000L (Analytik Jena), with cells kept on ice and mixed at regular intervals. Genomic DNA was extracted using guanidine/sarcosyl to lyse the cells, followed by phenol–chloroform extraction. After genomic DNA preparation, approximately 1.5 mg of DNA was digested with XhoI (DNA joint molecule analysis) or XhoI/NgoMIV (crossover/non-crossover analysis) and separated by one-dimensional gel electrophoresis on 0.6% agarose gels in 1× TBE buffer (90 mM Tris base, 90 mM boric acid, 2 mM EDTA pH 8.0) at 2 V cm⁻¹ for 21 h. Physical analysis of branched recombination intermediates using native/native two-dimensional gels was performed as previously described[87,88] with XhoI-digested genomic DNA loaded onto a 0.4% agarose gel (Seakem Gold, Lonza) in 1× TBE without ethidium bromide and run at 1 V cm⁻¹ for 21 h at room temperature. Gels were stained in 1× TBE containing ethidium bromide and portions of lanes containing DNA species of interest were excised and placed in a gel tray at 90° to the direction of electrophoresis. 0.8% agarose (Ultrapure, Invitrogen) in 1× TBE containing ethidium bromide was poured around the gel slices and allowed to solidify. Two-dimensional gel electrophoresis was performed at 4 °C in pre-chilled 1× TBE containing ethidium bromide at 5.3 V cm⁻¹ for 4 h. After one or two-dimensional gel electrophoresis, the DNA was transferred to a GeneScreen Plus membrane (Revvity) by alkaline transfer. The membranes were hybridized with a probe (Probe A)[89] for the *HIS4::LEU2* recombination hotspot, which was random-primed labelled with [α−³²P]dCTP using High Prime labelling mixture (11585592001, Roche). Hybridized membranes were exposed to a phosphor screen and imaged using the Amersham Typhoon phosphor imager (Cytiva). Southern blot images were adjusted for presentation using Fiji, applying only linear modifications to brightness and contrast. Signal intensities of different recombination intermediates were quantified relative to the total lane signal using ImageQuant software or Fiji. The signal at 0 h in SPM was used for background subtraction.

## Physical analysis of DSBs by Southern blotting

Genomic DNA was isolated from cells embedded in low-melting-temperature agarose plugs to prevent non-specific shearing of genomic DNA as described previously[90], except using 1% SDS instead of 1% sarcosyl. The DNA embedded in the plugs was digested as previously described[91], with adaptations. In brief, for each timepoint, one-third of an agarose plug was equilibrated four times in 5 ml of TE (10 mM Tris-HCl pH 7.5, 1 mM EDTA pH 7.5) for 15 min on a rotating wheel at room temperature. Agarose plugs were melted at 65 °C for 10 min, equilibrated at 42 °C for at least 20 min and digested with 0.5 U β-agarase I (New England Biolabs) at 42 °C for 45 min, followed by heat inactivation of β-agarase I at 65 °C for 15 min. After digestion with β-agarase I, the DNA was digested with 20 U of the respective restriction enzyme and an appropriate dilution of restriction enzyme buffer (rCutsmart, New England Biolabs) at 37 °C for 2 h. After 2 h, an additional 20 U of fresh restriction enzyme and half of the previously added volume of restriction enzyme buffer was added and the DNA was further digested at 37 °C for 2 h. Approximately 600 ng of digested DNA was loaded on a 0.8% agarose gel in 1× TBE and run at 2 V cm⁻¹ for 14 h, using 30 ng of BstEII-digested lambda DNA (New England Biolabs) as a molecular mass standard. After DNA transfer to a GeneScreen Plus membrane (Revvity) by alkaline transfer, the membranes were hybridized with an [α−³²P]dCTP-radiolabelled probe for the *ERG1* or *CCT6* hotspot and lambda DNA-BstEII digest (0.5 ng), exposed to a phosphor screen and imaged using the Amersham Typhoon phosphor imager (Cytiva). DSB signal intensities were quantified relative to the total lane signal using ImageQuant software. The signal at 0 h in SPM was used for background subtraction. The restriction enzymes used for digestion and the primer sequences used for probe amplification were as described previously[91]: *ERG1*, NcoI, (5′-CTGCCTACTCAAAACAGCAAAG, 5′-GTGAAGGAAGCACGTCAGAAAAAGC); *CCT6*, PstI, (5′-GCGTCCC GCAAGGACATTAG, 5′-TTGTGGCTAATGGTTTTGCGGTG).

## Quantification and statistical analysis

All quantifications and data analyses were performed using Excel (Microsoft) and Prism (v.9.5.1 for macOS; GraphPad) software. The number of analysed cells (*n*) and the statistical analyses are described in each figure legend. For multiple comparisons, the non-parametric Kruskal–Wallis test was performed in Prism, followed by a post hoc Dunn's test for multiple-comparison correction. For pairwise comparisons, two-tailed unpaired Mann–Whitney *U*-tests were used.

## Ethics and inclusion statement

This study complies with the ethics and inclusion guidelines of *Nature*.

## Reporting summary

Further information on research design is available in the Nature Portfolio Reporting Summary linked to this article.

## Data availability

Relevant data supporting the findings of this study are provided within the Article and its Supplementary Information. Source data for all images, Southern blots and western blots are available at Zenodo[92] (https://doi.org/10.5281/zenodo.15862742). Biological materials used in this study are available from the corresponding author on reasonable request.

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

**Acknowledgements** We thank A. Shinohara, A. MacQueen, A. Pichler, K. Weis, M. Shinohara, F. Stutz and F. Klein for reagents; M. Arter for initial work characterizing Yen1[ON]; N. Hunter for communicating unpublished data; V. Borde and M. Jagannathan for advice; V. Jantsch, M. Blanco and M. Xaver for reading the manuscript; and the staff at the Max Perutz Labs BioOptics facility for providing support for microscopy experiments. The Matos Lab was supported by the Swiss National Science Foundation SNSF (176108), the Austrian Science Foundation FWF (SFB Meiosis, 8807-B; 10.55776/F88) and the European Research Council ERC (101002629).

**Author contributions** A.H. performed all experiments except for the Southern blots performed in Figs. 1–3 and Extended Data Figs. 3, 5h and 6, which were performed by L.O., and the live-cell imaging experiments in Fig. 3 and Extended Data Figs. 1 and 7, which were performed by D.V.; J.M. conceived and supervised the project. A.H. and J.M. wrote the manuscript with input from all of the authors.

**Funding** Open access funding provided by University of Vienna.

**Competing interests** The authors declare no competing interests.

**Additional information**
**Correspondence and requests for materials** should be addressed to Joao Matos.

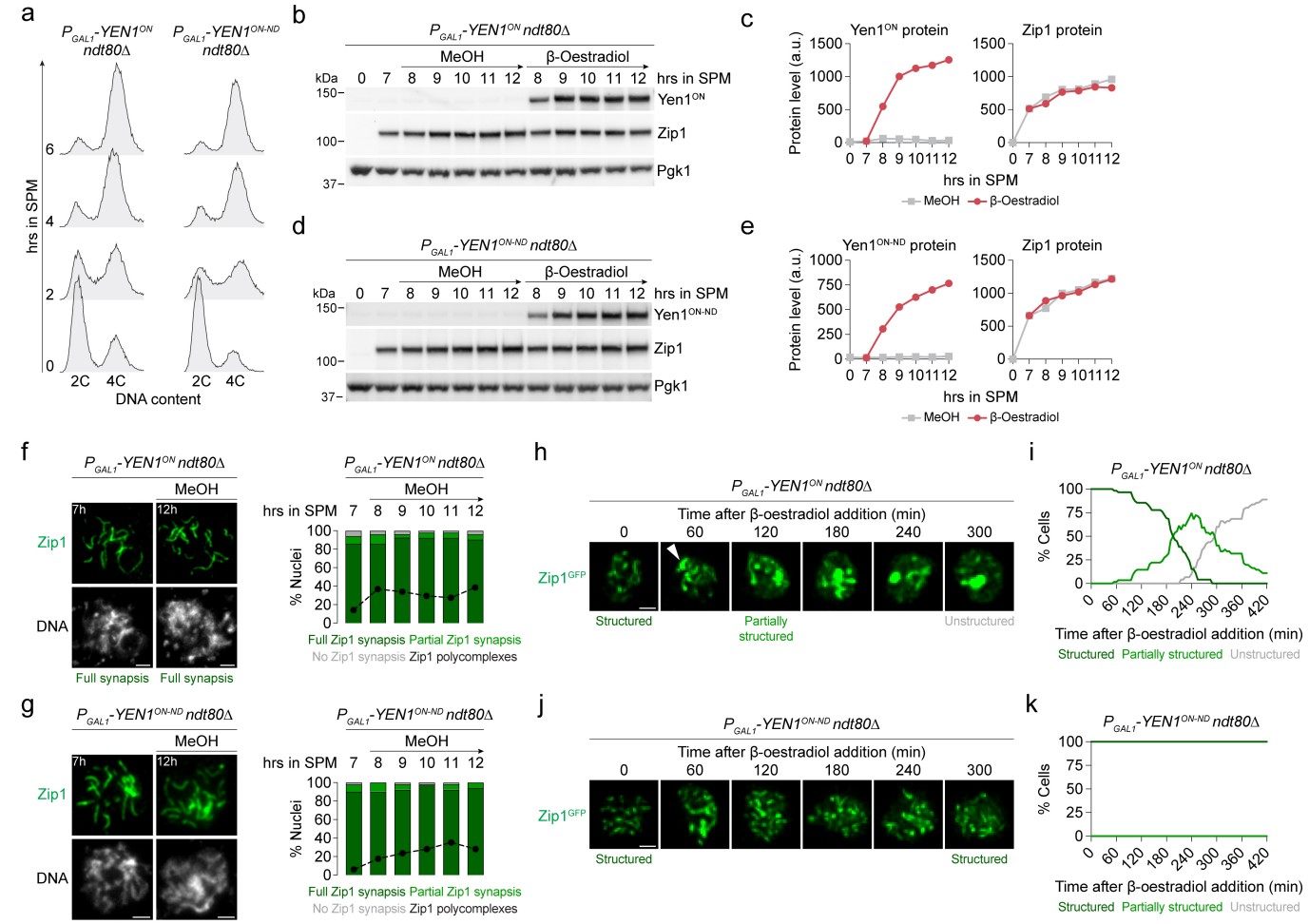

**Extended Data Fig. 1 | HJs stabilize the SC during meiotic pachytene.**
**a**, Representative fluorescence-activated cell sorting (FACS) analysis of DNA content at regular time intervals after induction of meiosis in SPM to monitor pre-meiotic DNA replication in cells with the indicated genotypes. Note that most cells have replicated their genomes after ~4 h in SPM. Corresponds to experiments in Fig. 1b–g and Extended Data Fig. 1b–g. **b, c**, Western blot analysis (b) and quantification of Yen1^ON and Zip1 protein levels normalized using Pgk1 as loading control (c) in cells at indicated times in SPM. Yen1^ON was induced by β-oestradiol addition (or MeOH as control) at 7 h. Corresponds to the experiment in Fig. 1b,c and Extended Data Fig. 1f. Yen1^ON protein levels after addition of β-oestradiol are also shown in Fig. 1d. **d, e**, As in (b) and (c), for $P_{GAL1}$-Yen1^ON-ND. Corresponds to the experiment in Fig. 1e,f and Extended Data Fig. 1g. Yen1^ON-ND protein levels after addition of β-oestradiol are also shown in Fig. 1g. **f**, Representative images of meiotic chromosome spreads at indicated times in SPM (left) and quantification of Zip1 synapsis and polycomplex formation (right) in the absence of Yen1^ON induction (MeOH-treated control) at 7 h (*n* = 50 nuclei per time point; representative of three biological replicates). Control for the experiment shown in Fig. 1b–d. **g**, As in (f), for $P_{GAL1}$-Yen1^ON-ND. Control for the experiment shown in Fig. 1e–g. **h, i**, Time-lapse image montage of Zip1^GFP in a cell nucleus (h) and quantification of structured Zip1^GFP signal (i) after Yen1^ON induction by β-oestradiol addition at ~7 h in SPM (t = 0 min). Arrowhead marks the emerging polycomplex. Mean of two biological replicates is plotted (*n* = 40 cells each). Corresponds to Supplementary Video 1. Scale bar = 1 µm. **j, k**, As in (h) and (i), for $P_{GAL1}$-Yen1^ON-ND. Mean of two biological replicates is plotted (*n* = 40 cells each). Corresponds to Supplementary Video 2.

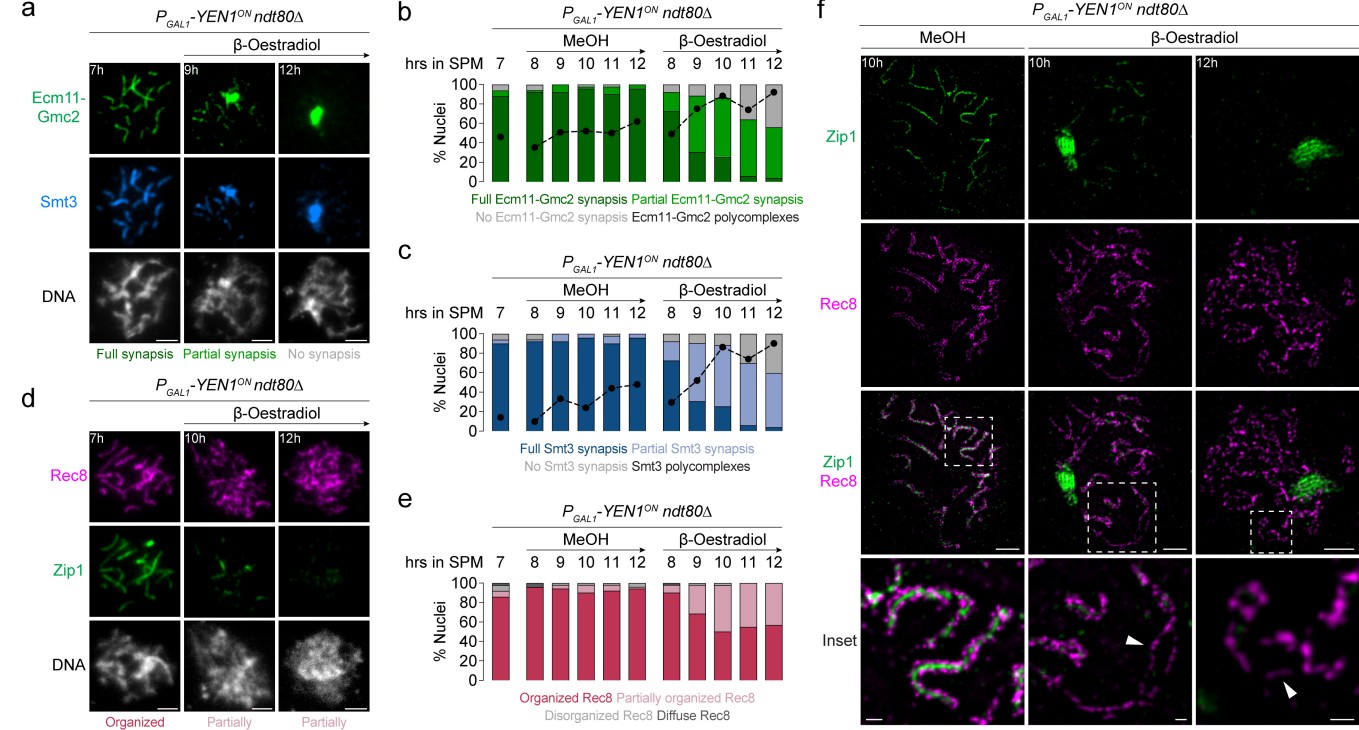

**Extended Data Fig. 2 | HJs stabilize the SC central region but are dispensable for the axis-loop organization of pachytene chromosomes. a**, Representative images of meiotic chromosome spreads at indicated times in SPM, immunostained for Ecm11-Gmc2 (green) and Smt3 (blue). Yen1[ON] expression was induced by β-oestradiol addition (or MeOH as control) at 7 h in SPM. Scale bars = 2 μm. **b, c**, Quantification of Ecm11-Gmc2 synapsis (b) and SC-associated Smt3 (c) from (a) (*n* = 50 nuclei per time point; representative of two biological replicates). **d**, Representative images of meiotic chromosome spreads at indicated times in SPM, immunostained for Zip1 (green) and Rec8 (magenta). Yen1[ON] expression was induced by β-oestradiol addition (or MeOH as control) at 7 h in SPM. Scale bars = 2 μm. **e**, Quantification of chromosome axis morphology based on Rec8 immunostaining from (d) (*n* = 50 nuclei per time point; representative of two biological replicates). **f**, Representative STED microscopy images of meiotic chromosome spreads at indicated times in SPM, immunostained for Zip1 (green) and Rec8 (magenta). Yen1[ON] expression was induced by β-oestradiol addition (or MeOH as control) at 7 h in SPM. Insets show magnified regions indicated by dashed lines. Arrowheads indicate example chromosome regions with split axes. Scale bars = 1 μm; scale bars in insets = 200 nm.

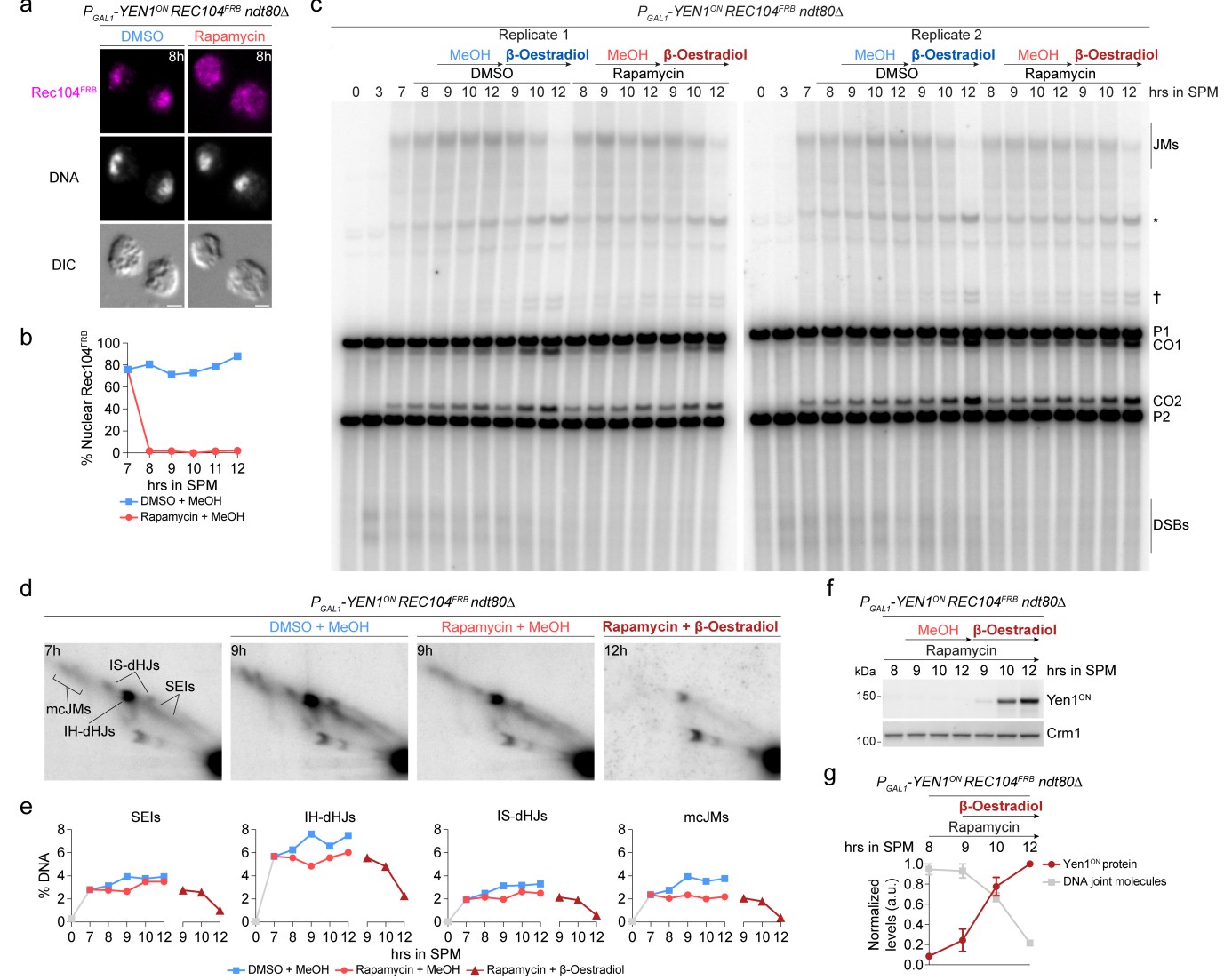

**Extended Data Fig. 3 | Recombination intermediates containing dHJs and SEIs are stable during pachytene. a**, Representative in situ immunofluorescence images of cells at indicated times in SPM after Rec104[FRB] anchor-away by rapamycin addition (or DMSO as control) at 7 h, immunostained for the HA epitope to visualize the subcellular localization of Rec104[FRB]. DNA is visualized with DAPI (grey). **b**, Quantification of cells with nuclear Rec104[FRB] at indicated times in SPM from (a) (*n* = 100 cells per time point). **c**, Southern blot analysis of recombination at the *HIS4::LEU2* recombination hotspot from the experiment described in Fig. 1h. Two biological replicates are shown; replicate 1 corresponds to the cropped blot in Fig. 1i. P1, parental 1; P2, parental 2; CO1, crossover 1; CO2, crossover 2; DSBs, double-strand breaks; JMs, DNA joint molecules. Asterisk indicates meiosis-specific recombinant band from gene conversion of the XhoI site closest to the DSB site. Dagger indicates ectopic recombination bands

between *HIS4::LEU2* and *leu2::hisG* at the native *LEU2* locus. **d**, Native/native two-dimensional Southern blot analysis of branched JMs at the *HIS4::LEU2* recombination hotspot from indicated time points of replicate 1 in (c). The different DNA JM species are depicted in the left blot image. SEI, single-end intermediate; IH-dHJ, interhomologue-dHJ; IS-dHJ, intersister-dHJ; mcJM, multichromatid DNA JM. **e**, Quantification of SEI, IH-dHJ, IS-dHJ and mcJM levels from (d), shown as the percentage of total DNA. **f**, Western blot analysis of Yen1[ON] protein levels in cells from indicated times in SPM of replicate 1 in (c). Crm1 served as protein loading control. Representative of two biological replicates. **g**, Quantification of Yen1[ON] protein levels from the Western blot in (f) and a biological replicate, normalized to the highest value (red line), along with mean JM levels from both replicates in (c) (grey line). Error bars represent the range.

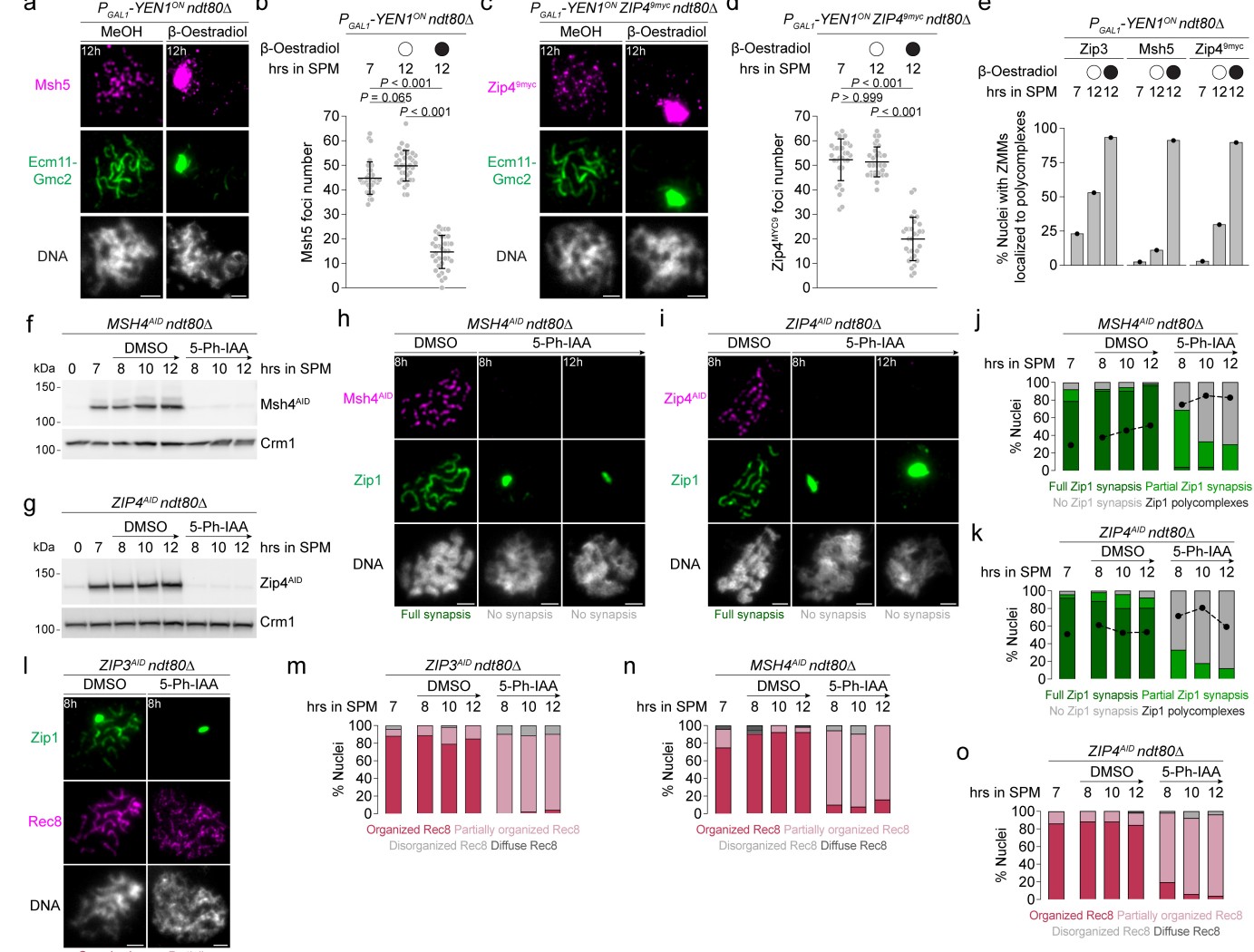

**Extended Data Fig. 4 | ZMMs are continuously required to maintain chromosome synapsis during prophase I. a**, Representative images of meiotic chromosome spreads at indicated times in SPM following Yen1^ON induction by β-oestradiol addition (or MeOH as control) at 7 h, immunostained for Ecm11-Gmc2 (green) and Msh5 (magenta). Scale bars = 2 μm. **b**, Quantification of Msh5 foci number from (a) (mean ± s.d.; *n* = 30 nuclei per time point; Kruskal-Wallis test (p ≤ 0.001) with Dunn's multiple comparison test). Representative of two biological replicates. **c, d**, As in (a) and (b), for analysis of Zip4^9myc (mean ± s.d.; *n* = 30 nuclei per time point; Kruskal-Wallis test (p ≤ 0.001) with Dunn's multiple comparison test). Representative of two biological replicates. **e**, Quantification of Zip3, Msh5 and Zip4^9myc aggregates associating with Ecm11-Gmc2 polycomplexes at indicated times in SPM. Yen1^ON was induced by β-oestradiol addition (or MeOH as control) at 7 h (*n* = 30 nuclei per time point; representative of two biological replicates). **f, g**, Western blot analysis of Msh4^AID (f) and Zip4^AID (g) protein levels in cells at indicated times in SPM. *Os*Tir1^F74G expression

was induced by addition of CuSO₄ at 6.5 h, and AID-dependent depletion was induced by 5-Ph-IAA addition (or DMSO as a control) at 7 h, as described in Fig. 2c. Crm1 served as protein loading control. Representative of two biological replicates. **h**, Representative images of meiotic chromosome spreads at indicated times in SPM from (f), immunostained for Zip1 (green) and Msh4^AID (magenta). Scale bars = 2 μm. **i**, As in (h), for *ZIP4^AID* in (g). **j, k**, Quantification of Zip1 synapsis and polycomplexes from (h) and (i) (*n* = 50 nuclei per time point; representative of two biological replicates). **l**, Representative images of meiotic chromosome spreads at indicated times in SPM, immunostained for Zip1 (green) and Rec8 (magenta). Zip3^AID depletion was induced by 5-Ph-IAA addition (or DMSO as control) at 7 h. Scale bars = 2 μm. **m-o**, Quantification of chromosome axis morphology based on Rec8 immunostaining in cells with indicated genotypes at specified times in SPM. Zip3^AID, Msh4^AID or Zip4^AID depletion was induced by 5-Ph-IAA addition (or DMSO as control) at 7 h (*n* = 50 nuclei per time point; representative of two biological replicates).

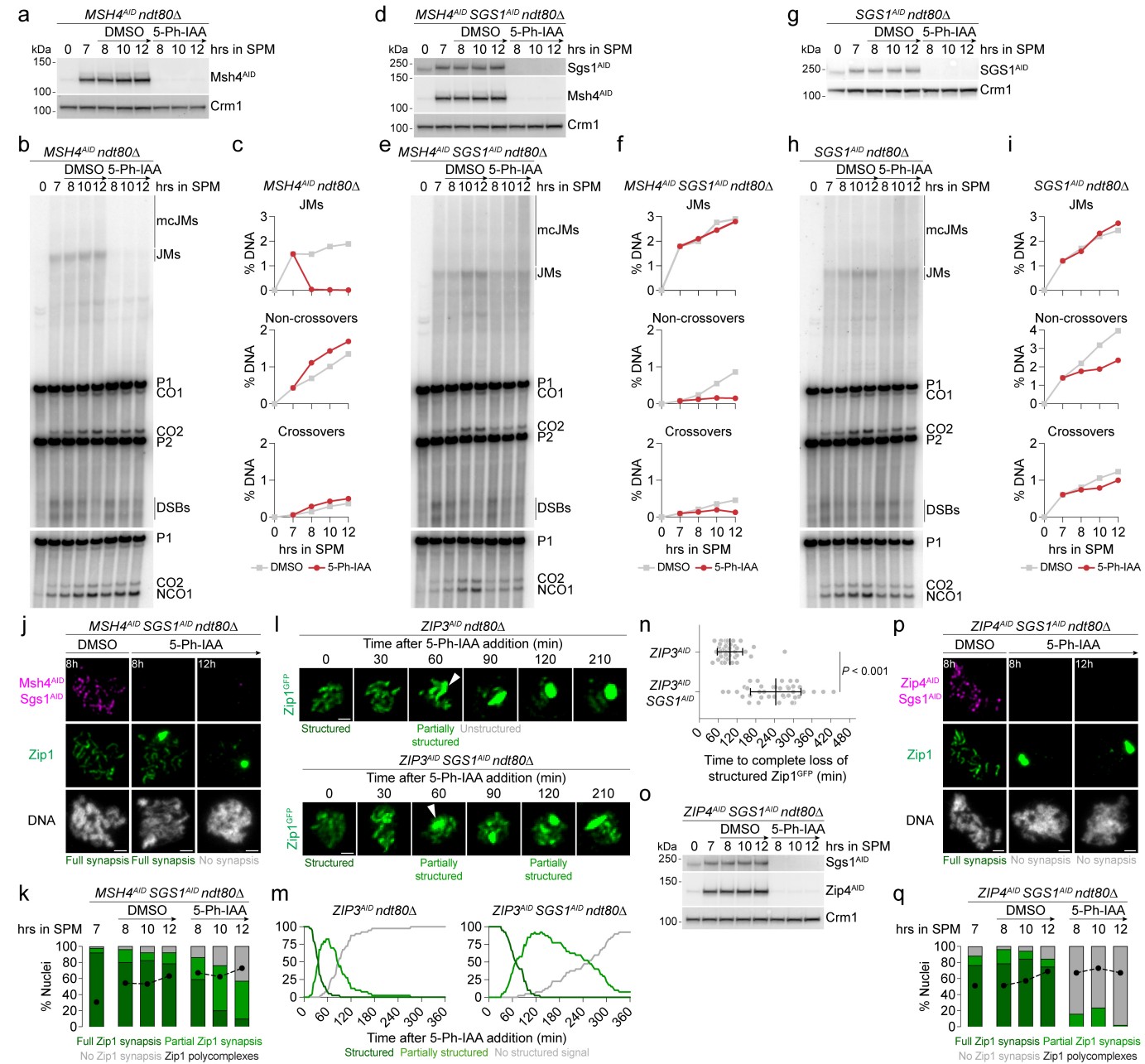

**Extended Data Fig. 5 | ZMMs protect dHJs from Sgs1-mediated dissolution throughout prophase I. a**, Western blot analysis of Msh4$^{AID}$ protein levels in cells at indicated times in SPM. *Os*Tir1$^{F74G}$ expression was induced by addition of CuSO$_4$ at 6.5 h in SPM, and Msh4$^{AID}$ depletion was induced by 5-Ph-IAA addition (or DMSO as a control) at 7 h, as described in Fig. 2c. Crm1 served as protein loading control. Representative of two biological replicates. **b, c**, Southern blot (b) and quantification of JMs, non-crossover (NCO1) and crossover (CO2) products (c) at the *HIS4::LEU2* recombination hotspot from (a). **d**, As in (a), for *MSH4$^{AID}$ SGS1$^{AID}$*. Representative of two biological replicates. **e,f**, As in (b) and (c), for *MSH4$^{AID}$ SGS1$^{AID}$* from (d). **g**, As in (a), for *SGS1$^{AID}$*. **h, i**, As in (b) and (c), for *SGS1$^{AID}$* from (g). **j**, Representative images of meiotic chromosome spreads at indicated times in SPM from (d), immunostained for Zip1 (green) and Msh4$^{AID}$/Sgs1$^{AID}$ (magenta). Scale bars = 2 μm. **k**, Quantification of Zip1 synapsis and polycomplexes

from (j) (*n* = 50 nuclei per time point; representative of two biological replicates). **l**, Time-lapse image montage of Zip1$^{GFP}$ in cell nuclei of indicated genotypes. *Os*Tir1$^{F74G}$ expression was induced by addition of CuSO$_4$ at 6.5 h in SPM, and Zip3$^{AID}$ and Sgs1$^{AID}$ depletion was induced by 5-Ph-IAA addition (or DMSO as a control) at -7 h in SPM (t = 0 min). Arrowhead marks the emerging polycomplex. Corresponds to Supplementary Video 3 and 4. Scale bar = 1 μm. **m**, Quantification of structured Zip1$^{GFP}$ signal in cells of indicated genotypes from (l) (*n* = 40 cells per genotype). **n**, Quantification of time to complete loss of structured Zip1$^{GFP}$ signal in cells of indicated genotypes from (m) (mean ± s.d; *n* = 40 cells per genotype; two-tailed, unpaired Mann-Whitney U test). **o**, As in (a), for *ZIP4$^{AID}$ SGS1$^{AID}$*. Representative of two biological replicates. **p**, As in (j), for (o). **q**, As in (k), for (p) (*n* = 50 nuclei per time point; representative of two biological replicates).

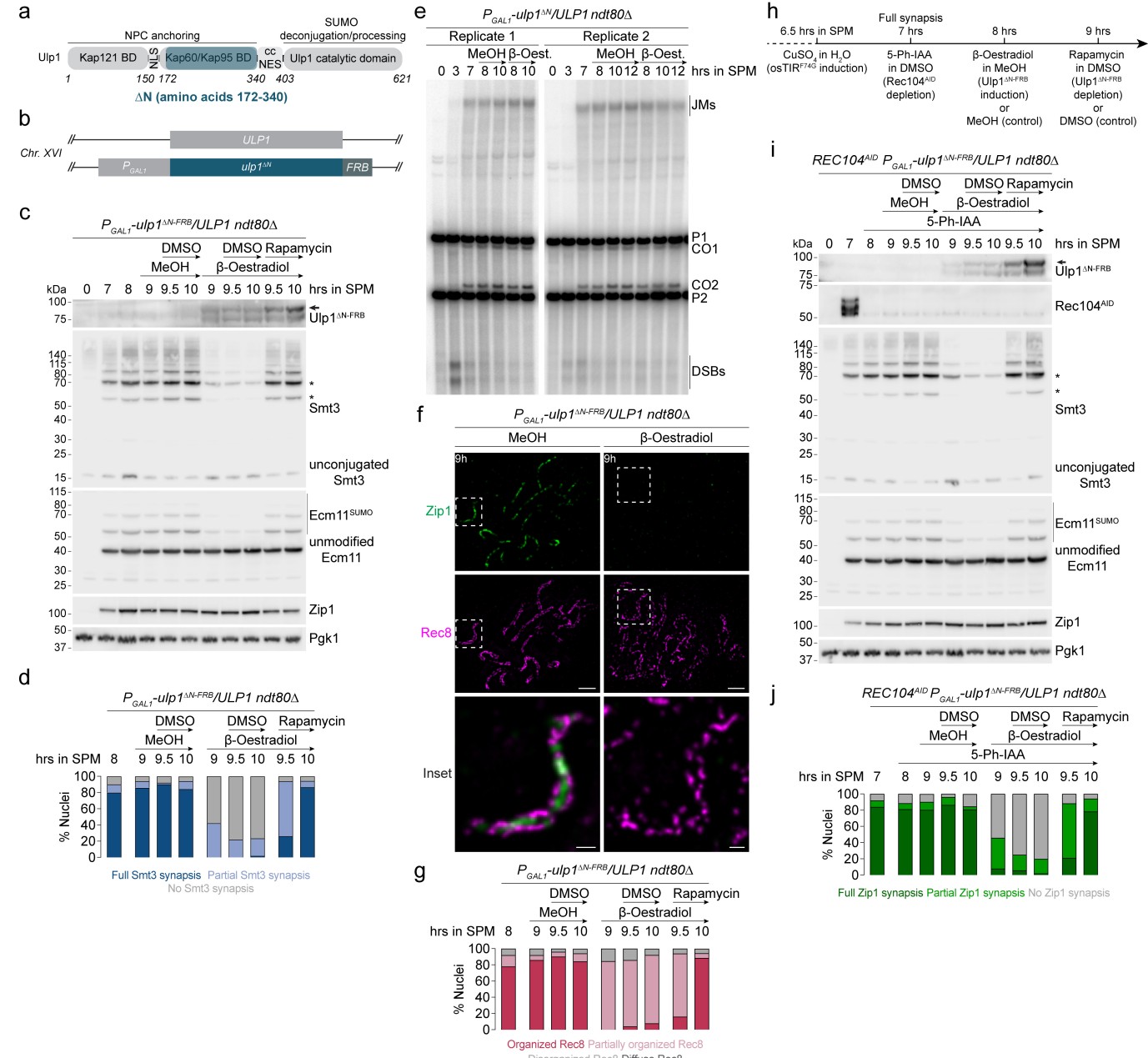

**Extended Data Fig. 6 | A molecular tool to reversibly induce SC disassembly independent of HJ resolution. a**, Ulp1 domain organization, adapted from[48]. In the *ulp1ΔN* mutant, an in-frame *GFP* coding sequence replaces parts of the nuclear pore complex anchoring region (amino acids 172–340) in the N-terminal domain. **b**, Schematic representation of the *ULP1* genomic locus. A diploid yeast strain heterozygous for the *ulp1ΔN-FRB* allele was used since *ULP1* is an essential gene. The native promoter of *ulp1ΔN-FRB* is replaced by $P_{GAL1}$, which can be induced by the addition of β-oestradiol. In addition, the C-terminal FRB domain of *ulp1ΔN-FRB* enables the conditional nuclear depletion using the rapamycin-dependent anchor-away system. **c**, Western blot analysis of Ulp1ΔN-FRB, Smt3, Ecm11 and Zip1 protein levels in cells at indicated times in SPM. Ulp1ΔN-FRB expression was induced by β-oestradiol addition (or MeOH as control) at 8 h, and nuclear depletion was induced by rapamycin addition (or DMSO as control) at 9 h. Arrow indicates the band corresponding to the expected molecular weight of Ulp1ΔN-FRB. Asterisks indicate putative SUMOylated and polySUMOylated Ecm11 bands in the anti-Smt3 blot. Samples were run on separate gels due to similar molecular masses of the proteins, with Pgk1 as a sample processing control. Representative of two biological replicates. **d**, Quantification of SC-associated Smt3 on meiotic chromosome spreads from (c) ($n = 50$ nuclei per time point; representative of two biological replicates). **e**, Southern blot analysis of JMs and DSBs at the *HIS4::LEU2* recombination hotspot. Ulp1ΔN was induced

by β-oestradiol addition (or MeOH as control) at 7 h in SPM. Two biological replicates are shown, with replicate 2 including an additional time point (12 h in SPM). Replicate 1 corresponds to the cropped blot in Fig. 3d. **f**, Representative STED microscopy images of meiotic chromosome spreads at indicated times in SPM from (c), immunostained for Zip1 (green) and Rec8 (magenta). Insets show magnified regions indicated by dashed lines. Note that although Zip1 is lost from chromosomes and Rec8 localization appears to be more discontinuous after Ulp1ΔN expression (see also (g)), the co-alignment of homologous axes is maintained. Scale bars = 1 μm; scale bars in insets = 200 nm. **g**, Quantification of chromosome axis morphology based on Rec8 immunostaining on meiotic chromosome spreads at indicated times in SPM from (c) ($n = 50$ nuclei per time point; representative of two biological replicates). **h**, Experimental setup to determine whether SC reassembly upon Ulp1ΔN-FRB expression/nuclear depletion depends on the formation and repair of de novo DSBs. **i**, Western blot analysis of Ulp1ΔN-FRB, Rec104AID, Smt3, Ecm11 and Zip1 protein levels in cells at indicated times in SPM, treated as described in (h). Arrow indicates the band corresponding to the expected molecular weight of Ulp1ΔN-FRB. Asterisks indicate putative SUMOylated and polySUMOylated Ecm11 bands in the anti-Smt3 blot. Samples were run on separate gels due to similar molecular masses of the proteins, with Pgk1 as a sample processing control. **j**, Quantification of Zip1 synapsis on meiotic chromosome spreads from (i) ($n = 50$ nuclei per time point).

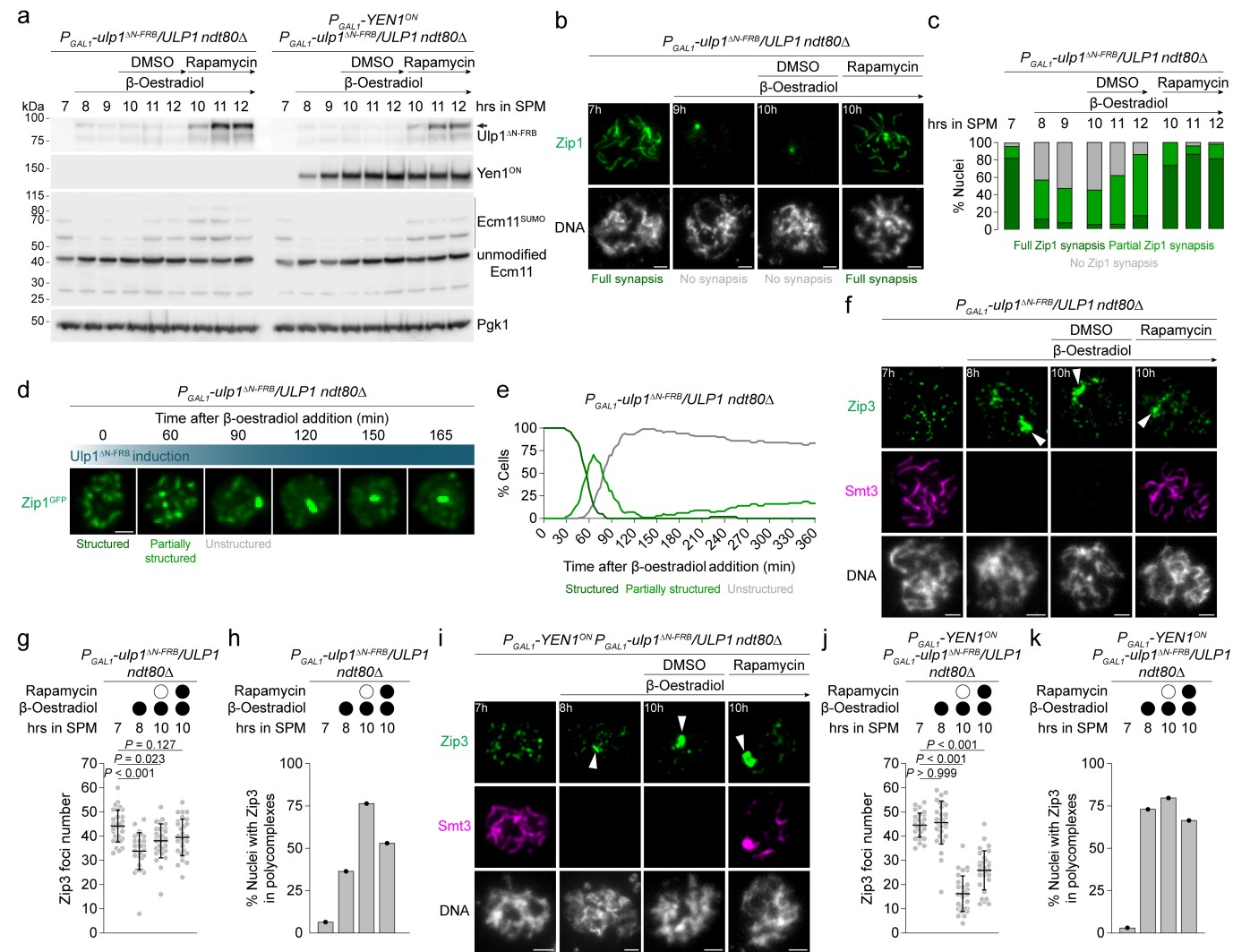

**Extended Data Fig. 7 | dHJ-ZMM protein interplay enables SC reassembly upon Ulp1^ΔN-FRB depletion. a**, Western blot analysis of Ulp1^ΔN, Yen1^ON and Ecm11 protein levels in cells with the indicated genotype at specified times in SPM, treated as described in Fig. 3f. Arrow indicates the band corresponding to the expected molecular weight of Ulp1^ΔN-FRB. Samples were run on separate gels due to similar molecular masses of the proteins, with Pgk1 as a sample processing control. **b**, Representative images of meiotic chromosome spreads of the indicated genotype from (a) at specified times in SPM, immunostained for Zip1 (green). Control for the experiment shown in Fig. 3g. Scale bars = 2 μm. **c**, Quantification of Zip1 synapsis from (b) (n = 50 nuclei per time point). **d**, Time-lapse image montage of Zip1^GFP in a cell nucleus after Ulp1^ΔN-FRB expression induced by β-oestradiol addition at 7 h in SPM (t = 0 min). Control for the experiment shown in Fig. 3i, with DMSO added after 120 min. Corresponds to

Supplementary Video 5. Scale bar = 1 μm. **e**, Quantification of structured Zip1^GFP signal from (d). Mean of two biological replicates is plotted (n = 40 cells each). **f**, Representative images of meiotic chromosome spreads at indicated times in SPM, immunostained for Zip3 (green) and Smt3 (magenta). Ulp1^ΔN-FRB expression was induced by β-oestradiol addition at 7 h in SPM, and nuclear depletion was initiated by rapamycin addition at 9 h in SPM. Arrowheads mark Zip3 aggregates. Scale bars = 2 μm. **g, h**, Quantification of Zip3 foci number (g) and fraction of nuclei exhibiting Zip3 aggregates (h) from (f) (mean ± s.d.; n = 30 nuclei per time point; Kruskal-Wallis test (p ≤ 0.001) with Dunn's multiple comparison test). **i**, As in (f), for the simultaneous expression of Ulp1^ΔN-FRB and Yen1^ON by β-oestradiol addition at 7 h in SPM. **j, k**, As in (g) and (h), for (i) (mean ± s.d.; n = 30 nuclei per time point; Kruskal-Wallis test (p ≤ 0.001) with Dunn's multiple comparison test).

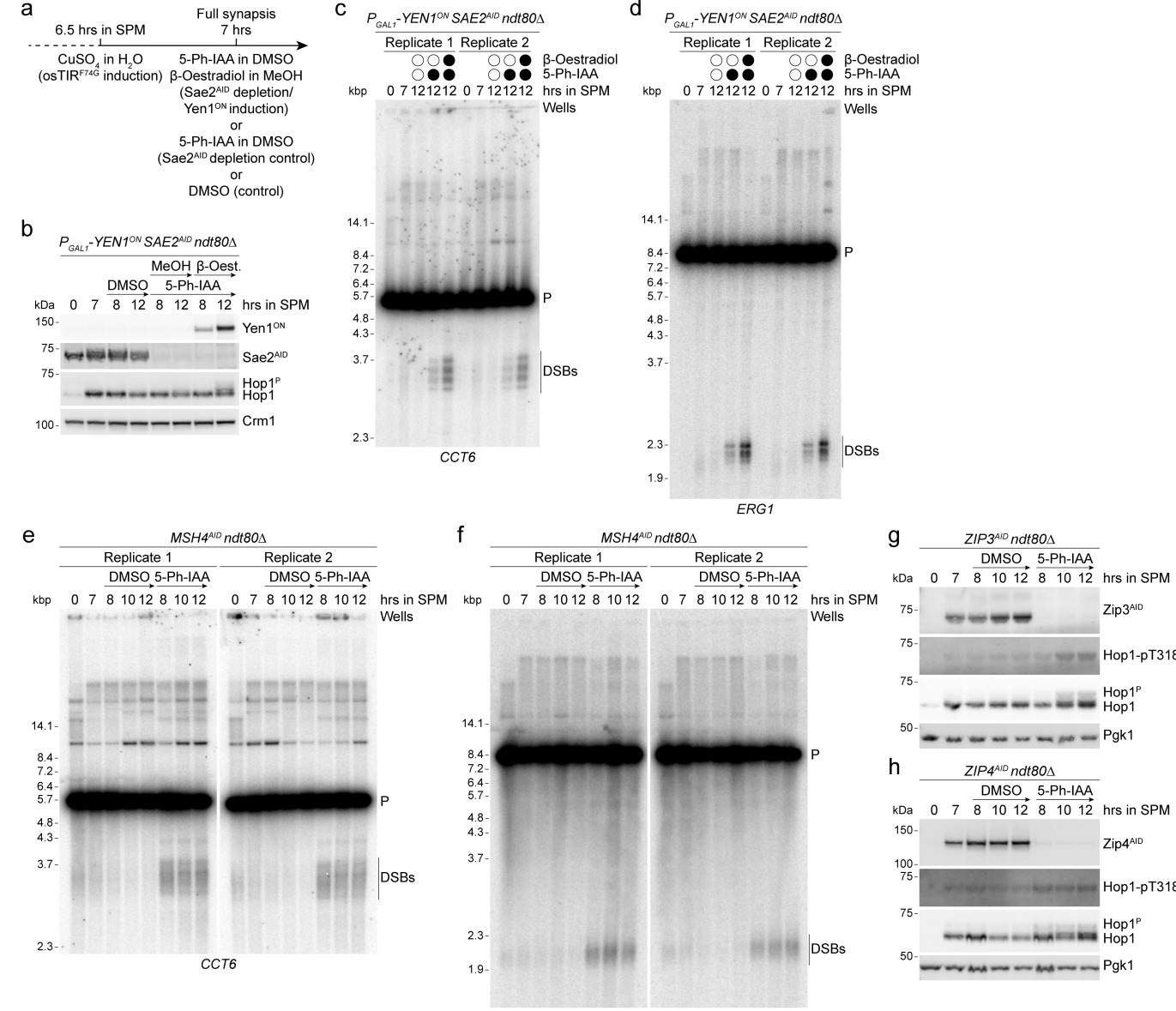

**Extended Data Fig. 8 | dHJ-ZMM protein interplay suppresses DSB formation.**
**a**, Experimental setup to measure DSB formation by Southern blotting after
Yen1$^{ON}$ expression in *ndt80Δ* cells. Sae2$^{AID}$ is depleted by 5-Ph-IAA addition
(or DMSO as control) alongside Yen1$^{ON}$ induction by β-oestradiol addition (or
MeOH as control) to prevent endonucleolytic removal of Spo11 and subsequent
DSB processing and repair. **b**, Western blot analysis of Yen1$^{ON}$, Sae2$^{AID}$ and Hop1
protein levels in cells at indicated times in SPM, treated as described in (a).
Hop1$^P$ indicates phosphorylated Hop1. Crm1 served as protein loading control.
Representative of two biological replicates. **c, d**, Southern blot analysis of DSB
formation at the *CCT6* (c) and *ERG1* (d) hotspots from (b). Two biological

replicates are shown side-by-side in the same blot for each hotspot; replicate 2
of the *CCT6* blot corresponds to the cropped blot in Fig. 4d. **e, f**, Southern blot
analysis of DSB formation at the *CCT6* (e) and *ERG1* (f) hotspots in cells with the
indicated genotype, with Msh4$^{AID}$ depletion by 5-Ph-IAA addition (or DMSO as
control) at 7 h in SPM. Two biological replicates are shown per locus; replicate 1
of the *CCT6* blot in (e) corresponds to the cropped blot in Fig. 4g. **g**, Western
blot analysis of Zip3$^{AID}$, Hop1 and Hop1-pT318 protein levels in cells at indicated
times in SPM. Zip3$^{AID}$ depletion was induced by 5-Ph-IAA addition (or DMSO as
control) at 7 h in SPM. Hop1$^P$ indicates phosphorylated Hop1. Pgk1 served as
protein loading control. **h**, As in (g), for *ZIP4$^{AID}$*.

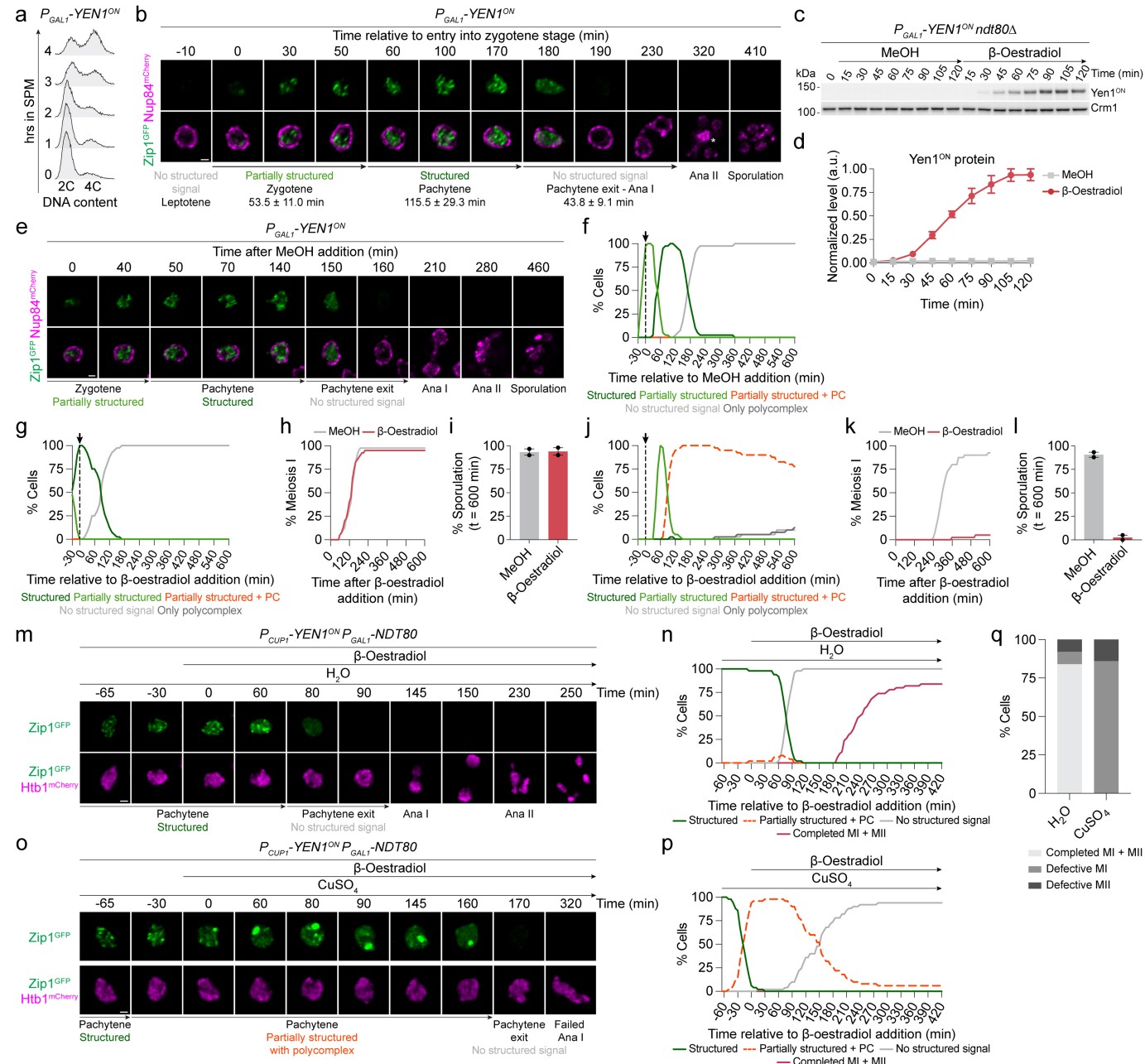

**Extended Data Fig. 9 | Premature resolution of recombination intermediates disrupts the SC and causes meiosis I failure. a**, Representative FACS analysis of DNA content in cells with the indicated genotype at regular time intervals after induction of meiosis in SPM. Corresponds to the experiment in Fig. 4i–l and Extended Data Fig. 9b–l. **b**, Time-lapse image montage of Zip1^GFP in a cell nucleus induced to enter meiosis in SPM, with imaging starting at 4.5 h in SPM (t = −30 min). The durations of the zygotene and pachytene stages of prophase I, and the time from pachytene exit to anaphase I are indicated (mean ± s.d of two biological replicates; n = 20 cells each). The brightness of Nup84^mCherry was adjusted to facilitate visibility of the signal. Asterisk indicates the gametogenesis uninherited nuclear compartment (GUNC). Ana, Anaphase. Corresponds to Supplementary Video 8. Scale bar = 1 μm. **c, d**, Western blot analysis (c) and quantification (d) of Yen1^ON protein levels, normalized to the highest value, in cells at indicated times after Yen1^ON induction by β-oestradiol addition (or MeOH as control) at 7 h in SPM. Error bars represent range of two biological replicates. **e**, Time-lapse image montage of Zip1^GFP in a cell nucleus after MeOH addition as a control in early to mid-zygotene (t = 0 min, -5 h in SPM). The brightness of Nup84^mCherry was adjusted to facilitate visibility of the signal. Corresponds to Supplementary Video 10. Scale bar = 1 μm. **f**, Quantification of structured

Zip1^GFP signal from (e). Mean of two biological replicates is plotted (n = 20 cells each). **g-i**, Quantification of structured Zip1^GFP signal (g), meiosis I entry (h), and sporulation (i) in pachytene cells at the time of β-oestradiol addition (t = 0 min, -5 h in SPM). Mean of two biological replicates is plotted (n = 20 cells each; error bars represent range). Corresponds to Supplementary Video 11. **j-l**, Quantification of structured Zip1^GFP signal (j), meiosis I entry (k), and sporulation (l) in pre-leptotene/leptotene cells at the time of β-oestradiol addition (t = 0 min, -5 h in SPM). Mean of two biological replicates is plotted (n = 20 cells each; error bars represent range). Corresponds to Supplementary Video 12. **m**, Time-lapse image montage of Zip1^GFP and Htb1^mCherry in a cell nucleus of the indicated genotype. H₂O was added as a control at 7 h in SPM (t = −65 min). Ndt80 expression was induced by β-oestradiol addition at ~8 h in SPM (t = 0 min). Corresponds to Supplementary Video 13. Scale bar = 1 μm. **n**, Quantification of structured Zip1^GFP signal and the fraction of cells undergoing meiosis I and II based on Htb1^mCherry labelled chromatin from (m) (n = 60 cells). **o**, As in (m), with CuSO₄ addition to induce expression of Yen1^ON at 7 h in SPM (t = −65 min). Corresponds to Supplementary Video 14. **p**, As in (n), for (o). **q**, Quantification of cells with completed or defective meiotic divisions from (n) and (p).

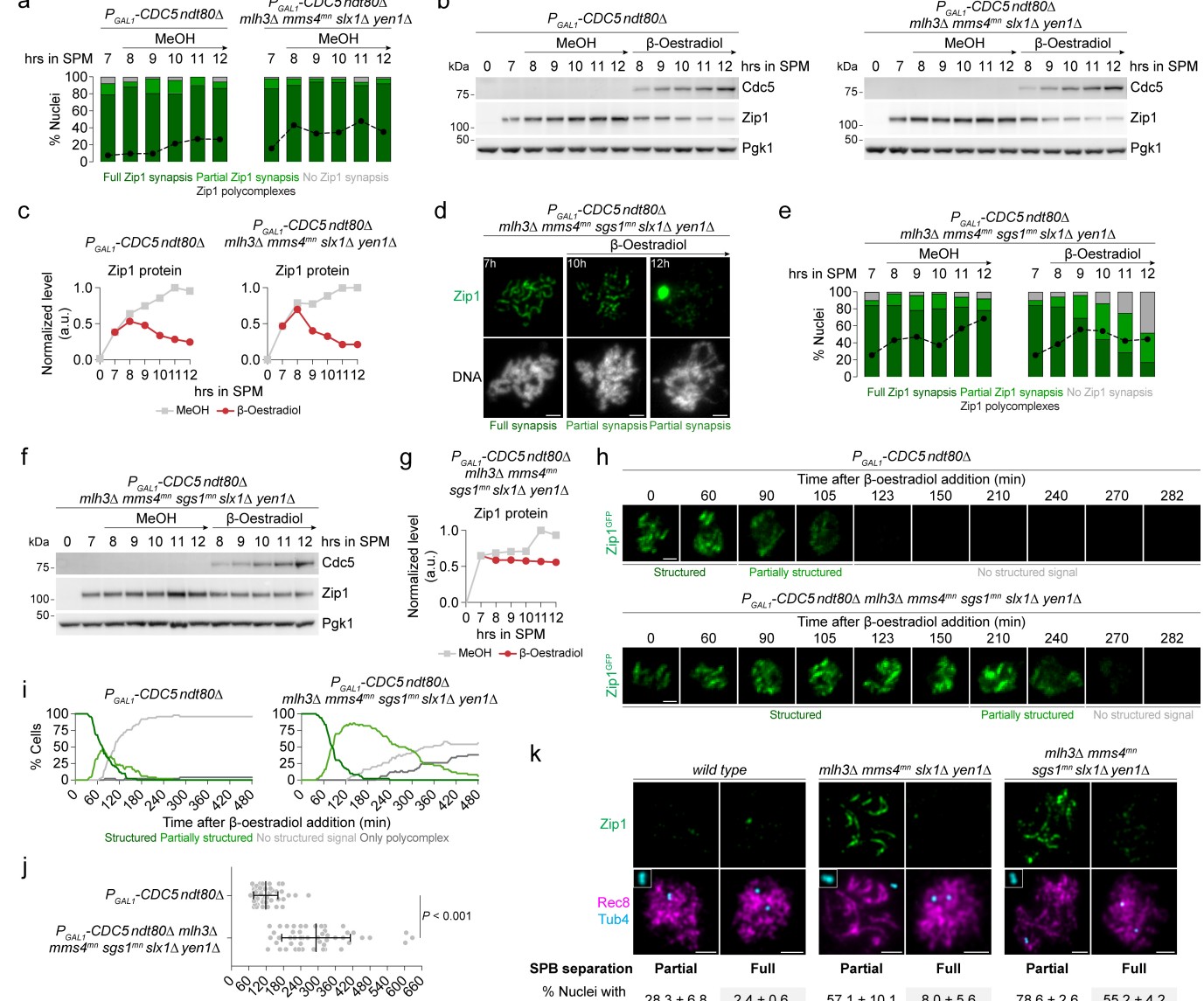

**Extended Data Fig. 10 | Cdc5 promotes SC disassembly in part through the activation of HJ resolvases. a**, Quantification of Zip1 synapsis and polycomplexes in the absence of Cdc5 induction (MeOH-treated control) at 7 h in SPM (*n* = 50 nuclei per time point). Controls for the experiments shown in Fig. 5a,b. **b**, Western blot analysis of Cdc5 and Zip1 protein levels in cells of indicated genotypes at specified times in SPM. Cdc5 was induced by β-oestradiol addition (or MeOH as control) at 7 h in SPM. Pgk1 served as protein loading control. Correspond to experiments in (a) and Fig. 5a,b. **c**, Quantification of Zip1 protein levels from (b), normalized to the highest value. **d**, Representative images of meiotic chromosome spreads at indicated times in SPM, immunostained for Zip1 (green). Cdc5 was induced by β-oestradiol addition (or MeOH as control) at 7 h in SPM. Scale bars = 2 μm. **e**, Quantification of Zip1 synapsis and polycomplexes from (d) (*n* = 50 nuclei per time point). **f**, As in (b), for cells of the indicated genotype. Corresponds to experiment in (d)

and (e). **g**, As in (c), for (f). **h, i**, Time-lapse image montage of Zip1^GFP in cell nuclei of indicated genotypes (h) and quantification of structured Zip1^GFP signal (i) after Cdc5 induction by β-oestradiol addition at ‑7 h in SPM (t = 0 min) (*n* = 50 cells per genotype). The bottom panel corresponds to Supplementary Video 17. Scale bars = 1 μm. **j**, Quantification of time to complete loss of structured Zip1^GFP signal in cells of indicated genotypes from (i) (mean ± s.d; *n* = 50 cells per genotype; two-tailed, unpaired Mann-Whitney U test). **k**, (Top) Representative images of meiotic chromosome spreads of the indicated genotypes at 6 h in SPM, immunostained for Zip1 (green), Rec8 (magenta) and the SPB component γ-tubulin/Tub4 (cyan). Insets show magnified, slightly separated SPBs. Scale bars = 2 μm. (Bottom) Quantification of nuclei with partial or full SPB separation containing Zip1 stretches (weighted mean ± weighted SD of two biological replicates; n = 152–174 nuclei per condition).

# Reporting Summary

## Statistics

For all statistical analyses, confirm that the following items are present in the figure legend, table legend, main text, or Methods section.

| n/a | Confirmed | |
|---|---|---|
| ☐ | ☒ | The exact sample size (*n*) for each experimental group/condition, given as a discrete number and unit of measurement |
| ☐ | ☒ | A statement on whether measurements were taken from distinct samples or whether the same sample was measured repeatedly |
| ☐ | ☒ | The statistical test(s) used AND whether they are one- or two-sided <br> *Only common tests should be described solely by name; describe more complex techniques in the Methods section.* |
| ☒ | ☐ | A description of all covariates tested |
| ☐ | ☒ | A description of any assumptions or corrections, such as tests of normality and adjustment for multiple comparisons |
| ☐ | ☒ | A full description of the statistical parameters including central tendency (e.g. means) or other basic estimates (e.g. regression coefficient) AND variation (e.g. standard deviation) or associated estimates of uncertainty (e.g. confidence intervals) |
| ☐ | ☒ | For null hypothesis testing, the test statistic (e.g. *F*, *t*, *r*) with confidence intervals, effect sizes, degrees of freedom and *P* value noted <br> *Give P values as exact values whenever suitable.* |
| ☒ | ☐ | For Bayesian analysis, information on the choice of priors and Markov chain Monte Carlo settings |
| ☒ | ☐ | For hierarchical and complex designs, identification of the appropriate level for tests and full reporting of outcomes |
| ☒ | ☐ | Estimates of effect sizes (e.g. Cohen's *d*, Pearson's *r*), indicating how they were calculated |

*Our web collection on statistics for biologists contains articles on many of the points above.*

## Software and code

Policy information about availability of computer code

| Data collection | Deltavision Ultra (GE Healthcare): AcquireUltra (version 1.2.3) <br> Abberior STEDYCON: STEDYCON smart control (version 7.1.53); <br> BD FACSCalibur: BD CellQuest Pro (4.0.2); <br> Zeiss Axio Imager A2: Zeiss ZEN Blue (3.3) <br> ChemiDoc MP Imaging System (Bio-Rad): Image Lab Software (2.4.0.03); <br> Amersham Typhoon phosphor imager (Cytiva): Amersham Typhoon control software (3.0.0.2) |
|---|---|
| Data analysis | Cytological and live-cell image analyses were performed using Fiji (version 2.14.0/1.54f). Live-cell and STED images were deconvolved with Huygens Professional (SVI, version 25.04) and analyzed using Fiji. Western blots were quantified in Fiji and prepared for presentation in Fiji and Adobe Photoshop (version 25.12.0). FACS data were analyzed using FlowJo (version 10.9.0). Southern blots were quantified with ImageQuant TL (version 8.1) or Fiji and adapted for presentation in Fiji. Graphs were generated, and all statistical analyses were performed in GraphPad Prism (version 9.5.1) or Microsoft Excel for Mac (version 16.87). Figures and schemes were assembled in Adobe Illustrator (version 25.0, 2021). |

For manuscripts utilizing custom algorithms or software that are central to the research but not yet described in published literature, software must be made available to editors and reviewers. We strongly encourage code deposition in a community repository (e.g. GitHub). See the Nature Portfolio guidelines for submitting code & software for further information.

## Data

Policy information about availability of data

All manuscripts must include a data availability statement. This statement should provide the following information, where applicable:

- Accession codes, unique identifiers, or web links for publicly available datasets
- A description of any restrictions on data availability
- For clinical datasets or third party data, please ensure that the statement adheres to our policy

> Relevant data supporting the findings of this study are provided within the article and its supplementary Information. Source data for all images, Southern blots, and Western blots are available at DOI: 10.5281/zenodo.15862742. Biological materials used in this study are available from the corresponding author upon reasonable request.

## Research involving human participants, their data, or biological material

Policy information about studies with human participants or human data. See also policy information about sex, gender (identity/presentation), and sexual orientation and race, ethnicity and racism.

| | |
|---|---|
| Reporting on sex and gender | The study did not involve human participants, their data, or biological material. |
| Reporting on race, ethnicity, or other socially relevant groupings | The study did not involve human participants, their data, or biological material. |
| Population characteristics | The study did not involve human participants, their data, or biological material. |
| Recruitment | The study did not involve human participants, their data, or biological material. |
| Ethics oversight | The study did not involve human participants, their data, or biological material. |

Note that full information on the approval of the study protocol must also be provided in the manuscript.

# Field-specific reporting

Please select the one below that is the best fit for your research. If you are not sure, read the appropriate sections before making your selection.

☒ Life sciences ☐ Behavioural & social sciences ☐ Ecological, evolutionary & environmental sciences

For a reference copy of the document with all sections, see nature.com/documents/nr-reporting-summary-flat.pdf

# Life sciences study design

All studies must disclose on these points even when the disclosure is negative.

| | |
|---|---|
| Sample size | All sample sizes and the number of biological replicates for each experiment are reported in the figure legends. No statistical methods were used to predetermine sample size. For cytological and live-cell imaging analyses, sample sizes were guided by preliminary experiments that helped establish an appropriate sample size for the purpose of this study. For Southern and Western blot assays, the number of experiments performed followed standard practices in the field. |
| Data exclusions | No data were excluded from the analyses. |
| Replication | Experimental findings were confirmed by performing multiple biological replicates as indicated in the figure legends. |
| Randomization | Not applicable. Experiments involved comparisons between control (wild type or other appropriate controls) and mutant or treated yeast cells; randomization was therefore neither necessary nor appropriate. |
| Blinding | Blinding was not used during data collection and analysis. All results involved side-by-side comparisons of mutant or treated yeast cells with appropriate controls. For cytology experiments, blinding was not feasible because treated yeast cells typically exhibit distinct and specific features, such as disassembly of the synaptonemal complex. |

# Reporting for specific materials, systems and methods

We require information from authors about some types of materials, experimental systems and methods used in many studies. Here, indicate whether each material, system or method listed is relevant to your study. If you are not sure if a list item applies to your research, read the appropriate section before selecting a response.

## Materials & experimental systems

| n/a | Involved in the study |
|-----|----------------------|
| ☐ | ☒ Antibodies |
| ☐ | ☒ Eukaryotic cell lines |
| ☒ | ☐ Palaeontology and archaeology |
| ☒ | ☐ Animals and other organisms |
| ☒ | ☐ Clinical data |
| ☒ | ☐ Dual use research of concern |
| ☒ | ☐ Plants |

## Methods

| n/a | Involved in the study |
|-----|----------------------|
| ☒ | ☐ ChIP-seq |
| ☒ | ☐ Flow cytometry |
| ☒ | ☐ MRI-based neuroimaging |

# Antibodies

| | |
|---|---|
| Antibodies used | Primary antibodies for cytological analysis included: rabbit anti-Zip1 (Grigaitis et al., 2020), guinea pig anti-Rec8 (Bommi et al., 2019), rabbit anti-Zip3 (Shinohara et al., 2008), rabbit anti-Msh5 (Shinohara et al., 2008), mouse anti-Myc (9E10, Cancer Research UK), guinea pig anti-Ecm11-Gmc2 (Voelkel-Meiman et al., 2019), mouse anti-Smt3/SUMO (4F2.F5.G2, Rockland Immunochemicals), guinea pig anti-Hop1 (Iwasaki et al., 2016), mouse anti-γ-tubulin/Tub4 (MPI-CBG A81; Matos et al., 2008), and mouse anti-HA.11 (16B12, BioLegend). |
| | Secondary antibodies for cytology were goat or donkey antibodies conjugated to Alexa Fluor 488, Alexa Fluor 555, and Alexa Fluor 647 (Invitrogen). For STED microscopy, secondary antibodies included goat anti-rabbit STAR ORANGE (Abberior) and goat anti-guinea pig STAR RED (Abberior). |
| | Primary antibodies for Western blotting were: rabbit anti-Myc conjugated to HRP (ab1326, Abcam), rabbit anti-Zip1 (Grigaitis et al., 2020), rabbit anti-Crm1 (gift from K. Weis), mouse anti-Myc (9E10, Cancer Research UK; 1:5000), mouse anti-GFP (clones 7.1/13.1, Roche; 1:2000), rabbit anti-Ecm11 (gift from A.Pichler), rabbit anti-Smt3/SUMO (gift from A. Pichler), mouse anti-Pgk1 (22C5D8, Invitrogen), guinea pig anti-Hop1 (Iwasaki et al., 2016), rabbit anti-Hop1-pT318 (Iwasaki et al., 2016), and mouse anti-HA.11 (16B12, BioLegend). |
| | Secondary antibodies for Western blotting included goat anti-mouse IgG conjugated to HRP (P0447, Agilent), swine anti-rabbit IgG conjugated to HRP (P0399, Agilent), goat anti-rabbit conjugated to IRDye 800CW (926-32211, LI-COR Biosciences), goat anti-mouse IgG conjugated to Alexa Fluor 680 (A21057, Invitrogen), and goat anti-guinea pig IgG conjugated to Alexa Fluor 647 (A21450, Invitrogen). |
| Validation | All home-made antibodies were characterized and validated with appropriate controls for both Western blotting and cytological analyses in previous publications, as cited above. For cytological and Western blot detection of tagged proteins using anti-Myc, anti-HA, or anti-GFP antibodies, untagged controls were included to confirm specificity. The specificity of anti-Smt3/SUMO (4F2.F5.G2, Rockland Immunochemicals) for immunofluorescence was confirmed by our deSUMOylation experiment (Figure 3 and Extended Data Figure 6). The specificity of anti-Crm1 (gift from K. Weis) and anti-Pgk1 (22C5D8, Invitrogen) was validated by detection of bands at the appropriate molecular weights in Western blotting. The specificity of anti-Ecm11 and anti-Smt3/SUMO (gifts from A. Pichler) was confirmed using an ecm11Δ knockout strain and deSUMOylation experiment (Figure 3 and Extended Data Figure 6), respectively. |

# Eukaryotic cell lines

Policy information about cell lines and Sex and Gender in Research

| | |
|---|---|
| Cell line source(s) | We used budding yeast strains, all derivatives of SK1. Detailed genotypes are provided in Supplementary Table 1. |
| Authentication | n/a |
| Mycoplasma contamination | n/a |
| Commonly misidentified lines (See ICLAC register) | n/a |

# Plants

| | |
|---|---|
| Seed stocks | The study did not involve plant seeds. |
| Novel plant genotypes | The study did not involve plants. |
| Authentication | The study did not involve plants. |

