## [Peer Review File · Nature]

Holliday junction-ZMM protein feedback enables meiotic crossover assurance

Corresponding Author: Professor Joao Matos

This file contains all reviewer reports in order by version, followed by all author rebuttals in order by version. Parts of this Peer Review File have been redacted as indicated to maintain the confidentiality of unpublished data.

Version 0:

Reviewer comments:

Referee #1

(Remarks to the Author)

Reciprocal exchange between homologous DNA duplexes (crossovers) are linked - physically and functionally - to a supramolecular structure called synaptonemal complex (SC). Both crossovers and SC are hallmark features of meiosis. Many aspects of the relationship between DNA recombination intermediates and synaptonemal complex remain unclear. Using conditional induction of a constitutive form of the Yen1 resolvase in meiotic prophase-arrested cells (cells containing unresolved DNA joint molecule recombination intermediates), the authors demonstrate that ectopic resolution of joint molecule meiotic recombination intermediates can destabilize ZMM protein ensembles, which play a critical role in the formation of those very same DNA joint molecule intermediates. The authors also show that ZMM ensembles play a continuous role in stabilizing/protecting the joint molecule from inappropriate resolution, which is consistent with indirect data published previously. Thus, the authors here have established that at late meiotic prophase, ZMMs preserve DNA joint molecules AND that DNA joint molecules preserve the ZMM ensemble.

In addition, the authors show that ectopic resolution of DNA joint molecules during mid-late prophase of meiosis can destabilize SC structures (central region components disappear but SC axis structures remain intact), which rely on ZMM protein function for their formation.

Another major finding of this study, which was obtained by inducing a version of Ulp1 that localizes to the nucleus instead of nuclear pores, is that the removal of SC itself does not trigger joint molecule resolution, and SCs can in fact re-establish after their removal so long as joint molecules have not resolved.

Also, I found interesting the data suggesting ZZS can partially stabilize joint molecules and SC structures when Zip3 and Msh4 are removed, so long as the Sgs1 helicase is also missing.

The experiments were performed in a rigorous manner and the collected data support the authors' interpretations, but with one important exception noted below.

The findings here are impactful. It is not necessarily surprising that ZMMs play a continued role in preserving DNA joint molecules after their initial establishment and that DNA joint molecules play a continued role in preserving associated ZMM ensembles (as with any complex, the components each play a role in the final architecture; removing one of the components, DNA or protein, will change that architecture and possibly destabilize the entire ensemble). It is also not completely surprising that SC structures rely on the integrity of ZMM ensembles, particularly since a major structural component of the SC in budding yeast is also a key component of ZMM ensembles (Zip1). However, to my knowledge this is the first study to directly demonstrate the existence of a reciprocal dependency between the DNA joint molecule and the protein structures that house it, during late prophase.

The study's major findings are also a significant contribution to the field because of how they speak to what might happen at the end of prophase in normal meiosis. While joint molecule resolution and SC disassembly normally occur nearly simultaneously at the end of meiotic prophase, it has been unclear whether SC disassembly triggers resolution of JMs or whether resolution of JMs might occur first to trigger SC disassembly. The authors have gone on to specifically address this question in more detail, by examining if ANY SC disassembly can occur at the end of meiotic prophase (given the proper upstream signals) in a mutant that lacks all four major resolvase enzymes. The authors found that SC does ultimately

disassemble in this situation, but with a delay. These data are the first (to my knowledge) to indicate that there is likely a combination of mechanisms that promote the disassembly of SC during normal meiosis, and one of these contributing mechanisms is DNA joint molecule resolution itself.

However, as noted above there is one claim made that is not warranted by the study's experimental findings. That SC maintenance promotes crossover assurance is not a rigorous conclusion based on the experimental data reported, and is not a claim supported by existing mutant data.

There are some areas of the narrative where greater clarity or more information should be provided to enable a full understanding of the new findings and how to situate them with prior work in this area.

Major comments

1. The experiments do not provide rigorous evidence that DNA joint molecule-dependent maintenance of chromosome synapsis functions to "enable crossover assurance", as is stated in the title. In addition, contrary evidence to this claim exists: *ecm11* or *gmc2* null mutants (missing fundamental structural components of the SC) have high spore viability, indicating that even when SC (synapsis) is absent, meiosis proceeds (albeit with a small delay at late prophase) and crossover assurance is mostly if not completely intact. Furthermore, crossover-proficient but SC-deficient *zip1* mutants like *zip1*[21-163] have high spore viability and no prophase delay. (Voelkel-Meiman et al., 2016, PLoS Genet.).

It is appealing to speculate that the CAPACITY for SC structures to disassemble and reassemble contributes to a crossover assurance mechanism by providing a "fresh start" (de novo recombination events) to homologous chromosomes that have either i) not yet undergone SC assembly, or ii) experienced SC disassembly due to a disruption in a joint molecule recombination intermediate. However, this study does not directly test this idea, that the SC's capability to disassemble provides crossover assurance to meiotic cells. (Personally, I think this idea should be discussed in this paper.)

One reason for the claim that SC maintenance is important for crossover assurance appears to be the experiment that found failed meiosis I upon Yen1-ON induction in NDT80+ cells (Fig. 4). The authors seem to conclude from this experiment that they have caused mature recombination intermediates to prematurely resolve, leading to SC disassembly, DNA damage signaling and consequent failure in meiosis I. There are alternative explanations for the data, however. Most critically, one cannot rule out the possibility that - in the experiment as performed - Yen1-ON may be interfering with the establishment of early recombination intermediates in a manner that causes meiotic failure. The next few points review my concerns with this experiment and its interpretation.

a. Line 232+ section is titled: "Premature resolution of recombination intermediates during pachytene disrupts the SC and causes meiosis I failure", but this is not a rigorous conclusion based on the data. For the conclusion the authors make, the experiment requires fully formed joint molecules prior to Yen1-ON induction. The authors assume that Yen1-ON has no activity/no impact on cells prior to 60 minutes (Line 243: "Since Yen1-ON accumulation requires 60-120 min ... we decided to follow the subpopulation initially in early zygotene ..."). I cannot think of a reason to expect Yen1-ON to hold off its activity in cells for 60-120 min. During time 0-120 min in early zygotene cells, Yen1-ON protein could well be interfering with DSB processing and the development of early recombination intermediates, preventing the accumulation of joint molecule structures.

b. The authors might argue that Yen1-ON must not be affecting joint molecule development between 0-120 min because, as they say in line 245: "As predicted, cells induced to express Yen1-ON in early zygotene successfully entered pachytene and reached full chromosome synapsis" However, how would the authors know that these live-imaged cells entered pachytene? The pachytene stage is cytologically defined by observing morphological changes in the DNA/chromosome axis. Chromosome axes become thicker and chromatin takes on a more condensed state (Greek term pachy = thick), allowing chromosomes to individualize from one another and be visible as distinct entities. DNA was not examined in these live imaging experiments, and I'm not sure that one could distinguish pachytene from zygotene in the small live cell nucleus even if DNA had been labeled. One needs to do chromosome spreads to analyze the morphological state of the chromatin AND the presence of SC structures (see below).

c. The authors might argue that live imaging shows that pachytene has been reached because of observed SC assembly. However, SCs do initiate from centromeres in addition to recombination sites, thus rudimentary SC assembly is not necessarily indicative of plentiful mature joint molecules. Even more importantly, there is a problem with "calling" SC formation for any particular mutant background using live imaging only; it can give misleading results. First, it is extremely challenging to quantitatively measure SCs in live nuclei to determine that they are "full length" and at maximal number. Second, nucleoplasmic Zip1 that is not in polycomplex nor assembled on chromosomes can appear as linear elements in the nucleus (i.e. "structured"), resembling SC structures (see MacQueen and Roeder, 2009, Curr. Biol.). The nucleus in the 50, 70, 130 and 170 min timepoints (Fig. 4i) could have small SC stretches emanating from centromeres only, or alternatively could have completely disassembled SC and only nucleoplasmic Zip1 structures. Chromosome spreads must be used in parallel in this experiment to assess and quantify SC assembly in the NDT80+ genetic background, to ensure that the conclusions made are rigorous and accurate.

d. Line 302: "Consistent with this view, premature resolution of recombination intermediates during pachytene triggers SC disassembly, de novo DSB formation, and DNA damage signaling ultimately disrupting entry into the first meiotic division." The interpretation is that Yen1-ON induction at early zygotene results in an impact ONLY at pachytene, at which stage otherwise normal SC disassembles, leading to DNA damage signaling that ultimately causes cells to halt progression through meiosis I. Given the data, however, one cannot rule out the possibility that Yen1-ON interfered with the early processing of recombination intermediates at zygotene, leading to the checkpoint arrest and secondarily little or no SC assembly.

e. The authors observed in this section that the induction of Yen1-ON at the early zygotene timepoint "prevented virtually all from reaching a state of full synapsis and completely blocked meiosis I nuclear division." The authors conclude that "premature processing of recombination intermediates during wild type meiosis leads to disruption of chromosome synapsis.

As a consequence, cell cycle progression is disrupted, most likely due to de novo DSB formation and ensuing DNA damage signalling.” To imply that the chromosome synapsis defect causes cell cycle arrest is not rigorous. The Yen1-ON induction causes a severe disruption in meiosis, but not necessarily because of disrupted chromosome synapsis per se. *ecm11* and *gmc2* mutants lack any SC structure, but (apart from a small delay in prophase progression) have good meiotic outcomes - nearly wild-type sporulation and spore viability. This argues strongly against an SC deficiency per se having a toxic effect on meiosis. As indicated above, the toxic effect on meiosis observed in the presented experiment may well have to do with Yen1 ON interfering early on with the formation of recombination intermediates (not with their resolution), and this early interference perhaps not only prevents SC assembly but also leads to a major cell cycle checkpoint.

f. To rigorously address premature resolution of intact joint molecules, the authors should instead do an experiment where cells are accumulated at late prophase because of an absence of Ndt80, then induce Yen1-ON, and an hour later induce/replenish Ndt80. This would allow the authors to observe the meiotic outcome of Yen1-ON only after mature joint molecules have formed.

2. It is important that the nuclease-dead Yen1 control protein binds to DNA joint molecules as well as Yen1-ON protein, particularly as the Yen1-ON protein (with nine Cdk consensus sites mutated to alanine) may have a higher binding affinity than the wild-type version of Yen1. The authors appear to have used the appropriate control protein in their experiments, as they call it the Yen1-ON-ND, and imply that it is the ON version of Yen1 (carrying the nine serine to alanine mutations) with catalytic mutations E193AE195A. However, it remains uncertain because the actual mutated residues are not listed in full and the authors instead refer to a prior study (ref 50), but that study seems to only use a Yen1-ND (i.e. a Yen1 protein with only the E193AE195A mutations). For full clarity the residues changed in each Yen1 protein should be explicitly stated in the main figure.

3. The study's introduction presents ZMM proteins, but not the fundamental structural components of the SC central region (Zip1 and Ecm11-Gmc2). This seems odd given that SC is a major dimension of the study. It should be made clear to the reader early on that Zip1 is a fundamental component of both ZMM/recombination ensembles and SC.

4. Lines 80-81: Citations should be provided: Humphryes et al., 2013, PLoS Genet. showed that Ecm11-Gmc2 is a component of the yeast SC central region, while Voelkel-Meiman et al., 2013, PLoS Genet. specifically demonstrated that Ecm11-Gmc2 is a component of the SC central element substructure.

5. The narrative in lines 94-113 begins by posing the possibility that “downstream formation of nascent recombination intermediates could play a role in the continuous re-establishment of chromosome synapsis. To examine this possibility...” This statement implies the idea that SCs are continuously falling apart and reestablishing de novo (in a normal meiosis). What is the basis for suggesting this? I think it could be misleading. Voelkel-Meiman et al., 2012, PLoS Genet. showed that SCs continuously accumulate central region material and progressively grow more substantial after several hours of *ndt80* arrest, which is not consistent with the idea that SCs continuously disassemble and then re-establish. Pollard et al., 2023, Front. Cell. Dev. Bio. analyzed SC assembly and found, for the most part, processive assembly (apart from a rare class of abortive events from small SC structures that were almost never seen in wild type). Furthermore, the experiment performed in this section does not address whether recombination intermediates play a role in “continuous re-establishment” of synapsis, but instead addresses whether SC maintenance relies on ongoing recombination intermediate formation. (The experiment involved Anchors Away-depletion of Rec104 and observation of diminished DSBs and halted accumulation of DNA joint molecules but no strong change in SC structures). The authors do make an appropriate rigorous conclusion at the end of the section (“de novo DSB formation, although initially required for SC assembly, is largely dispensable for SC maintenance”) but this conclusion addresses the question of SC maintenance, not “re-establishment” of SC on chromosomes.

6. Line 299: “recombination intermediates play a key role in the establishment and maintenance of chromosome synapsis, a process that most likely involves continuous polymerization of the SC”. The authors show that recombination intermediates stabilize SC structures, but their data do not address whether recombination intermediates promote the continuous polymerization of SC structures. The existence of a rudimentary or early SC structure might intrinsically promote its continuous polymerization, so in that sense maintenance of any kind of SC structure will indirectly promote its continuous polymerization. However, this sentence leaves the impression that somehow maintaining the rudimentary or early structure requires continuous polymerization of SC; to my knowledge there is no data that this is the case for budding yeast SC. When Voelkel-Meiman et al., 2012, PLoS Genet. looked for a “treadmilling” of SC components into and out of the structure, that study found only evidence for SC component addition, no evidence that SC components exited full-length SC structures to any measurable degree during prophase.

7. Line 124: “it is therefore likely that all ZMM proteins directly or indirectly require recombination intermediates for continued chromosome association during pachytene”. Because it is relevant to the mechanistic consideration of the data, either here or in the Introduction the narrative should briefly discuss what is known about the interdependencies of ZMM proteins (i.e. which of the ZMMs rely on which for their localization to recombination sites/chromosomes)?

8. Lines 146-160 are confusing. The narrative begins by posing the question: “To determine if recombination intermediates contribute to SC maintenance independently of ZMMs...” But the section ends with the conclusion: “...these observations suggest that dHJs are required for the continuous association of ZMMs with chromosomes...ZMMs prevent dHJ dissolution by non-crossover pathways, while also promoting SC maintenance.” The experimental question and conclusion do not align with one another.

I think the issue is the original question requires an experiment that removed ZMMs (to look for whether recombination intermediates promote SC maintenance in the absence of ZMMs), but the data suggested ZZS proteins may still bind joint

molecules and stabilize SC when Zip3/Sgs1 or Msh4/Sgs1 are co-depleted, thus the experimental approach was not capable of asking that original question. However, the experiment removing ZZS and co-depleting ZZS+Sgs1 does still address the question (and the findings of immediate SC disassembly in either situation indicate there is no role for recombination intermediates in stabilizing SC independent of ZMMs).

9. The authors provide data (Fig. 4) on de novo DSB formation when SCs are depleted in *ndt80* arrested cells, and in the Discussion (line 300) say: “our work also suggest that SC maintenance contributes to the suppression of DSB formation on chromosomes that have already successfully formed ZMM-stabilized dHJs.” This statement should situate the present data in context of earlier work indicating the same conclusion, particularly Subramanian et al., 2016, PLoS Biol., who reported that conditional depletion of Zip1 leads to accumulation of Hop1-pT318 and Mek1 and Mer2-dependent DSBs at several hotspots in *ndt80*-arrested cells.

10. The “control” nuclei in Fig. 2e,n do not look like they have “full synapsis”, even apart from the presence of polycomplex. Have the authors measured the cumulative lengths of SCs in *ndt80* vs. ZIP3-AID *ndt80* nuclei? Spore viabilities of the *zip3* null, *msh4* null, ZIP3-AID and MSH4-AID homozygotes should be provided (the legend says they are normal, but the actual numbers should be given; apologies if I missed it). This is particularly important since *zip3* null mutants actually display SC structures that initiate from centromeres. Also, polycomplex frequency should be plotted on the graphs as they were in earlier figures. Even if the AID-tagged ZMM proteins (Zip3-AID and MSH4-AID) are less than functional, the experiment still shows that these ZMMs play a role in SC stabilization; I do not believe the tagged proteins must be fully functional for this data to be conclusive. However, I do feel the study needs to be careful in presenting the fusion strains, so that the reader does not assume these proteins are necessarily fully functional. Also this is important for the reader to be able to fully interpret the findings – if the Zip3-AID or Msh4-AID is delayed in SC assembly, then SCs at the start of the depletion may be “weaker” (less material in the central region) than the SCs at the start of the Yen1-ON experiment, and easier to disturb upon ZMM removal. Again, I think the effect is clear and strong conclusions can be made about the role of ZMMs but the authors should be fully transparent in possible differences between the experimental contexts of Fig. 1 vs. Fig. 2.

11. Discussion in lines 311-321 proposes a model that is not aligned with existing mutant data. The authors suggest in this model that meiotic nuclei missing SC cannot proceed to the first meiotic division because of ongoing DSBR. However, *ecm11* and *gmc2* and several *zip1* non-null mutants proceed through meiosis with high spore viability despite the absence of SC. The authors need to incorporate this important information into their model.

12. Line 325: “For these reasons, we propose that dHJs are more than passive intermediate structures that form in the process of crossing over. They have a central regulatory function in coordinating meiotic progression with crossover assurance” This claim seems overblown. The DNA joint molecule is important for triggering SC assembly and for maintaining the structure, but the data do not point to a “regulatory” function for dHJs such that these joint molecules would be able to monitor SC lengths and up/down regulate them. Moreover, the evidence that SC structures themselves play a critical role in meiotic progression is slim, thus it does not seem likely that dHJs promote meiotic progression through their capacity to maintain SC structures. The fact that disassembled SC could allow chromosomes another chance at forming an interhomolog recombination event may well contribute to crossover assurance, but this mechanism seems more about the presence/absence of SC not whether dHJs actively regulate them.

Minor Comments

13. The Abstract states that joint molecules pose “a risk to chromosome segregation”. Is there evidence that DNA joint molecules pose more of a risk to chromosome segregation than unrecombined chromosomes or bivalents with a crossover? Evidence for this claim should be explained in the Introduction if it is to stay in the Abstract.

14. The Introduction ends by stating the dHJ-ZMM interplay coordinates meiotic progression with crossover assurance. If the authors keep this claim, crossover assurance should be defined here as it is a relatively obscure term.

15. It is interesting that dual removal of Zip3 and Sgs1 or of Msh4 and Sgs1 leads to a delay in SC disassembly, whereas removal of ZZS leads to rapid SC disassembly regardless of whether Sgs1 is co-depleted. The authors propose the delay in SC disassembly observed in Zip3+Sgs1 co-depletion is due to ZZS complex associating with the recombination intermediate (and protecting it from Sgs1). Have the authors tried to look for ZZS on chromosomes in the doubly-depleted Zip3 Sgs1 cells?

16. Fig. 3: The authors show that SCs can assemble de novo at late pachytene. It would be appropriate to cite prior work: Fig. S2 of Voelkel-Meiman et al., 2012, PLoS Genet. showed that *zip1*-depleted cells brought to mid-late prophase (20 hr or 26 hr *ndt80* arrest; aligned condensed chromosomes with substantial Red1 accumulated on axes but no SC) can undergo substantial SC assembly with frequent long SC stretches. This work did not test whether dHJs nor de novo DSBs are required but should be cited as earlier evidence that SC establishment on unsynapsed chromosomes at late prophase is possible.

17. Line 159: “in turn, ZMMs prevent dHJ dissolution by non-crossover pathways, while also promoting SC maintenance.” Do we know, at this late stage when the depletion is happening, that Sgs1 is promoting dissolution by a noncrossover pathway? Ext. Data Fig. 5c shows a reduction in crossovers at each timepoint in addition to a reduction in non-crossovers. Why then say Sgs1 is promoting the non-crossover path?

18. The section titled: “dHJs enable reversible SC disassembly” is challenging to wrap one’s head around; why not say

“dHJs enable SC re-assembly”?

19. Fig. 1i: Clarify: why are there CO products at 7/8 hr, prior to inducing Yen1-ON?

20. Fig. 1j: Clarify: for the quantification of joint molecules in +rapamycin/minus rapamycin conditions, were these from the cells with or without Yen1-ON exposure?

21. Fig. 1k says the graph plots Zip1 distribution/assembled SC as in the experiment in “j”, which refers back to “h”. This is confusing; if these nuclei are from the experiment described in h, then we expect to see fewer SCs after 8 hr in both DMSO- and rapamycin-treated cells (because of YEN1-ON induction) (like in “m”). Perhaps there should be an area above these graphs in j, k, l that indicates both dimensions of the experiment: the rapamycin exposure (or not) and the estradiol exposure (or not) (this is done to some extent in m already).

22. Error bars are missing on plots in Ext. Data Figs 1, 3, 11, and in main Fig. 2h,m (but present in Figs 1 and 3e).

23. Ext. Data Fig. 10a: Label the FACS analysis to indicate when zygotene entry is thought to be (at the 4 hr row). For clarity the authors should also indicate on the image as well as in the legend in e and h, exactly when B estradiol was applied relative to the 0 hr timepoint in (a) (indicate for all the hour after introduction into sporulation media so that they can all be cross-compared easily).

24. A note about data availability: The hundreds of images for each timepoint that were generated for the study are not included in the article (nor do I think they should be). But the statement probably should be modified so that it does not suggest absolutely all the data reported is in the manuscript itself.

Referee #2

(Remarks to the Author)

Henggeler et al. report about the interplay between mature meiotic recombination intermediates (double Holliday junctions, dHJs) and formation and maintenance of the synaptonemal complex (SC). Using a wealth of sophisticated, elegant and precise molecular biology tools, they revealed that dHJ are essential for the nucleation and maintenance of the SC through the protective action of ZMM proteins. Hence both normal and abnormal disruption of dHJs leads to SC disappearance and associated lack of chromosome synapsis without compromising the axis loop organization of chromosomes. Overall, they postulate a regulatory role of dHJs leading to crossover assurance, which is the “goal” of meiosis to ensure proper homolog segregation.

This is an excellent manuscript that provides a better understanding of the interplay between dHJ and SC, and ultimately about the essence of meiotic recombination which is crossover formation. The experiments are well introduced, well performed and conclusive. A list of mostly minor comments is provided below as they appear along the text. This reviewer feels that these results being directly connected to the formation of the class I interfering crossovers, the manuscript would benefit from at least some integration of these findings in light of crossover interference. It looks like the proposed model here for crossover formation and assurance could be self-sufficient without additional layers of regulation while the corresponding crossovers show interference. How are these different features integrated? Does this postulate that crossover assurance is independent from crossover interference?

- Fig. 1: The presentation is a bit misleading between panels i, j and l. It looks like these last two panels correspond to quantification of gels that are not exactly the one presented in panel i. Providing quantification of the gel shown would look relevant here.

- Lines 92-93: The experimental system used compellingly shows that dHJ are resolved without severely disrupting the axis-loop organization. However, this is expected since chromosomes segregate under their compacted form at anaphase I after resolution of recombination intermediates. Overall, it would be good that the authors distinguish between what is expected, based on what is already known, from what is brand new from the present work.

- Ext. Data Fig. 4m and lines 131-132: How to explain the stronger phenotype of the mutants of the ZMM genes compared to YEN1-ON that shows a milder Rec8 disorganization phenotype?

- Fig. 2 and Ext. Data Fig. 5: In Fig. 2i, how to explain that crossovers accumulation is not affected by the depletion of Zip3? One argument could be the persistence of Zip3 on recombination intermediates that ensures the crossover-biased resolution. However, the disappearance of JMs shows that they are rapidly processed after Zip3 depletion, which invalidates the previous argument. Hence, the doubling in crossovers from panel i is independent of Zip3, hence ZMMs.

- Ext. Data Fig. 5a-c: There is a clear drop in CO formation in ZIP3-AID SGS-AID, which is not seen in ZIP3-AID alone. Does it mean that some COs (half of the final amount) result from JMs processing by Sgs1, potentially in combination with structure specific nucleases?

- Fig. 4: It might be relevant to insist on the fact that crossovers are made after YEN1-ON induction. These crossovers represent physical connections between homologs which, in combination with cohesins that maintain sister chromatids together, should be suitable for accurate segregation, as supported by video 11 in an otherwise WT context. Therefore, in the absence of *ndt80* after YEN1-ON induction, homologous chromosomes are more than engaged together, they show

reciprocal exchanges. Hence, the fact that DSBs are formed again after YEN1-ON expression shows that dHJs and ZMM proteins ensure the DSB-repressive state, and this repressive state is independent of the homolog engagement per se as suggested by Thacker et al. (2014).

- Lines 260-261: The authors postulate that the cell cycle progression is likely due to “de novo DSB formation and ensuing DNA damage signaling”. Early expression of YEN1-ON likely resolves early recombination intermediates upstream of dHJ, leaving unrepaired the corresponding DSBs. These unrepaired DSBs should also significantly contribute to the disruption of the cell cycle progression. Overall, it may be wise to include early in the manuscript what is the consequence of expression of YEN1-ON i.e. unspecific resolution of any branched structure, including bona fide dHJ as well as early recombination intermediates, leading to crossover/noncrossover products and unrepaired DSBs, respectively.

- Chapter about Cdc5 starting line 265: The authors used the *mlh3Δ mms4mn slx1Δ yen1Δ* quadruple mutant as a “complete” resolvase mutant. As reported at least by the Hunter group (Zakharyevich et al., 2012), this mutant still shows crossover formation that is Sgs1-dependent. Therefore, the fact that the SC eventually disassembles in this quadruple mutant, although it is delayed, could indicate that dHJs are eventually resolved in an Sgs1-dependent manner without involving a second Cdc5 regulated pathway involving phosphorylation of SC components. Performing the experiment in such a quintuple background will solve this issue. Without such an experiment, it is impossible to conclude about the two pathways for SC disassembly proposed by the authors (see also lines 339-342).

- Fig. 5j: Note that based on the configuration of the heteroduplex DNA tracts, the dHJ resolution drawn is not seen in vivo, as reported by Marsolier-Kergoat et al. (2018).

Typos:

- Ext. Data Fig. 3b y-axis: rec104 instead of Rec10.

- Ext. Data Fig. 2f: There are two panels +B-estradiol @10h. Is the right panel 10 or 12h?

Referee #3

(Remarks to the Author)

This is an outstanding paper that will contribute enormously to multiple parts of our understanding of the meiotic process. The paper is comprised of beautiful molecular genetics and cytology. Most importantly, it stands as a 'complete study'. I am at a loss to suggest what other experiments might be done. It is a masterpiece in so many ways.

The most fundamental and unique feature of the meiotic process is a highly programmed series of interactions between homologues that collectively set the stage for their segregation at MI. Recombination-mediated processes occur in physical and functional linkage with meiotic axial chromosome structure, with interplay in both directions, before, during, and after formation and dissolution of the synaptonemal complex (SC), a highly conserved meiosis-specific structure that links homolog axes along their lengths. This program takes hours in budding yeast, but days or weeks in some organisms. Since DNA recombination itself can be completed in minutes, this dramatic prolongation of meiotic prophase is likely to reflect the complexity of whole chromosome events, especially homolog pairing. These considerations raise a fundamental question: How are the DNA events of recombination and whole chromosome processes locally coordinated in time and space?

The SC central components are required for reorganization of the recombination complexes (Rad51, Mer3, and Msh4) from an on-axis position to a between-axis (thus, on the SC central region) position concomitant with SC installation. Thus, whereas in most organisms, DSB-initiated recombinational interactions directly mediate both homology searching and homolog coalignment, the SC is required, through its central components, for the maintenance and/or turnover of the recombination proteins required for maturation of the DSBs into crossovers.

In this article, the authors use molecular genetic, cytological (classical and high resolution) and live imaging approaches to investigate the role of Holliday junctions (HJs) in SC stability. They exploit all the strengths of the *S. cerevisiae* model system to develop an ingenious experimental system that allows them to induce nucleolytic dissolution of HJs after homolog synapsis and analyze the effect on SC stability monitored by Zip1, Ecm11-Gmc2 and Smt3/SUMO.

The authors were able to show that dissolution of the HJs led to the disappearance of these three proteins between homologs and the formation of polycomplexes, without causing severe changes in the chromosome axis (Rec8). This dissolution of the SC is accompanied by a delocalisation of proteins described as binding to HJs and necessary for the formation of COs and the initiation of the SC (Zip3, Msh5 and Zip4). The authors then rigorously demonstrated that conditional depletion of these three proteins in pachytene also leads to the disassembly of the SC (marked by loss of Zip1 from the chromosome and disorganization of Rec8 pattern), associated with an increase in NCOs and a decrease in HJ. This result shows that these proteins are not only required for the formation of CO-recombination intermediates, but also for their stabilization and protection against dissolution by the Sgs1-Top3-Rmi1 (STR) complex. Finally, the authors showed that dissolution of the SC, while maintaining the presence of the HJs, allows reassembly of the SC without the formation of new DSBs. The proposed model is based on these results and gives a central regulatory function in coordinating meiotic progression with crossover assurance.

The paper is well written, with detailed explanations and appropriate conclusions. The article is dense, with a lot of data, but the rigorous approach and the logical presentation of the results make it easy to read. The data is well analyzed and

presented. Sample sizes, error bars and P-values are given. The statistical tests used are well chosen. I look forward to seeing it published.

Minor comments

- Lines 139-145: Zip3 depletion increases NCOs, but COs are unchanged. In *S. cerevisiae*, Zip3 is a marker for sites that are destined to become COs. What could explain why the number of COs does not vary, whereas it decreases when ZIP3 and SGS1 are removed? In addition, it seems to me that the JM decrease is much greater than the increase in NCOs (Fig. 2h,i). Similarly, when ZIP3 and SGS1 are removed, there is an increase in JMs and a decrease in COs and NCOs (Ext. Data Fig. 5c). This increase appears to be greater than the decrease in COs and NCOs. Can this difference be explained by the distribution of new DSBs?

- Line 150 – “Similar results were obtained by combining conditional depletion of Msh4AID and Sgs1AID (Extended Data Fig. 5d,e).” What about JM, CO and NCO in the absence of Msh4 and MSH4 SGS1?

Version 2:

Reviewer comments:

Referee #1

(Remarks to the Author)

I remain positive about the experiments in this study and the impact they have on considering how resolution of crossover intermediates might coordinate mechanistically with other cellular processes at the end of meiotic prophase. I remain concerned about the lack of rigor in the Title statement and in similar statements made within the manuscript (such as the last sentence of the introduction), where a claim specifically about the maintenance of synaptonemal complex is made.

OE of Yen1 (as the authors show) leads to at least two things: i) a dissolution of the mature, tripartite synaptonemal complex (SC) and ii) a dissolution of ZMM (recombination) structures. The functional consequence of OE of Yen1 and concomitant induction of Ndt80 (Ext Data Fig. 10) is severe: meiosis I and II divisions fail to complete, and chromosome bridges are observed. However, the conclusion from this functional experiment should be that loss of EITHER ZMM (recombinosome) structures OR SC (or both) may cause the meiotic catastrophe observed. Instead, the title states that maintenance of synapsis enables crossover assurance. This is a misleading statement, as the experimental data do not specifically implicate the SC in the severe meiotic failure observed when HJs are prematurely resolved (Yen1-OE). Furthermore (as I pointed out in my prior review), one would expect SC-deficient mutants to show this sort of meiotic catastrophe, but SC-deficient mutants do not show this phenotype.

I agree with the authors that some SC-deficient mutants (i.e. *ecm11* or *gmc2* null mutants) have increased DSBs and DNA repair processes and that these mutants also show a prophase delay (although I note that the *zip1*[Δ 21-163] separation-of-function mutant lacks SC structures but does not exhibit the prophase delay). I also agree with the authors that a deficiency in chromosome synapsis (deriving from the untimely processing of recombination intermediates) results in ongoing formation of DSBs and DSB repair intermediates, which results in increased DNA damage signalling and triggers the recombination checkpoint, causing a prolonged prophase I... There is good evidence for this model from prior studies, as the authors point out. The issue with the assertion made in the current manuscript Title, that maintenance of the SC “enables crossover assurance”, is that the claim is not rigorously supported by the data provided in the manuscript (alternative explanations exist for the severe meiotic division defect observed when Yen1 is overexpressed) and does not align with the fact that SC-deficient mutants (*ecm11* or *gmc2*) exhibit robust (more than a wild type number of) crossovers and do NOT show meiosis I division failure.

Another way to put my concern: It is formally possible that the experiment in Ext. Data Fig. 10m-p (overexpression of Yen1 and release from *ndt80* block) - if carried out in an *ecm11* or *gmc2* mutant (which lacks SC in the first place) - would show the same severe defect in meiotic progression and anaphase bridges. This outcome would indicate that the meiotic catastrophe observed upon Yen1-OE is not due to failure to maintain a mature SC structure, but instead due to failure in something else – that something else could be a failure to maintain ZMM/recombinosome ensembles and/or a failure to properly coordinate resolution with cohesin/condensin remodeling at the exchange site.

While catastrophic chromosome segregation/meiotic failure is not the phenotype of an *ecm11* or *gmc2* SC-deficient mutant, the authors point out (line 344) that a meiotic cell missing HJ resolvases does show this severe meiotic failure. This raises the third possibility that the meiotic failure observed in Ext. Data Fig. 10m-p is due to a few recalcitrant HJ junctions that fail to resolve properly when Yen1 is overexpressed.

Because of alternative explanations for the meiotic failure observed when Yen1 is overexpressed (that some HJs are not resolved or that ZMM dissolution at the developing chiasmata is the cause instead of absence of SC), in conjunction with the knowledge that SC structures are dispensable for meiotic success and robust crossover recombination, it is misleading and unrigorous to claim that the maintenance of SYNAPSIS (the SC) is what is functionally important for crossover assurance.

Finally, I wish to respond to the following comment the authors made in their response:

[[[The reviewer states that “*ecm11* and *gmc2* mutants lack any SC structure,”. Taken literally, this statement is in direct disagreement with the literature (Humphryes et al. 2013) and with the data in Figure for Reviewers 2e-g, in which it is clear

that Zip1 can still associate with the chromosomal axes in ecm11 gmc2 mutants.]]]

With respect, I completely disagree with this assertion. Zip1 accumulation on chromosome axes is not equivalent to the synaptonemal complex, just as tubulin accumulation on chromosomes is not equivalent to a spindle. In fact, authors from the Humphryes et al. (2013) study collaborated with authors from the Voelkel-Meiman (2013) study to show (with superresolution microscopy) that the Zip1 accumulating on chromosome axes in the ecm11 mutant fails to arrange into a tripartite synaptonemal complex structure. Zip1 accumulates on axes in the absence of Ecm11 or Gmc2, just as Ecm11 protein can accumulate on chromosomes in certain situations when Zip1 is altered. To say that accumulation of any SC structural component on a chromosome axis is equivalent to synaptonemal complex formation is dismissive of the grand SC structure itself (and certainly not rigorous).

Referee #2

(Remarks to the Author)

The authors appropriately addressed my concerns.

Version 3:

Reviewer comments:

Referee #1

(Remarks to the Author)

The authors have addressed my few concerns in their final version of this compelling study. They use creative approaches to reveal new and very interesting information about late meiotic prophase chromosome structures, structures that are critical for maintaining genome integrity across generations. Finally, the findings are communicated clearly, thoroughly, and with rigor.

Referee #2

(Remarks to the Author)

The points raised by referee 1 in the previous round of review have been satisfactorily addressed.

We thank the reviewers for their positive and constructive feedback, which has helped us improve the manuscript further. Below, we provide a point-by-point response to their remaining comments:

Note 1: Author responses are highlighted in blue.

Note 2: Line numbers refer to the revised manuscript unless otherwise specified.

Note 3: Figure numbers refer to the revised manuscript unless otherwise specified.

Note 4: Figures for Reviewers 1–3, containing data not included in the revised manuscript, are embedded in the rebuttal letter.

Referees' comments:

Referee #1:

Reciprocal exchange between homologous DNA duplexes (crossovers) are linked - physically and functionally - to a supramolecular structure called synaptonemal complex (SC). Both crossovers and SC are hallmark features of meiosis. Many aspects of the relationship between DNA recombination intermediates and synaptonemal complex remain unclear. Using conditional induction of a constitutive form of the Yen1 resolvase in meiotic prophase-arrested cells (cells containing unresolved DNA joint molecule recombination intermediates), the authors demonstrate that ectopic resolution of joint molecule meiotic recombination intermediates can destabilize ZMM protein ensembles, which play a critical role in the formation of those very same DNA joint molecule intermediates. The authors also show that ZMM ensembles play a continuous role in stabilizing/protecting the joint molecule from inappropriate resolution, which is consistent with indirect data published previously. Thus, the authors here have established that at late meiotic prophase, ZMMs preserve DNA joint molecules AND that DNA joint molecules preserve the ZMM ensemble. In addition, the authors show that ectopic resolution of DNA joint molecules during mid-late prophase of meiosis can destabilize SC structures (central region components disappear but SC axis structures remain intact), which rely on ZMM protein function for their formation.

Another major finding of this study, which was obtained by inducing a version of Ulp1 that localizes to the nucleus instead of nuclear pores, is that the removal of SC itself does not trigger joint molecule resolution, and SCs can in fact re-establish after their removal so long as joint molecules have not resolved.

Also, I found interesting the data suggesting ZZS can partially stabilize joint molecules and SC structures when Zip3 and Msh4 are removed, so long as the Sgs1 helicase is also missing.

The experiments were performed in a rigorous manner and the collected data support the authors' interpretations, but with one important exception noted below.

The findings here are impactful. It is not necessarily surprising that ZMMs play a continued role in preserving DNA joint molecules after their initial establishment and that DNA joint molecules play a continued role in preserving associated ZMM ensembles (as with any complex, the components each play a role in the final architecture; removing one of the components, DNA or protein, will change that architecture and possibly destabilize the entire ensemble). It is also not completely surprising that SC structures rely on the integrity of ZMM ensembles, particularly since a major structural component of the SC in budding yeast is also a key component of ZMM ensembles (Zip1). However, to my knowledge this is the first study

to directly demonstrate the existence of a reciprocal dependency between the DNA joint molecule and the protein structures that house it, during late prophase.

The study's major findings are also a significant contribution to the field because of how they speak to what might happen at the end of prophase in normal meiosis. While joint molecule resolution and SC disassembly normally occur nearly simultaneously at the end of meiotic prophase, it has been unclear whether SC disassembly triggers resolution of JMs or whether resolution of JMs might occur first to trigger SC disassembly. The authors have gone on to specifically address this question in more detail, by examining if ANY SC disassembly can occur at the end of meiotic prophase (given the proper upstream signals) in a mutant that lacks all four major resolvase enzymes. The authors found that SC does ultimately disassemble in this situation, but with a delay. These data are the first (to my knowledge) to indicate that there is likely a combination of mechanisms that promote the disassembly of SC during normal meiosis, and one of these contributing mechanisms is DNA joint molecule resolution itself.

We thank the reviewer for providing thorough feedback on our manuscript and for acknowledging the quality, significance and impact of our work.

However, as noted above there is one claim made that is not warranted by the study's experimental findings. That SC maintenance promotes crossover assurance is not a rigorous conclusion based on the experimental data reported, and is not a claim supported by existing mutant data.

There are some areas of the narrative where greater clarity or more information should be provided to enable a full understanding of the new findings and how to situate them with prior work in this area.

Major comments

1. The experiments do not provide rigorous evidence that DNA joint molecule-dependent maintenance of chromosome synapsis functions to "enable crossover assurance", as is stated in the title. In addition, contrary evidence to this claim exists: *ecm11* or *gmc2* null mutants (missing fundamental structural components of the SC) have high spore viability, indicating that even when SC (synapsis) is absent, meiosis proceeds (albeit with a small delay at late prophase) and crossover assurance is mostly if not completely intact. Furthermore, crossover-proficient but SC-deficient *zip1* mutants like *zip1*[Δ 21-163] have high spore viability and no prophase delay. (Voelkel-Meiman et al., 2016, PLoS Genet.).

We appreciate the feedback from the reviewer and acknowledge the importance of clarifying the interpretation of our findings in light of what is known for SC central element mutants. As detailed in the three response points below, we believe that our interpretation is NOT contradicted by the existing literature. This is further supported by new experimental data in Figure for Reviewers 1 and 2.

1. Prophase I is prolonged in SC central element mutants

Evidence of a significant delay. Contrary to the assertion that *ecm11* Δ or *gmc2* Δ mutants show only minimal prophase progression delay, multiple studies report a pronounced delay. For example, Humphries et al. (2013) showed significantly reduced sporulation in these

mutants (only ~2% by 28 hours in sporulation media vs. ~25% in wild type). Voelkel-Meiman et al. (2016) reported that almost all of the *ecm11Δ* nuclei remained in pachytene at 28 hours, whereas > 50% of wild-type nuclei had progressed to later stages.

To independently test if - in our experimental conditions - SC central element mutants display delayed meiotic progression, we have performed live-cell imaging of *ecm11Δ gmc2Δ* strains. To monitor the duration of prophase I with unprecedented precision we generated a functional internally yEGFP-tagged version of Rad51 (Rad51^{iyEGFP}), as described in Liu et al. (2023) (Figure for Reviewers 1a). Rad51^{iyEGFP} formed foci (and occasionally threads) in control prophase I cells, but not in *spo11Δ* mutants, in which it displayed a diffuse nuclear signal (Figure for Reviewers 1b). When we measured the duration from Rad51^{iyEGFP} foci formation (early DSB repair) to their resolution, we found that *ecm11Δ gmc2Δ* mutants took **1.6 x longer** than control cells (Figure for Reviewers 1c-e). Since Rad51^{iyEGFP} shows a diffuse nuclear signal during meiotic divisions, we were also able to estimate the time from leptotene to the onset of anaphase I (Figure for Reviewers 1f). This analysis showed a similar increase in duration in *ecm11Δ gmc2Δ* mutants, indicating that prophase I is substantially – and specifically – extended in the absence of the central elements of the SC.

FIGURE REDACTED

FIGURE LEGEND REDACTED

2. DSB repair intermediates and DNA damage response activation delay meiotic progression in *ecm11Δ gmc2Δ* mutants

It is well established that SC central region mutants experience increased DSB formation (1.7-1.8x wild type levels; Mu et al. 2020, Lee et al. 2021) and elevated crossover frequency (1.1-2.8x wild type, depending on genetic interval; Voelkel-Meiman et al. 2016). Our analysis of Rad51^{iyEGFP} suggests, without providing formal proof, that the steady state DSB number may not be increased at any given time in individual cells but rather that DSB measurements reflect an overall increase at the population level. We think that this is the case because we did not observe a noticeable increase in foci number nor an overall increase in Rad51^{iyEGFP} signal intensity in *ecm11Δ gmc2Δ* mutants. Our data therefore show that *ecm11Δ gmc2Δ* mutants cells spend on average much longer with DSB repair intermediates, possibly due to more cycles of DSB formation/repair (Figure for Reviewers 1c-e).

In light of these observations, the delay observed in prophase I (Figure for Reviewers 1f) could be explained by prolonged activation of the DNA damage response/recombination checkpoint. To test this possibility we performed conditional depletion of Rec104, using an auxin-inducible degron allele (*REC104^{AID}*), and monitored Rad51^{iyEGFP} and meiosis I nuclear division. Rec104^{AID} depletion in cells that had already initiated DSB formation (pre-accumulated Rad51) led to earlier Rad51^{iyEGFP} disappearance and premature anaphase I entry (~150 minutes vs. ~265 minutes in controls), indicating that ongoing DSB formation contributes to the prophase I delay in *ecm11Δ gmc2Δ* mutants (Figure for Reviewers 1g-k). Notably, conditional DSB down-regulation also resulted in premature, as well as defective, meiosis I nuclear division, most likely due to insufficient crossover formation (Figure for Reviewers 1h,j, l and m).

These data indicate that persistent DSB formation/repair cycles eventually lead to the formation of sufficient crossovers that can sustain chromosome segregation and spore viability in *ecm11Δ gmc2Δ* mutants.

3. Partial synapsis in *ecm11Δ gmc2Δ* mutants

The reviewer writes “...*ecm11* or *gmc2* null mutants (missing fundamental structural components of the SC) have high spore viability, indicating that even when SC (synapsis) is absent,...”. Even though formally correct, this statement could be perceived as implying that *ecm11* or *gmc2* null mutants do not retain the ability to assemble any parts of the SC. We would like to point out that such an interpretation would be incorrect. Despite the well documented defects in SC polymerization, Zip1 has been reported to localize to meiotic chromosomes as foci and short linear stretches in *ecm11Δ gmc2Δ* mutants (Humphryes et al. 2013). We confirmed this finding using live-cell imaging and STED microscopy. Tracking Zip1^{mCherry} in living cells co-expressing Rad51^{iyEGFP} revealed a significantly delayed but robust accumulation of Zip1^{mCherry} signal in *ecm11Δ gmc2Δ* mutants, which appeared to be partially structured (Figure for Reviewers 2a-d). We emphasize that the Zip1^{mCherry} signal was generally less structured than in wild type, likely reflecting defective assembly (Figure for Reviewers 2a, b). Parallel STED microscopy analyses of Zip1 and Rec8-containing cohesin on chromosome spreads showed that, despite taking significantly longer, *ecm11Δ gmc2Δ* mutants still assembled large Zip1 stretches (Figure for Reviewers 2e, f). Moreover, analyses of inter-axes distances (marked by Rec8) showed that, at the time in which large Zip1 stretches assemble in *ecm11Δ gmc2Δ* mutants, the distance between homolog axes is very similar to control cells (Figure for Reviewers 2g). This indicates that homologs are eventually brought into close proximity in the absence of Ecm11 and Gmc2.

These findings suggest that the partial loading of SC components in *ecm11Δ gmc2Δ* mutants could facilitate DSB down-regulation over time, though less efficiently than in wild type. Additionally, alternative mechanisms, such as Tel1/ATM signalling or checkpoint adaptation, may contribute further to limiting DSB formation during the extended prophase I in *ecm11Δ gmc2Δ* mutants. Similar explanations are also likely to apply to the separation of function allele of Zip1 (*zip1-N1*; *zip1[Δ21-163]* mutant), where both the mutant Zip1 version and Ecm11 are not absent from meiotic chromosomes, but are observed as foci and linear stretches (Voelkel-Meiman et al. 2016).

In conclusion, for the reasons outlined above, we are convinced that *ecm11Δ gmc2Δ* mutants generate sufficient crossovers to support meiotic nuclear division **NOT** because they “still” have properly functioning crossover assurance, but rather due to mis-regulated DSB formation/repair, which ultimately results in enough crossovers for chromosome segregation. Related phenotypes have been described in other mutants, such as *sgs1Δ*, in which excess recombination intermediates compensate for the lack of conventional DNA JM maturation (de Muyt et al., 2012). We view the data for *ecm11Δ gmc2Δ* mutants as supportive of our model. It is consistent with the SC having a role in the suppression of DSB formation (defective in *ecm11Δ gmc2Δ* mutants) in response to the accumulation of crossover precursors containing double Holliday junctions. *ecm11Δ gmc2Δ* mutants fail to efficiently establish this negative feedback, leading to prolonged formation of DSBs and a prophase I delay.

Although we could include these new data in the manuscript, we do not believe they would strengthen it. We have discussed this matter with several experts in the field and the consensus was that, given the well-established literature showing that SC central element mutants exhibit elevated DSB formation and delayed prophase I progression, the overall phenotypes actually support our model. We also note that Reviewers 2 and 3 did not share concerns regarding the literature on SC central element mutants.

FIGURE REDACTED

FIGURE LEGEND REDACTED

It is appealing to speculate that the CAPACITY for SC structures to disassemble and reassemble contributes to a crossover assurance mechanism by providing a “fresh start” (de novo recombination events) to homologous chromosomes that have either i) not yet undergone SC assembly, or ii) experienced SC disassembly due to a disruption in a joint molecule recombination intermediate. However, this study does not directly test this idea, that

the SC's capability to disassemble provides crossover assurance to meiotic cells. (Personally, I think this idea should be discussed in this paper.)

We agree with the reviewer that the reversible properties of the SC structure may constitute an important feature of a crossover assurance mechanism. Indeed, we have shown that disruption of SC maintenance, either by premature dHJ resolution or conditional ZMM depletion, triggers de novo DSB formation and activation of the DNA damage response (Figure 4a-h; Extended Data Fig. 9). This converts a local synapsis "problem" into a cellular response - to DNA damage - that precludes meiotic progression. From our point of view, and given the space limitations, the idea of SC maintenance as a mechanism for crossover assurance has been sufficiently discussed in the manuscript (see lines 333-356).

One reason for the claim that SC maintenance is important for crossover assurance appears to be the experiment that found failed meiosis I upon Yen1-ON induction in NDT80+ cells (Fig. 4). The authors seem to conclude from this experiment that they have caused mature recombination intermediates to prematurely resolve, leading to SC disassembly, DNA damage signaling and consequent failure in meiosis I. There are alternative explanations for the data, however. Most critically, one cannot rule out the possibility that - in the experiment as performed - Yen1-ON may be interfering with the establishment of early recombination intermediates in a manner that causes meiotic failure. The next few points review my concerns with this experiment and its interpretation.

We appreciate the reviewer's insights and acknowledge that the nucleolytic activity of Yen1^{ON} is unlikely to be restrained upon expression. However, we would like to clarify key experimental aspects that support our conclusion that premature resolution of recombination intermediates during pachytene disrupts SC integrity and leads to meiosis I failure. In addition, prompted by really good suggestions from the reviewer, we have conducted new experiments that further reinforce our findings (see point f, below).

a. Line 232+ section is titled: "Premature resolution of recombination intermediates during pachytene disrupts the SC and causes meiosis I failure", but this is not a rigorous conclusion based on the data. For the conclusion the authors make, the experiment requires fully formed joint molecules prior to Yen1-ON induction. The authors assume that Yen1-ON has no activity/no impact on cells prior to 60 minutes (Line 243: "Since Yen1-ON accumulation requires 60-120 min ... we decided to follow the subpopulation initially in early zygotene ..."). I cannot think of a reason to expect Yen1-ON to hold off its activity in cells for 60-120 min. During time 0-120 min in early zygotene cells, Yen1-ON protein could well be interfering with DSB processing and the development of early recombination intermediates, preventing the accumulation of joint molecule structures.

While Yen1^{ON} is expected to become active on recombination intermediates once expressed, folded and imported into the nucleus, we want to point out that protein accumulation using the GAL4-ER system is not immediate - it occurs relatively slowly for the Yen1^{ON}-myc18 fusion, which is ~130 kDa. In Extended Data Fig. 1b-c, we plotted Yen1^{ON} accumulation and it can be seen that 1h after induction (8h time point) the protein is already detectable, but that it keeps increasing until 2h after induction. The experiment in Figure 1l shows that JM processing by Yen1^{ON} can only be detected 1-2 h after induction, which correlates with higher levels of

Yen1^{ON} protein (new data in Extended Data Fig. 3f, g). This fits very well with the timing of SC loss, which inversely correlated with the increase in Yen1^{ON} levels (Figure 1d).

To resolve better the timing of Yen1^{ON} accumulation, we have performed a new experiment in which we collect protein samples at 15 min intervals upon induction (new data in Extended Data Fig. 10c, d). We observed that, even though detectable at very low levels 30 min after induction, robust Yen1^{ON} detection takes 45-60 minutes, with the peak of expression occurring much later. For these reasons, we are convinced that the activity of Yen1^{ON} is not immediate upon induction of its expression from the GAL1 promoter. We conservatively estimate that it will take at least 45 min for Yen1^{ON} to be present in the nucleus in sufficient amounts to trigger JM processing.

In the live cell imaging experiments in Figure 4i-l and Extended Data Figure 10a-l, we estimated the duration of the zygotene stage to be ~55 min based on monitoring SC dynamics using Zip1^{GFP}. This estimate agrees well with data from previous live cell imaging studies using the same Zip1^{GFP} construct (White et al. 2004). For our analysis of the effect of Yen1^{ON} expression during pachytene (Figure 4i-l; Extended Data Figure 10e, f), we focused on cells that were in early/mid zygotene at the time of β -estradiol addition. More precisely, we only analysed cells that already showed obvious early SC precursors within the first 30 min of imaging before the addition of β -estradiol. As such, these cells were already well advanced in the zygotene stage at the time of Yen1^{ON} induction. Thus, by the time Yen1^{ON} expression is first detected by Western blotting (around 30-45 min), **most cells** should have “safely” reached the pachytene stage.

As discussed in response to point f, below, we have performed additional experiments that are fully in line with these findings and their interpretation.

b. The authors might argue that Yen1-ON must not be affecting joint molecule development between 0-120 min because, as they say in line 245: “As predicted, cells induced to express Yen1-ON in early zygotene successfully entered pachytene and reached full chromosome synapsis” However, how would the authors know that these live-imaged cells entered pachytene? The pachytene stage is cytologically defined by observing morphological changes in the DNA/chromosome axis. Chromosome axes become thicker and chromatin takes on a more condensed state (Greek term pachy = thick), allowing chromosomes to individualize from one another and be visible as distinct entities. DNA was not examined in these live imaging experiments, and I’m not sure that one could distinguish pachytene from zygotene in the small live cell nucleus even if DNA had been labeled. One needs to do chromosome spreads to analyze the morphological state of the chromatin AND the presence of SC structures (see below).

We agree with the reviewer that it is difficult to assess the exact prophase I substage based on an individual snapshot of Zip1^{GFP}. Due to the large number of chromosomes packed into a small nucleus, resolution limitations, and extensive chromosome movements in living cells, it is difficult to assess staging with high precision. However, we kindly ask the reviewer to take into consideration that live cell imaging gives us the advantage of being able to follow the entire history of each cell and thus know its cell cycle progression prior to any experimental manipulation. Moreover, since cells follow a stereotypical progression, with similar timings (see for example Extended Data Figure 10b), we are confident that we can accurately

determine at which stage most cells are based on Zip1^{GFP} signal intensity, structure and their entire history.

The new data generated in response to point f, in which we used a modified/complementary setup suggested by the reviewer, supports very nicely our observations and data interpretation.

c. The authors might argue that live imaging shows that pachytene has been reached because of observed SC assembly. However, SCs do initiate from centromeres in addition to recombination sites, thus rudimentary SC assembly is not necessarily indicative of plentiful mature joint molecules. Even more importantly, there is a problem with “calling” SC formation for any particular mutant background using live imaging only; it can give misleading results. First, it is extremely challenging to quantitatively measure SCs in live nuclei to determine that they are “full length” and at maximal number. Second, nucleoplasmic Zip1 that is not in polycomplex nor assembled on chromosomes can appear as linear elements in the nucleus (i.e. “structured”), resembling SC structures (see MacQueen and Roeder, 2009, *Curr. Biol.*). The nucleus in the 50, 70, 130 and 170 min timepoints (Fig. 4i) could have small SC stretches emanating from centromeres only, or alternatively could have completely disassembled SC and only nucleoplasmic Zip1 structures. Chromosome spreads must be used in parallel in this experiment to assess and quantify SC assembly in the NDT80+ genetic background, to ensure that the conclusions made are rigorous and accurate.

Using chromosome spreads, we do not observe extensive Zip1 stretches above background in cells that have completely disassembled the SC after Yen1^{ON} expression (Figure 1b; Extended Data Figure 2f). Similarly, cells that have completely disassembled the SC after Zip3^{AID}, Msh4^{AID} or Zip4^{AID} depletion show no obvious Zip1 signal above background (Figure 2e; Extended Data Figure 4h and i). Therefore, we argue that extensive centromeric synapsis is unlikely to occur after Yen1^{ON}-mediated SC disassembly.

It is worth noting that there are some differences in the budding yeast strain backgrounds used for the analysis of meiotic events, in particular between the SK1 strain (used in this study and commonly used in the field) and the BR1919 strain background (referred to by the Reviewer). SK1 strains can reach pachytene with full chromosome synapsis within 4-5 hours after meiosis induction (Padmore et al. 1991), whereas the BR strains spend approximately 11 hours undergoing SC formation and reach pachytene at 16-21 hours after meiosis induction (Voelkel-Meiman et al. 2012; Pollard et al. 2023). The shorter time taken to establish chromosome synapsis likely suggests that other regulatory controls, such as the number and timing of synapsis initiation sites, may be responsible for synapsis in the two different strain backgrounds. Indeed, in SK1 strains, almost all SC assembly is initiated at recombination sites (Henderson et al. 2004, 2005), whereas in BR1919 strains, synapsis from centromeres, in addition to synapsis from recombination sites, may play an important role in establishing full synapsis (Fung et al. 2004; Tsubouchi et al. 2008; MacQueen et al. 2009).

As mentioned above, live cell imaging has the advantage of allowing us to follow an individual cell through prophase I and thus know its cell cycle progression history. By comparing the intensity of the Zip1^{GFP} signal and the timing of prophase I progression, as well as comparing the signals between control and β -estradiol-treated cells, we think we can exclude that the signal is exclusively from nucleoplasmic Zip1^{GFP}, but rather from proper chromosome synapsis. The nucleoplasmic Zip1^{GFP} signal becomes very apparent in our experiments using Ulp1 ^{Δ N-FRB} expression to induce SC disassembly in pachytene arrested *ndt80* Δ mutants

(Supplementary Videos 5 and 6). While the nucleoplasmic Zip1^{GFP} signal appears to be structured next to a more prominent polycomplex, the signal was not comparable to the thread-like, intense Zip1^{GFP} signal observed before induction.

Finally, we respectfully disagree with the reviewer that chromosome spreads would be useful in these experiments (*NDT80* wild type). Given the limited synchrony of meiotic populations, it would be extremely challenging (most likely impossible) to ascertain which subset of the population would be affected by Yen1^{ON} expression and at which stage.

d. Line 302: "Consistent with this view, premature resolution of recombination intermediates during pachytene triggers SC disassembly, de novo DSB formation, and DNA damage signaling ultimately disrupting entry into the first meiotic division." The interpretation is that Yen1-ON induction at early zygotene results in an impact ONLY at pachytene, at which stage otherwise normal SC disassembles, leading to DNA damage signaling that ultimately causes cells to halt progression through meiosis I. Given the data, however, one cannot rule out the possibility that Yen1-ON interfered with the early processing of recombination intermediates at zygotene, leading to the checkpoint arrest and secondarily little or no SC assembly.

This issue has been addressed above (point a) and is further addressed in response to point f, below.

e. The authors observed in this section that the induction of Yen1-ON at the early zygotene timepoint "prevented virtually all from reaching a state of full synapsis and completely blocked meiosis I nuclear division." The authors conclude that "premature processing of recombination intermediates during wild type meiosis leads to disruption of chromosome synapsis. As a consequence, cell cycle progression is disrupted, most likely due to de novo DSB formation and ensuing DNA damage signalling." To imply that the chromosome synapsis defect causes cell cycle arrest is not rigorous. The Yen1-ON induction causes a severe disruption in meiosis, but not necessarily because of disrupted chromosome synapsis per se. *ecm11* and *gmc2* mutants lack any SC structure, but (apart from a small delay in prophase progression) have good meiotic outcomes - nearly wild-type sporulation and spore viability. This argues strongly against an SC deficiency per se having a toxic effect on meiosis. As indicated above, the toxic effect on meiosis observed in the presented experiment may well have to do with Yen1 ON interfering early on with the formation of recombination intermediates (not with their resolution), and this early interference perhaps not only prevents SC assembly but also leads to a major cell cycle checkpoint.

We do not argue that the absence of a mature SC structure per se is directly detrimental to meiotic progression, but rather that a deficiency in chromosome synapsis (deriving from the untimely processing of recombination intermediates) results in ongoing formation of DSBs and DSB repair intermediates. This results in increased DNA damage signalling and triggers the recombination checkpoint, which is responsible for prolonging prophase I. This interpretation is discussed in lines 280-283 and 329-332.

As discussed above, we do not agree with the reviewer that *ecm11*Δ or *gmc2*Δ mutants have 'only a small delay in prophase progression' (Figure for Reviewers 1). The prolonged prophase I is not a direct consequence of the lack of a polymerized SC structure, but rather a consequence of ongoing DSB formation that triggers the recombination checkpoint.

Consistent with this, inhibition of DSB formation in prophase I by depletion of Rec104^Δ leads to earlier progression to the first meiotic division (Figure for Reviewers 1g-k), and the delay in sporulation in *ecm11Δ* and *gmc2Δ* mutants is largely alleviated by deletion of *SPO11* (Humphryes et al. 2013).

The reviewer states that “*ecm11* and *gmc2* mutants lack any SC structure.” Taken literally, this statement is in direct disagreement with the literature (Humphryes et al. 2013) and with the data in Figure for Reviewers 2e-g, in which it is clear that Zip1 can still associate with the chromosomal axes in *ecm11 gmc2* mutants.

Finally, we would like to ask the reviewer to take into consideration that NOT having Ecm11/Gmc2 is different from having a problem in assembling Ecm11/Gmc2 into the SC. It is entirely possible to envision that the central element proteins could have both a structural role in building the SC and a signalling role in delaying meiotic progression in conditions that lead to synapsis defects. In other words: *ecm11Δ gmc2Δ* mutants may not arrest meiotic progression indefinitely because their protein products could be part of the signalling mechanism that delays meiotic progression.

f. To rigorously address premature resolution of intact joint molecules, the authors should instead do an experiment where cells are accumulated at late prophase because of an absence of Ndt80, then induce Yen1-ON, and an hour later induce/replenish Ndt80. This would allow the authors to observe the meiotic outcome of Yen1-ON only after mature joint molecules have formed.

We thank the reviewer for suggesting this experimental design, which strongly complements our experiments in *NDT80* (wild type) cells and, as discussed below, makes our conclusions even stronger.

In order to combine conditional expression of Yen1^{ON} with conditional expression of Ndt80, we used *P_{GAL}-NDT80* (Matos et al. 2008, Carlile et al. 2008) and placed Yen1^{ON} under the control of the copper-inducible promoter (*P_{CUP1}*) (Janke et al. 2004). This strong promoter typically leads to a very rapid and high expression of the protein under its control (Janke et al. 2004). Indeed, the addition of 2 μM CuSO₄ triggered rapid and efficient disassembly of the SC and formation of polycomplexes within just 1 hour of Yen1^{ON} induction (new data in Extended Data Fig. 10o, p; Supplementary Video 14). This is considerably faster than the *P_{GAL1}* promoter used to drive Yen1^{ON} expression (e.g. see Supplementary Video 1).

Using this system, we observed that induction of Ndt80 expression in control pachytene cells (~8hrs in SPM) led to rapid loss of Zip1^{GFP} signal within ~60-90 min (Extended Data Fig. 10m, n; Supplementary Video 13). Subsequently, 84% of the cells underwent meiosis I and II and became bi- and tetra-nucleated, with two or four identically sized DNA masses respectively. By contrast, cells induced to express Yen1^{ON} 65 min prior to the induction of Ndt80 disassembled Zip1^{GFP} prematurely and remained with a large polycomplex for a prolonged period, up to 180-240 min after Ndt80 induction (Extended Data Fig. 10o-p; Supplementary Video 14). Nevertheless, most cells expressing Yen1^{ON} eventually lost all Zip1^{GFP} signal, including the polycomplex, likely because the accumulating Ndt80 “forced” exit from prophase I. Interestingly, these cells attempted to undergo meiosis I nuclear division, as assessed by HTB1^{mCherry} labelling of chromatin, but the vast majority failed, displaying either asymmetric chromosome distribution or DNA bridges (Extended Data Fig. 10p, q).

These data suggest that unscheduled dHJ processing during pachytene triggers SC disassembly, leading to de novo DSB formation and the accumulation of DSB repair intermediates. In turn, these DSBs and early repair intermediates activate the DNA damage response, delaying exit from prophase I. Cells that eventually progress beyond prophase I - likely due to Ndt80-mediated accumulation of M-phase promoting factors (e.g., M-phase cyclins) - exhibit clear defects in chromosome segregation.

In summary, this experiment complements very nicely our previous experiments and supports a model in which dHJ-mediated maintenance of chromosome synapsis is very important for timely suppression of DSB formation and for assurance that progression into meiosis I occurs in the presence of crossover precursors (dHJs) in all chromosomes. Failure to have dHJs results in the re-initiation of DSB formation and delayed meiotic progression.

These new data is now included in the revised manuscript, Extended Data Figure 10m-q. Text changes in lines 270-278 describe the findings.

2. It is important that the nuclease-dead Yen1 control protein binds to DNA joint molecules as well as Yen1-ON protein, particularly as the Yen1-ON protein (with nine Cdk consensus sites mutated to alanine) may have a higher binding affinity than the wild-type version of Yen1. The authors appear to have used the appropriate control protein in their experiments, as they call it the Yen1-ON-ND, and imply that it is the ON version of Yen1 (carrying the nine serine to alanine mutations) with catalytic mutations E193AE195A. However, it remains uncertain because the actual mutated residues are not listed in full and the authors instead refer to a prior study (ref 50), but that study seems to only use a Yen1-ND (i.e. a Yen1 protein with only the E193AE195A mutations). For full clarity the residues changed in each Yen1 protein should be explicitly stated in the main figure.

We thank the reviewer for pointing out the incorrect citation. In our experiment, we indeed used the nuclease-dead version of YEN1^{ON}, which carries the E193A and E195A mutations, as described in Blanco et al. 2014, in addition to the mutation of nine serines in CDK consensus sites to alanines. The nuclease-dead Yen1^{ON} was first used in meiotic cells in Arter et al. 2018, and we have therefore adjusted the citation in the manuscript in line 77. We have described the specific catalytic dead point mutations in detail in the Methods subsection 'Strain construction'.

3. The study's introduction presents ZMM proteins, but not the fundamental structural components of the SC central region (Zip1 and Ecm11-Gmc2). This seems odd given that SC is a major dimension of the study. It should be made clear to the reader early on that Zip1 is a fundamental component of both ZMM/recombination ensembles and SC.

The introduction is intentionally kept simple and general to engage readers outside the field and meiosis researchers who do not work with *S. cerevisiae*. We believe the manuscript provides sufficient context on SC architecture (lines 23–25) and describes key proteins and protein complexes involved in the SC structure of budding yeast, such as Zip1 (lines 65–66) and Ecm11-Gmc2 (lines 82–83). While Zip1 is mentioned as part of the ZMMs in line 38, we have not explicitly distinguished its dual roles as a ZMM and an SC structural component, as our focus remains on its function in the latter.

4. Lines 80-81: Citations should be provided: Humphryes et al., 2013, PLoS Genet. showed that Ecm11-Gmc2 is a component of the yeast SC central region, while Voelkel-Meiman et al., 2013, PLoS Genet. specifically demonstrated that Ecm11-Gmc2 is a component of the SC central element substructure.

We thank the reviewer for pointing out the missing citations and have added them to the manuscript. For consistency, we have added the citations for Zip1 (line 65), Smt3 (line 83) and Rec8 (line 87) where they were first described as part of the SC structure.

5. The narrative in lines 94-113 begins by posing the possibility that “downstream formation of nascent recombination intermediates could play a role in the continuous re-establishment of chromosome synapsis. To examine this possibility...” This statement implies the idea that SCs are continuously falling apart and reestablishing de novo (in a normal meiosis). What is the basis for suggesting this? I think it could be misleading. Voelkel-Meiman et al., 2012, PLoS Genet. showed that SCs continuously accumulate central region material and progressively grow more substantial after several hours of ndt80 arrest, which is not consistent with the idea that SCs continuously disassemble and then re-establish. Pollard et al., 2023, Front. Cell. Dev. Bio. analyzed SC assembly and found, for the most part, processive assembly (apart from a rare class of abortive events from small SC structures that were almost never seen in wild type). Furthermore, the experiment performed in this section does not address whether recombination intermediates play a role in “continuous re-establishment” of synapsis, but instead addresses whether SC maintenance relies on ongoing recombination intermediate formation. (The experiment involved Anchors Away-depletion of Rec104 and observation of diminished DSBs and halted accumulation of DNA joint molecules but no strong change in SC structures). The authors do make an appropriate rigorous conclusion at the end of the section (“de novo DSB formation, although initially required for SC assembly, is largely dispensable for SC maintenance”) but this conclusion addresses the question of SC maintenance, not “re-establishment” of SC on chromosomes.

We agree with the comment of the reviewer and have adapted the description of the experiment in lines 100-101. It now reads: “downstream formation of nascent recombination intermediates could play a role in maintaining the SC structure.”

6. Line 299: “recombination intermediates play a key role in the establishment and maintenance of chromosome synapsis, a process that most likely involves continuous polymerization of the SC”. The authors show that recombination intermediates stabilize SC structures, but their data do not address whether recombination intermediates promote the continuous polymerization of SC structures. The existence of a rudimentary or early SC structure might intrinsically promote its continuous polymerization, so in that sense maintenance of any kind of SC structure will indirectly promote its continuous polymerization. However, this sentence leaves the impression that somehow maintaining the rudimentary or early structure requires continuous polymerization of SC; to my knowledge there is no data that this is the case for budding yeast SC. When Voelkel-Meiman et al., 2012, PLoS Genet. looked for a “treadmilling” of SC components into and out of the structure, that study found only evidence for SC component addition, no evidence that SC components exited full-length SC structures to any measurable degree during prophase.

We acknowledge that our work does not directly demonstrate that sites with ZMM-stabilized dHJs continuously polymerize the SC structure. This is why we originally stated, “a process that most likely involves...”. However, we believe this scenario is likely, based on our experiments using Ulp1^{ΔN-FRB}, which show that reversible SC disassembly requires dHJs/ZMMs (Figure 3). Moreover, our data suggest that ZMM-stabilized dHJ sites are critical for SC maintenance, as conditional depletion of Zip4/ZZS and Sgs1 leads to complete SC disruption (Extended Data Figure 5o-q).

In response to the reviewer’s comment, we have revised the text to adopt a more cautious interpretation of our findings. In lines 326-327, it now reads: “a process that we postulate involves...”.

7. Line 124: “it is therefore likely that all ZMM proteins directly or indirectly require recombination intermediates for continued chromosome association during pachytene”. Because it is relevant to the mechanistic consideration of the data, either here or in the Introduction the narrative should briefly discuss what is known about the interdependencies of ZMM proteins (i.e. which of the ZMMs rely on which for their localization to recombination sites/chromosomes)?

In lines 124–128, we provide a concise overview of the ZMM proteins examined in our study. Given the already dense nature of the manuscript, an additional discussion of the recruitment hierarchy would require significant space, which we believe would be counterproductive. However, we trust that interested readers can easily access the relevant literature for further insights.

8. Lines 146-160 are confusing. The narrative begins by posing the question: “To determine if recombination intermediates contribute to SC maintenance independently of ZMMs...” But the section ends with the conclusion: “...these observations suggest that dHJs are required for the continuous association of ZMMs with chromosomes...ZMMs prevent dHJ dissolution by non-crossover pathways, while also promoting SC maintenance.” The experimental question and conclusion do not align with one another.

I think the issue is the original question requires an experiment that removed ZMMs (to look for whether recombination intermediates promote SC maintenance in the absence of ZMMs), but the data suggested ZZS proteins may still bind joint molecules and stabilize SC when Zip3/Sgs1 or Msh4/Sgs1 are co-depleted, thus the experimental approach was not capable of asking that original question. However, the experiment removing ZZS and co-depleting ZZS+Sgs1 does still address the question (and the findings of immediate SC disassembly in either situation indicate there is no role for recombination intermediates in stabilizing SC independent of ZMMs).

We agree with the reviewer that this could be confusing. To address this issue we have separated the concluding sentences into a new paragraph (lines 166-167), to make it clear that it is a general conclusion of the entire section of the manuscript.

9. The authors provide data (Fig. 4) on de novo DSB formation when SCs are depleted in *ndt80* arrested cells, and in the Discussion (line 300) say: “our work also suggest that SC maintenance contributes to the suppression of DSB formation on chromosomes that have already successfully formed ZMM-stabilized dHJs.” This statement should situate the present

data in context of earlier work indicating the same conclusion, particularly Subramanian et al., 2016, PLoS Biol., who reported that conditional depletion of Zip1 leads to accumulation of Hop1-pT318 and Mek1 and Mer2-dependent DSBs at several hotspots in *ndt80*-arrested cells.

We have discussed in detail what has been described regarding the SC formation-dependent down-regulation of DSB formation in lines 221-225, and have cited the main studies providing experimental data leading to this interpretation. This includes the study by Subramanian et al. (2016). We have cited this study both in the context of the model that SC formation suppresses the formation of new DSBs (lines 221-223) and that this involves the displacement of HORMAD proteins from the chromosome axes (lines 223-225). The discussion of the conclusions of these studies comes immediately before the description of our experimental findings on Hop1 re-localization to meiotic chromosomes and de novo DSB formation upon Yen1^{ON} expression. Therefore, we believe that we have embedded the findings of Subramanian et al. (2016) in the appropriate context in our manuscript.

In the Discussion (lines 299-301 in the original manuscript, now lines 327-329), we have nevertheless modified the sentence to better emphasise our findings. It now reads: “Our work also suggests that dHJ-ZMM-mediated SC maintenance contributes to the stable suppression of DSB formation on recombining chromosomes (Fig. 5h).”

10. The “control” nuclei in Fig. 2e,n do not look like they have “full synapsis”, even apart from the presence of polycomplex. Have the authors measured the cumulative lengths of SCs in *ndt80* vs. ZIP3-AID *ndt80* nuclei? Spore viabilities of the *zip3* null, *msh4* null, ZIP3-AID and MSH4-AID homozygotes should be provided (the legend says they are normal, but the actual numbers should be given; apologies if I missed it). This is particularly important since *zip3* null mutants actually display SC structures that initiate from centromeres. Also, polycomplex frequency should be plotted on the graphs as they were in earlier figures. Even if the AID-tagged ZMM proteins (Zip3-AID and MSH4-AID) are less than functional, the experiment still shows that these ZMMs play a role in SC stabilization; I do not believe the tagged proteins must be fully functional for this data to be conclusive. However, I do feel the study needs to be careful in presenting the fusion strains, so that the reader does not assume these proteins are necessarily fully functional. Also this is important for the reader to be able to fully interpret the findings – if the Zip3-AID or Msh4-AID is delayed in SC assembly, then SCs at the start of the depletion may be “weaker” (less material in the central region) than the SCs at the start of the Yen1-ON experiment, and easier to disturb upon ZMM removal. Again, I think the effect is clear and strong conclusions can be made about the role of ZMMs but the authors should be fully transparent in possible differences between the experimental contexts of Fig. 1 vs. Fig. 2.

We thank the reviewer for this valuable feedback. In the revised manuscript, we have now included a table summarizing the spore viability of all tagged constructs (Supplementary Table 2). For comparison, we have also added the corresponding spore viability of strains carrying deletions of the respective genes as positive controls for loss-of-function phenotypes. Notably, Zip3^{AID} and Msh4^{AID} exhibit wild-type levels of spore viability, while Zip4^{AID} shows a slight reduction (~ 95% spore viability; n = 216 spores). However, all tagged alleles differ significantly from the respective deletions, which result in very low spore viability. Additionally, as suggested by the reviewer, we have now incorporated plots displaying the frequency of nuclei with polycomplexes in Figure 2f,n and Extended Data Figures 4j,k and 5k,q.

11. Discussion in lines 311-321 proposes a model that is not aligned with existing mutant data. The authors suggest in this model that meiotic nuclei missing SC cannot proceed to the first meiotic division because of ongoing DSB. However, *ecm11* and *gmc2* and several *zip1* non-null mutants proceed through meiosis with high spore viability despite the absence of SC. The authors need to incorporate this important information into their model.

In lines 338-340, we stated that increased DNA damage signalling delays meiotic progression, not that it blocks it indefinitely, as the reviewer implies. As documented in the literature and further supported by Figure for Reviewers 1, *ecm11Δ gmc2Δ* mutants exhibit a significant prophase I delay due to ongoing DSB formation and repair. This is fully consistent with our model.

We respectfully disagree with the suggestion that mutant analyses should be incorporated into the discussion. Central element mutants are not SC-null (Humphryes et al. 2013; Figure for Reviewers 2), as the reviewer seems to suggest. Furthermore, as outlined in point (e), the absence of Ecm11/Gmc2 is different from a defect in assembling Ecm11/Gmc2 into the SC while the proteins remain present in the nucleus. *ecm11Δ gmc2Δ* mutants may not arrest meiotic progression indefinitely, potentially because they can still load Zip1 or because Ecm11/Gmc2 play a role in the signalling mechanism that delays meiotic progression — among several other possible explanations.

12. Line 325: “For these reasons, we propose that dHJs are more than passive intermediate structures that form in the process of crossing over. They have a central regulatory function in coordinating meiotic progression with crossover assurance” This claim seems overblown. The DNA joint molecule is important for triggering SC assembly and for maintaining the structure, but the data do not point to a “regulatory” function for dHJs such that these joint molecules would be able to monitor SC lengths and up/down regulate them. Moreover, the evidence that SC structures themselves play a critical role in meiotic progression is slim, thus it does not seem likely that dHJs promote meiotic progression through their capacity to maintain SC structures. The fact that disassembled SC could allow chromosomes another chance at forming an interhomolog recombination event may well contribute to crossover assurance, but this mechanism seems more about the presence/absence of SC not whether dHJs actively regulate them.

As in other points above, the reviewer has not considered in the comments the literature demonstrating that the SC contributes to the downregulation of DSB formation (for example Thacker et al. 2014; Subramanian et al. 2016; Mu et al. 2020; Lee et al. 2021). We hope that our new experiments with *Rad51^{iyEGFP}* will help further clarify this point (Figure for Reviewers 1). We also hope that the new data from the experiment suggested by the reviewer (point f, above; Extended Data Fig. 10m-q), will help her/him appreciate how important for meiotic progression it is to have dHJs.

We stand by our original discussion point.

Minor Comments

13. The Abstract states that joint molecules pose “a risk to chromosome segregation”. Is there

evidence that DNA joint molecules pose more of a risk to chromosome segregation than unrecombined chromosomes or bivalents with a crossover? Evidence for this claim should be explained in the Introduction if it is to stay in the Abstract.

We do not intend to compare the risks of interlinked versus non-recombined homologs in meiosis. Rather, we want to point out that proper homolog segregation in meiosis I paradoxically depends on the covalent linkage of homologs through the formation of dHJs. Thus, in lines 3-5 we wrote: “Despite posing a risk to chromosome segregation, HJs accumulate during meiotic prophase I as intermediates in the process of crossing-over.”.

14. The Introduction ends by stating the dHJ-ZMM interplay coordinates meiotic progression with crossover assurance. If the authors keep this claim, crossover assurance should be defined here as it is a relatively obscure term.

Unfortunately, we are not sure that we understand the precise concern raised by the reviewer. The last sentence of the introduction is a “teaser” for what is coming in the subsequent parts of the manuscript. In our opinion, the preceding paragraphs - which are limited in space - prepare the reader to grasp the key message.

15. It is interesting that dual removal of Zip3 and Sgs1 or of Msh4 and Sgs1 leads to a delay in SC disassembly, whereas removal of ZZS leads to rapid SC disassembly regardless of whether Sgs1 is co-depleted. The authors propose the delay in SC disassembly observed in Zip3+Sgs1 co-depletion is due to ZZS complex associating with the recombination intermediate (and protecting it from Sgs1). Have the authors tried to look for ZZS on chromosomes in the doubly-depleted Zip3 Sgs1 cells?

We have attempted to analyse the localization of the ZZS complex on meiotic chromosomes of Zip3^{AID} and SGS1^{AID}-depleted cells using published antibodies against Zip4 and Zip2 (Chua et al. 1998; Shinohara et al. 2008). Unfortunately, however, in our hands neither antibody (the batches that were kindly provided to us) produced specific stainings when compared to chromosome spreads from *zip4*Δ or *zip2*Δ cells, respectively.

16. Fig. 3: The authors show that SCs can assemble de novo at late pachytene. It would be appropriate to cite prior work: Fig. S2 of Voelkel-Meiman et al., 2012, PLoS Genet. showed that *zip1*-depleted cells brought to mid-late prophase (20 hr or 26 hr *ndt80* arrest; aligned condensed chromosomes with substantial Red1 accumulated on axes but no SC) can undergo substantial SC assembly with frequent long SC stretches. This work did not test whether dHJs nor de novo DSBs are required but should be cited as earlier evidence that SC establishment on unsynapsed chromosomes at late prophase is possible.

We thank the reviewer for bringing this published data to our attention. In Voelkel-Meiman et al., 2012, the authors use an inducible *P_{GAL1}-ZIP1* allele, which mimics a *zip1*Δ mutant, and combine it with *ndt80*Δ. In such a strain, initial DSB formation/repair occur in the absence of Zip1, resulting in severely delayed and reduced formation of dHJs (Storzazzi et al. 1996; Börner et al. 2004). Eventual expression of Zip1 from the *GAL1* promoter would be expected to stabilise the formation of recombination intermediates, which then contributes to Zip1 assembly on the paired homologous chromosomes and to the suppression of new cycles of DSB formation. By contrast, in our experiments, we induce SC disassembly in pachytene

arrested *ndt80*Δ cells that have already undergone recombination and formed ZMM-stabilized dHJs.

Despite these key differences we agree that both experimental approaches demonstrate the capacity of paired homologs to polymerize the SC structure. We have therefore added an introductory sentence to the paragraph starting in line 172 to refer to these previous observations. It reads “In budding yeast, paired chromosome axes retain the ability to assemble the SC structure even in late prophase I (Voelkel-Meiman et al, 2012).”.

17. Line 159: “in turn, ZMMs prevent dHJ dissolution by non-crossover pathways, while also promoting SC maintenance.” Do we know, at this late stage when the depletion is happening, that Sgs1 is promoting dissolution by a noncrossover pathway? Ext. Data Fig. 5c shows a reduction in crossovers at each timepoint in addition to a reduction in non-crossovers. Why then say Sgs1 is promoting the non-crossover path?

Before answering the specific comment posed by the reviewer, we will address a general aspect of the physical analysis of recombination at *HIS4::LEU2*, which has led to well justified comments from the 3 reviewers. This issue pertains to the detection of a crossover diagnostic fragment in *ndt80*Δ mutants, which is well documented in the literature, but not well understood at the mechanistic level. We provide a short summary of what is known about these crossovers.

The *HIS4::LEU2* system has been developed by Nancy Kleckner and colleagues and used in dozens of publications to uncover fundamental aspects of meiotic recombination. Using a similar system, the lab of Michael Lichten has first reported that while the vast majority of crossovers - in particular type I - are Ndt80-dependent, there is a small subset that is not (Allers and Lichten, 2001). Similar findings were described in many other publications using the *HIS4::LEU2* system (e.g. Zakharyevich et al., 2012; Arter et al., 2018). Later on, the Lichten lab showed that Sgs1 is important for the accumulation of both crossovers and non-crossovers in *ndt80*Δ mutants (De Muyt et al., 2012) and the Hunter lab went on to show that deletion of all HJ resolvases (quadruple resolvase mutant) leads to a near complete elimination of crossovers, with exception of a small subset (Zakharyevich et al., 2012). They also found in the same publication that virtually all crossovers are eliminated if *SGS1* is additionally deleted (quintuple mutant). Even though not extensively discussed in Zakharyevich et al., 2012, which had a different focus, these data indicate that there is a small subset of Ndt80-independent crossovers that is HJ resolvase-independent, ZMM-independent, but Sgs1-dependent. Our data is fully consistent with these previous findings.

Going back to the points raised by the reviewer: “in turn, ZMMs prevent dHJ dissolution by non-crossover pathways, while also promoting SC maintenance.” Do we know, at this late stage when the depletion is happening, that Sgs1 is promoting dissolution by a noncrossover pathway?” Yes. When ZMMs are conditionally depleted, the processing of DNA joint molecules is accompanied by a specific increase in non-crossovers (Figure 2g,h, for Zip3 depletion; new data in Extended Data Figure 5b,c for Msh4 depletion). Therefore, the ZMM-stabilised JMs, which were crossover-designated, are preferentially converted into non-crossovers. The generation of non-crossovers is clearly mediated by Sgs1, as seen in Figure 2k,l and Extended Data Figure 5e, f.

We do however agree with the reviewer that depletion of Sgs1 also reduces slightly the accumulation of the Ndt80-independent crossovers. This reduction at late stages is most likely

explained by *ndt80Δ* mutants forming some level of DSBs, even in late prophase. These DSBs (clearly detected in the Southern blots) give rise to both Sgs1-dependent non-crossovers and, a minor subset of, Sgs1-dependent crossovers. In the revised manuscript we now have included data for the control Sgs1 depletion (new data in Extended Data Figure 5g-i), in which it can be appreciated that – in the absence of Zip3/Msh4 depletion - the accumulation of both non-crossovers and crossovers (minor amount) is Sgs1-dependent. We also add a note in the text (lines 146-150), stating that: “We note that depletion of Sgs1 also led to a small reduction in the accumulation of crossovers in *ndt80Δ* mutants (Extended Data Fig. 5g-i). This observation is consistent with previous work showing that *ndt80Δ* mutants accumulate a small proportion of Sgs1-dependent crossovers (Allers and Lichten 2001, de Muyt et al. 2012, Zakharyevich et al, 2012). The precise origin of these crossovers remains unknown.”

18. The section titled: “dHJs enable reversible SC disassembly” is challenging to wrap one’s head around; why not say “dHJs enable SC re-assembly”?

We appreciate the reviewer’s suggestion and understand the importance of a clear and intuitive title. However, we believe that “dHJs enable reversible SC disassembly” more accurately reflects our findings and the biological process we aim to describe. While it may take the reader a bit longer to fully grasp the implications, this phrasing conveys more precise information and better captures the dynamic nature of the dHJ-SC interplay compared to “dHJs enable SC re-assembly.”

19. Fig. 1i: Clarify: why are there CO products at 7/8 hr, prior to inducing Yen1-ON?

See point 17.

20. Fig. 1j: Clarify: for the quantification of joint molecules in +rapamycin/minus rapamycin conditions, were these from the cells with or without Yen1-ON exposure?

The quantification of DNA joint molecules in Figure 1j is from the culture that was not treated with β -estradiol to induce Yen1^{ON} expression. Based on the comments of the reviewer and of reviewer 2, we have improved the presentation of the data in Figure 1h-m. Specifically, we improved the scheme depicting all the different treatments (Figure 1h) and introduced a new colour code representing the different treatments, which was implemented in all subsequent panels and data associated with this experiment (Figure 1i-m; Extended Data Figure 3). In addition, the graph legends in Figure 1j, l and Extended Data Figure 3b, e now include the description of all treatments.

21. Fig. 1k says the graph plots Zip1 distribution/assembled SC as in the experiment in “j”, which refers back to “h”. This is confusing; if these nuclei are from the experiment described in h, then we expect to see fewer SCs after 8 hr in both DMSO- and rapamycin-treated cells (because of YEN1-ON induction) (like in “m”). Perhaps there should be an area above these graphs in j, k, l that indicates both dimensions of the experiment: the rapamycin exposure (or not) and the estradiol exposure (or not) (this is done to some extent in m already).

As outlined in point 20, we have changed the presentation of the data in Figure 1h-m and adapted the figure legends to make the interpretation more intuitive.

22. Error bars are missing on plots in Ext. Data Figs 1, 3, 11, and in main Fig. 2h,m (but present in Figs 1 and 3e).

In the figure legends we detail whether the plotted data comes from multiple replicates of the experiment, or, in some cases, from one representative example of several experiments. In the figures highlighted by the reviewer, the data in the plots is from one representative experiment. The reason why, in a few cases, we display a representative experiment is that the kinetics of meiotic induction is subject to day-to-day variation, which we always assess by FACS analysis of DNA replication. When we observe variation, we do not plot data together as it would introduce noise. Importantly, all key findings were reproduced multiple times in independent experiments and, in many cases, using orthogonal approaches.

23. Ext. Data Fig. 10a: Label the FACS analysis to indicate when zygotene entry is thought to be (at the 4 hr row). For clarity the authors should also indicate on the image as well as in the legend in e and h, exactly when B estradiol was applied relative to the 0 hr timepoint in (a) (indicate for all the hour after introduction into sporulation media so that they can all be cross-compared easily).

We perform FACS analysis of DNA content to ensure that the kinetics of meiotic entry and progression is comparable across experiments. It also provides future readers of the manuscript – in particular those that perform meiotic synchronization in different ways, or strains – a point of reference for comparison. Since the synchrony of the cultures is limited, we do not think that it would be helpful to indicate entry into the zygotene stage.

We have adapted the figure legends of Figure 4i, j and Extended Data Figure 10e-l to indicate the time of β -estradiol addition relative to the time in sporulation medium.

24. A note about data availability: The hundreds of images for each timepoint that were generated for the study are not included in the article (nor do I think they should be). But the statement probably should be modified so that it does not suggest absolutely all the data reported is in the manuscript itself.

We agree with and thank the reviewer for pointing this out. We have modified the data availability statement to avoid confusion about what is considered relevant data. It now reads: “All relevant data supporting the findings of this study are available from the corresponding author upon reasonable request. Biological materials used in this study may be obtained from the corresponding author.”

Referee #2:

Henggeler et al. report about the interplay between mature meiotic recombination intermediates (double Holliday junctions, dHJs) and formation and maintenance of the synaptonemal complex (SC). Using a wealth of sophisticated, elegant and precise molecular biology tools, they revealed that dHJ are essential for the nucleation and maintenance of the SC through the protective action of ZMM proteins. Hence both normal and abnormal disruption of dHJs leads to SC disappearance and associated lack of chromosome synapsis without compromising the axis loop organization of chromosomes. Overall, they postulate a regulatory role of dHJs leading to crossover assurance, which is the “goal” of meiosis to ensure proper homolog segregation.

This is an excellent manuscript that provides a better understanding of the interplay between dHJ and SC, and ultimately about the essence of meiotic recombination which is crossover formation. The experiments are well introduced, well performed and conclusive. A list of mostly minor comments is provided below as they appear along the text.

We thank the reviewer for the positive and encouraging feedback on our work and for acknowledging both the sophisticated methods used and the quality of our data. We have attempted to address the remaining comments below.

This reviewer feels that these results being directly connected to the formation of the class I interfering crossovers, the manuscript would benefit from at least some integration of these findings in light of crossover interference. It looks like the proposed model here for crossover formation and assurance could be self-sufficient without additional layers of regulation while the corresponding crossovers show interference. How are these different features integrated? Does this postulate that crossover assurance is independent from crossover interference?

We appreciate the insightful comments from the reviewer and agree that the relationship between crossover interference and our findings is a fascinating topic. However, given the ongoing - and unresolved - debate about the mechanisms underlying crossover interference, we would prefer to refrain from discussing it at this stage. While we recognize the importance of this question for the field, we believe it extends beyond the scope of our manuscript, which focuses on the mechanism of crossover assurance. We hope the reviewer understands our decision – given the space limitation and how dense the manuscript already is – to keep the discussion centred on the core aspects of our study.

- Fig. 1: The presentation is a bit misleading between panels i, j and l. It looks like these last two panels correspond to quantification of gels that are not exactly the one presented in panel i. Providing quantification of the gel shown would look relevant here.

Based on the comments of the reviewer and of reviewer 1, we have improved the presentation of the data in Figure 1h-m and Extended Data Figure 3. The quantification in the graphs in Figure 1j and l is based on the Southern blot shown in Figure 1i as well as a biological replicate and shows the mean and range of these two experiments, as indicated in the legend of Figure 1j and l. However, the gel source data of the second replicate was not provided in the original manuscript, which led to the confusion. In the revised manuscript we have now included the gel source data from the second experiment in the Extended Data Figure 3c. For consistency,

we have also included the biological replicate of the experiments in Extended Data Figure 6e and Extended Data Figure 9e, f.

- Lines 92-93: The experimental system used compellingly shows that dHJ are resolved without severely disrupting the axis-loop organization. However, this is expected since chromosomes segregate under their compacted form at anaphase I after resolution of recombination intermediates. Overall, it would be good that the authors distinguish between what is expected, based on what is already known, from what is brand new from the present work.

We agree with the reviewer and have revised the paragraph discussing our observations on axis-loop organization upon Yen1^{ON} expression (lines 82–97). We now place our data in the context of previous studies on chromosome axis regulation and cohesin release (lines 85–90), in order to better highlight the new aspects of our findings.

- Ext. Data Fig. 4m and lines 131-132: How to explain the stronger phenotype of the mutants of the ZMM genes compared to YEN1-ON that shows a milder Rec8 disorganization phenotype?

Our data show that depletion of Zip3^{AID} or Msh4^{AID} leads to a rapid loss of DNA joint molecules, within one hour of 5-Ph-IAA addition to induce protein degradation (Figure 2g,h for Zip3^{AID}; new data in Extended Data Figure 5b,c for Msh4^{AID}). Concurrent with the loss of DNA joint molecules, the SC is rapidly disassembled, with most nuclei showing no chromosomal Zip1 staining (Figure 2e,f for Zip3^{AID}; Extended Data Figure 4h, j for Msh4^{AID}). By comparison, SC disassembly upon Yen1^{ON} expression is slower (Figure 1b, c; see also Supplementary Videos 1 and 3). As discussed in response to Reviewer 1, this is because Yen1^{ON} protein levels induced from the *GAL1* promoter take time to accumulate (See response to point (a) from Reviewer 1; Extended Data Figure 1b,c; new data in Extended Data Figure 10c,d), and loss of chromosome synapsis inversely correlates with Yen1^{ON} protein levels (Figure 1d). Furthermore, in this case, dHJ resolution occurs prior to ZMM dissociation, which is likely to explain why it is slower to have an effect in disrupting SC maintenance. The appearance of more disorganised Rec8 immunostaining patterns correlates with loss of both DNA joint molecules and SC, thus the milder phenotype observed with Yen1^{ON} expression by comparison with acute ZMM protein degradation.

- Fig. 2 and Ext. Data Fig. 5: In Fig. 2i, how to explain that crossovers accumulation is not affected by the depletion of Zip3? One argument could be the persistence of Zip3 on recombination intermediates that ensures the crossover-biased resolution. However, the disappearance of JMs shows that they are rapidly processed after Zip3 depletion, which invalidates the previous argument. Hence, the doubling in crossovers from panel i is independent of Zip3, hence ZMMs.

We would like to point out that the conditional ZMM degradation experiments were performed in a *ndt80Δ* mutant background. *ndt80Δ* mutants arrest meiotic progression with synapsed homologs and unresolved dHJs (Xu et al. 1995; Allers and Lichten, 2001), because nucleases that could lead to dHJ resolution are not yet active. As discussed in response to minor comment 17 from Reviewer 1, the origin of basal crossover fragment 2 (CO2) in pachytene

arrested *ndt80Δ* is currently unknown. Nevertheless, from previous work (Allers and Lichten 2001, de Muyt et al. 2012, Zakharyevich et al, 2012) this small subset of crossovers is resolvase-independent but STR-dependent, which is consistent with our findings.

- Ext. Data Fig. 5a-c: There is a clear drop in CO formation in ZIP3-AID SGS-AID, which is not seen in ZIP3-AID alone. Does it mean that some COs (half of the final amount) result from JMs processing by Sgs1, potentially in combination with structure specific nucleases?

Please see response to minor comment 17 from Reviewer 1.

- Fig. 4: It might be relevant to insist on the fact that crossovers are made after YEN1-ON induction. These crossovers represent physical connections between homologs which, in combination with cohesins that maintain sister chromatids together, should be suitable for accurate segregation, as supported by video 11 in an otherwise WT context. Therefore, in the absence of *ndt80* after YEN1-ON induction, homologous chromosomes are more than engaged together, they show reciprocal exchanges. Hence, the fact that DSBs are formed again after YEN1-ON expression shows that dHJs and ZMM proteins ensure the DSB-repressive state, and this repressive state is independent of the homolog engagement per se as suggested by Thacker et al. (2014).

We agree with the reviewer.

- Lines 260-261: The authors postulate that the cell cycle progression is likely due to “de novo DSB formation and ensuing DNA damage signaling”. Early expression of YEN1-ON likely resolves early recombination intermediates upstream of dHJ, leaving unrepaired the corresponding DSBs. These unrepaired DSBs should also significantly contribute to the disruption of the cell cycle progression. Overall, it may be wise to include early in the manuscript what is the consequence of expression of YEN1-ON i.e. unspecific resolution of any branched structure, including bona fide dHJ as well as early recombination intermediates, leading to crossover/noncrossover products and unrepaired DSBs, respectively.

We agree with the reviewer and have modified the first sentence of the results in lines 57-60 accordingly: “To assess whether recombination intermediates are required for the maintenance of chromosome synapsis, we used *ndt80Δ* mutants in combination with conditional expression of an engineered structure-selective endonuclease, Yen1^{ON}, which can efficiently process a broad range of branched recombination intermediates”.

- Chapter about Cdc5 starting line 265: The authors used the *mlh3Δ mms4mn slx1Δ yen1Δ* quadruple mutant as a “complete” resolvase mutant. As reported at least by the Hunter group (Zakharyevich et al., 2012), this mutant still shows crossover formation that is Sgs1-dependent. Therefore, the fact that the SC eventually disassembles in this quadruple mutant, although it is delayed, could indicate that dHJs are eventually resolved in an Sgs1-dependent manner without involving a second Cdc5 regulated pathway involving phosphorylation of SC components. Performing the experiment in such a quintuple background will solve this issue. Without such an experiment, it is impossible to conclude about the two pathways for SC disassembly proposed by the authors (see also lines 339-342).

We agree with the suggestion of the reviewer and have now performed the experiments with the quintuple mutant. This analysis was particularly challenging due to the complex genotype required. In the *sgs1^{mn} mms4^{mn} slx4 Δ yen1 Δ mlh3 Δ* mutant background, we observed a significant delay in Cdc5-mediated SC disassembly compared to the control, and a moderate delay relative to the quadruple resolvase mutant (new data in revised Extended Data Figure 11d–j). However, SC disassembly was still completed in more than 50% of nuclei within the experimental timeframe. Furthermore, we have now performed live-cell imaging to monitor SC disassembly during the prophase I-to-metaphase I transition in *NDT80* wild type cells (new data in Figure 5f), to complement our analysis of chromosome spreads (Extended Data Figure 11k). Consistently, we observed a delay in both the quadruple and quintuple mutants compared to the control, with the quintuple mutant exhibiting a moderate delay relative to the quadruple mutant. However, SC disassembly was still completed in all cells analyzed.

We note that the delay in SC disassembly in *NDT80* cells was significantly smaller than what we observed in the Cdc5-induction experiments (*ndt80 Δ*). This is probably due to Cdc5 expression being sufficient to trigger SC disassembly in *ndt80 Δ* cells (Sourirajan and Lichten, 2008), but not with the same efficiency as it does so in the presence of other *NDT80*-dependent factors, such as Clb1-CDK, which is known to stimulate Cdc5 function.

- Fig. 5j: Note that based on the configuration of the heteroduplex DNA tracts, the dHJ resolution drawn is not seen in vivo, as reported by Marsolier-Kergoat et al. (2018).

We thank the reviewer for pointing out the error in the scheme in Fig. 5g. We have adapted the scheme according to what was reported in Marsolier-Kergoat et al. 2018 and Ahuja et al. 2021.

Typos:

- Ext. Data Fig. 3b y-axis: rec104 instead of Rec10.

- Ext. Data Fig. 2f: There are two panels +B-estradiol @10h. Is the right panel 10 or 12h?

We thank the reviewer for pointing out the typos. We have corrected them in the revised manuscript.

Referee #3:

This is an outstanding paper that will contribute enormously to multiple parts of our understanding of the meiotic process. The paper is comprised of beautiful molecular genetics and cytology. Most importantly, it stands as a 'complete study'. I am at a loss to suggest what other experiments might be done. It is a masterpiece in so many ways.

The most fundamental and unique feature of the meiotic process is a highly programmed series of interactions between homologues that collectively set the stage for their segregation at MI. Recombination-mediated processes occur in physical and functional linkage with meiotic axial chromosome structure, with interplay in both directions, before, during, and after formation and dissolution of the synaptonemal complex (SC), a highly conserved meiosis-specific structure that links homolog axes along their lengths. This program takes hours in budding yeast, but days or weeks in some organisms. Since DNA recombination itself can be completed in minutes, this dramatic prolongation of meiotic prophase is likely to reflect the complexity of whole chromosome events, especially homolog pairing. These considerations raise a fundamental question: How are the DNA events of recombination and whole chromosome processes locally coordinated in time and space?

The SC central components are required for reorganization of the recombination complexes (Rad51, Mer3, and Msh4) from an on-axis position to a between-axis (thus, on the SC central region) position concomitant with SC installation. Thus, whereas in most organisms, DSB-initiated recombinational interactions directly mediate both homology searching and homolog coalignment, the SC is required, through its central components, for the maintenance and/or turnover of the recombination proteins required for maturation of the DSBs into crossovers.

In this article, the authors use molecular genetic, cytological (classical and high resolution) and live imaging approaches to investigate the role of Holliday junctions (HJs) in SC stability. They exploit all the strengths of the *S. cerevisiae* model system to develop an ingenious experimental system that allows them to induce nucleolytic dissolution of HJs after homolog synapsis and analyze the effect on SC stability monitored by Zip1, EcM11-Gmc2 and Smt3/SUMO.

The authors were able to show that dissolution of the HJs led to the disappearance of these three proteins between homologs and the formation of polycomplexes, without causing severe changes in the chromosome axis (Rec8). This dissolution of the SC is accompanied by a delocalisation of proteins described as binding to HJs and necessary for the formation of COs and the initiation of the SC (Zip3, Msh5 and Zip4). The authors then rigorously demonstrated that conditional depletion of these three proteins in pachytene also leads to the disassembly of the SC (marked by loss of Zip1 from the chromosome and disorganization of Rec8 pattern), associated with an increase in NCOs and a decrease in HJ. This result shows that these proteins are not only required for the formation of CO-recombination intermediates, but also for their stabilization and protection against dissolution by the Sgs1-Top3-Rmi1 (STR) complex. Finally, the authors showed that dissolution of the SC, while maintaining the presence of the HJs, allows reassembly of the SC without the formation of new DSBs. The proposed model is based on these results and gives a central regulatory function in coordinating meiotic progression with crossover assurance.

The paper is well written, with detailed explanations and appropriate conclusions. The article is dense, with a lot of data, but the rigorous approach and the logical presentation of the results make it easy to read. The data is well analyzed and presented. Sample sizes, error bars and P-values are given. The statistical tests used are well chosen. I look forward to seeing it published.

We would like to thank the reviewer for the very encouraging feedback. We have attempted to address all open minor comments.

Minor comments

- Lines 139-145: Zip3 depletion increases NCOs, but COs are unchanged. In *S. cerevisiae*, Zip3 is a marker for sites that are destined to become COs. What could explain why the number of COs does not vary, whereas it decreases when ZIP3 and SGS1 are removed?

We would kindly ask the reviewer to see the response to minor comment 17 from Reviewer 1.

In addition, it seems to me that the JM decrease is much greater than the increase in NCOs (Fig. 2h,i). Similarly, when ZIP3 and SGS1 are removed, there is an increase in JMs and a decrease in COs and NCOs (Ext. Data Fig. 5c). This increase appears to be greater than the decrease in COs and NCOs. Can this difference be explained by the distribution of new DSBs?

Regarding the quantitative discrepancy between the loss of DNA joint molecules and the increase in non-crossovers in Zip3^{AID}-depleted cells (Figure 2h), we would like to elaborate on the crossover/non-crossover assay used in these experiments. Crossover and non-crossover products are monitored at the *HIS4::LEU2* recombination hotspot by means of a BamHI/NgoMIV restriction site polymorphism located directly at the DSB site. A double digest with XhoI and NgoMIV allows the monitoring of gene conversion events proximal to the DSB site, as well as four types of crossover products (Figure for Reviewers 3, below). Depending on DSB formation on either the BamHI or NgoMIV parental homolog, four different diagnostic fragments for crossovers can be assessed (CO1-4) and two specific fragments for non-crossovers (NCO1-2). However, NCO1 is the only fragment that is unambiguously diagnostic of a non-crossover product, as NCO2 is the same size as CO3. Thus, NCO1 and CO2 can be compared to assess changes in the overall crossover and non-crossover ratio. In Figure 2g and k, only the NCO1 and CO2 fragments are shown and quantified. Since this assay only allows detection of a representative subset of the total recombinants at *HIS4::LEU2*, quantitative comparisons are not possible. To improve clarity, we have adapted the Southern blot labels in Figure 2g,k and Extended Data Figure 5b,e,h to allow the reader to distinguish the origin of the diagnostic fragments.

Figure for Reviewers 3. *XhoI*/*NgoMIV* double digest assay for crossover and non-crossover events at the *HIS4::LEU2* recombination hotspot. a, Map of the *HIS4::LEU2* recombination hotspot showing the DSB site and the location of Probe A used in Southern analysis. *XhoI* restriction sites (grey dashed lines) and the *BamHI*/*NgoMIV* restriction site polymorphism (black dashed line) are indicated, along with the corresponding diagnostic fragments and their sizes. Notably, NCO1 (in bold) is the only band unambiguously diagnostic of a non-crossover product, representing a subset of total recombinants. Adapted from Owens et al., 2018. **b,** Southern blot of genomic DNA digested with *XhoI* and *NgoMIV* from the experiment in Figure 2g, showing all detectable diagnostic fragments.

- Line 150 – “Similar results were obtained by combining conditional depletion of Msh4AID and Sgs1AID (Extended Data Fig. 5d,e).” What about JM, CO and NCO in the absence of Msh4 and MSH4 SGS1?

On the suggestion of the reviewer, we have extended the analyses to Msh4^{AID}, alone or in combination with Sgs1^{AID}. These data, in the revised Extended Data Figure 5a-f, confirmed our previous observations with Zip3^{AID} depletion, showing that Msh4^{AID} depletion also leads to a rapid loss of DNA joining molecules. Consistent with observations made with Zip3^{AID}, we also observed an increase in non-crossovers (NCO1) upon Msh4^{AID} depletion, while crossover levels were similar to the control (Extended Data Figure 5b,c). Simultaneous depletion of Msh4^{AID} and Sgs1^{AID} resulted in the stabilization and accumulation of DNA joint molecules (Extended Data Figure 5e,f).

Note 1: Author responses are highlighted in blue.

Note 2: Figures for Reviewers 1–3, containing data not included in the revised manuscript, are embedded in the previous rebuttal letter.

Referees' comments:

Referee #1:

I remain positive about the experiments in this study and the impact they have on considering how resolution of crossover intermediates might coordinate mechanistically with other cellular processes at the end of meiotic prophase. I remain concerned about the lack of rigor in the Title statement and in similar statements made within the manuscript (such as the last sentence of the introduction), where a claim specifically about the maintenance of synaptonemal complex is made.

OE of Yen1 (as the authors show) leads to at least two things: i) a dissolution of the mature, tripartite synaptonemal complex (SC) and ii) a dissolution of ZMM (recombination) structures. The functional consequence of OE of Yen1 and concomitant induction of Ndt80 (Ext Data Fig. 10) is severe: meiosis I and II divisions fail to complete, and chromosome bridges are observed. However, the conclusion from this functional experiment should be that loss of EITHER ZMM (recombinosome) structures OR SC (or both) may cause the meiotic catastrophe observed. Instead, the title states that maintenance of synapsis enables crossover assurance. This is a misleading statement, as the experimental data do not specifically implicate the SC in the severe meiotic failure observed when HJs are prematurely resolved (Yen1-OE). Furthermore (as I pointed out in my prior review), one would expect SC-deficient mutants to show this sort of meiotic catastrophe, but SC-deficient mutants do not show this phenotype.

I agree with the authors that some SC-deficient mutants (i.e. *ecm11* or *gmc2* null mutants) have increased DSBs and DNA repair processes and that these mutants also show a prophase delay (although I note that the *zip1*[Δ 21-163] separation-of-function mutant lacks SC structures but does not exhibit the prophase delay). I also agree with the authors that a deficiency in chromosome synapsis (deriving from the untimely processing of recombination intermediates) results in ongoing formation of DSBs and DSB repair intermediates, which results in increased DNA damage signalling and triggers the recombination checkpoint, causing a prolonged prophase I... There is good evidence for this model from prior studies, as the authors point out. The issue with the assertion made in the current manuscript Title, that maintenance of the SC "enables crossover assurance", is that the claim is not rigorously supported by the data provided in the manuscript (alternative explanations exist for the severe meiotic division defect observed when Yen1 is overexpressed) and does not align with the fact that SC-deficient mutants (*ecm11* or *gmc2*) exhibit robust (more than a wild type number of) crossovers and do NOT show meiosis I division failure.

Another way to put my concern: It is formally possible that the experiment in Ext. Data Fig. 10m-p (overexpression of Yen1 and release from *ndt80* block) - if carried out in an *ecm11* or *gmc2* mutant (which lacks SC in the first place) - would show the same severe defect in meiotic progression and anaphase bridges. This outcome would indicate that the meiotic catastrophe observed upon Yen1-OE is not due to failure to maintain a mature SC structure, but instead due to failure in something else – that something else could be a failure to maintain ZMM/recombinosome ensembles and/or a failure to properly coordinate resolution with cohesin/condensin remodeling at the exchange site.

While catastrophic chromosome segregation/meiotic failure is not the phenotype of an *ecm11* or *gmc2* SC-deficient mutant, the authors point out (line 344) that a meiotic cell missing HJ resolvases does show this severe meiotic failure. This raises the third possibility that the

meiotic failure observed in Ext. Data Fig. 10m-p is due to a few recalcitrant HJ junctions that fail to resolve properly when Yen1 is overexpressed.

Because of alternative explanations for the meiotic failure observed when Yen1 is overexpressed (that some HJs are not resolved or that ZMM dissolution at the developing chiasmata is the cause instead of absence of SC), in conjunction with the knowledge that SC structures are dispensable for meiotic success and robust crossover recombination, it is misleading and unrigorous to claim that the maintenance of SYNAPSIS (the SC) is what is functionally important for crossover assurance.

Finally, I wish to respond to the following comment the authors made in their response: [The reviewer states that “ecm11 and gmc2 mutants lack any SC structure,”. Taken literally, this statement is in direct disagreement with the literature (Humphryes et al. 2013) and with the data in Figure for Reviewers 2e-g, in which it is clear that Zip1 can still associate with the chromosomal axes in ecm11 gmc2 mutants.]

With respect, I completely disagree with this assertion. Zip1 accumulation on chromosome axes is not equivalent to the synaptonemal complex, just as tubulin accumulation on chromosomes is not equivalent to a spindle. In fact, authors from the Humphryes et al. (2013) study collaborated with authors from the Voelkel-Meiman (2013) study to show (with superresolution microscopy) that the Zip1 accumulating on chromosome axes in the ecm11 mutant fails to arrange into a tripartite synaptonemal complex structure. Zip1 accumulates on axes in the absence of Ecm11 or Gmc2, just as Ecm11 protein can accumulate on chromosomes in certain situations when Zip1 is altered. To say that accumulation of any SC structural component on a chromosome axis is equivalent to synaptonemal complex formation is dismissive of the grand SC structure itself (and certainly not rigorous).

Response from the authors:

We thank the reviewer for the continued engagement with our work and for the constructive feedback provided. We are grateful for the recognition of its impact and for the thoughtful insights, particularly regarding how our conclusions are framed in relation to the broader meiosis literature. We now address the remaining concern regarding the assertion that *maintenance of chromosome synapsis enables crossover assurance*, which is reflected in the original *Title* and the final sentence of the *Introduction*.

i. On the interpretation of the Yen1-ON phenotype and the role of SC maintenance

We fully agree with the reviewer that the phenotypes observed upon premature resolution of Holliday junctions (via Yen1-ON expression) are the result of multiple, interconnected consequences, namely: loss of ZMM proteins from chromosomes, SC disassembly, and checkpoint activation due to the formation of new DSBs. As the reviewer rightly notes, our data do not isolate the SC structure as the *sole* functional determinant of the catastrophic meiotic outcome. We have tried to reflect this complexity throughout the manuscript and emphasize that our interpretation is grounded in the **interdependence** between ZMMs, dHJs, and the SC.

Importantly, our model does not argue that the tripartite SC is strictly required for crossover formation, as evidenced by the viability and crossover levels in a subset of SC-deficient mutants. Rather, we propose that **ZMM-stabilized recombination intermediates promote the maintenance of SC structures**, and that this state contributes to a negative feedback loop that suppresses additional DSB formation and ensures that recombination and chromosome synapsis are temporally aligned with meiotic progression.

In this light, the *Yen1-ON* experiments (including those in the inducible Ndt80 background) demonstrate that disruption of recombination intermediates destabilizes both SC and ZMM ensembles, and that loss of this integrated structure results in checkpoint activation and failed chromosome segregation. Whether one views this from the perspective of recombination control, ZMM function, or SC architecture, the conclusion is the same: this interdependent supramolecular ensemble has an active, regulatory role in meiotic progression and crossover assurance.

ii. An experimental note regarding the comparison between the inducible Ndt80 experiment and *ecm11/gmc2* mutants

In the *ndt80Δ* background, Yen1-ON expression triggers HJ resolution and ZMM dissociation from chromosomes, causing loss of synapsis and de novo DSB formation. When DSBs form, the DNA damage response is activated, which would normally prevent Ndt80 accumulation (in a *NDT80* cell), as thoroughly documented by studies from the Hollingsworth lab (among others). In the inducible Ndt80 experiments, because *NDT80* is under the control of pGAL1, the induction system over-rides this regulation and cells are eventually forced to enter meiosis I with unrepaired DSBs and recombination intermediates. This leads to predictable massive chromosome segregation problems, which should not be compared to the phenotypes of *ecm11/gmc2* mutants.

In our view, the beauty of this experiment (suggested by Reviewer 1 and complemented by the live imaging experiments in NDT80-WT background) is that it shows very clearly that when HJ are severed, the consequence is absolutely massive – cells are unable to undergo meiosis I efficiently. A separate question is how do *ecm11/gmc2* mutants remain able to eventually downregulate DSB formation, which is far beyond the scope of our work but that merits further investigation.

iii. On SC-deficient mutants and the necessity of synapsis for crossover assurance

We respectfully disagree that the phenotypes of *ecm11Δ* and *gmc2Δ* mutants contradict our model. As the new data provided in our previous response (Figure for Reviewers 1 and 2) and extensive prior work show, these mutants display elevated DSB levels, extended prophase I, and delayed “synapsis” kinetics. While they ultimately generate sufficient crossovers, this occurs via mis-regulated recombination and checkpoint adaptation, rather than a robust assurance mechanism. Our experiments with DSB depletion, clearly demonstrate that excessive DSB formation is required for meiosis I in the absence of *ecm11/gmc2* (Figure for Reviewers 2, previous rebuttal letter). The fact that *ecm11/gmc2* mutants can eventually “rescue” chromosome segregation via extended DSB cycles supports our view that the feedback mechanism provided by stabilized dHJs and SC maintenance plays a central role in coordinating crossover completion with timely meiotic progression. We also note that our model accommodates the observation that partial chromosome synapsis—mediated by short Zip1 stretches—is possible even in SC central element mutants. We DO NOT equate this with fully formed tripartite SC (as correctly pointed out by the reviewer).

iv. Title and framing of the central claim

That said, we appreciate the reviewer’s concern that the framing of our *Title* and last sentence of the *Introduction*, may be perceived as overly centered on synapsis and crossover assurance. We recognize that, from a different interpretive angle, our findings might be viewed as primarily emphasizing the *protection of recombination intermediates* or the *function of ZMMs*, with SC maintenance as a consequence rather than a causal factor.

In response, and in the spirit of constructive compromise, we have made the following changes to broaden the framing:

- **Title:**

From: *“Holliday junctions maintain meiotic chromosome synapsis to enable crossover assurance”*

To: ***“Holliday junction-ZMM protein feedback enables meiotic crossover assurance”***

- **Final sentences of the Introduction:**

From: *“We uncovered a reciprocal functional interplay between dHJs and ZMM proteins that is crucial for maintenance of the synaptonemal complex structure. By maintaining chromosome synapsis, which suppresses de novo DSB formation, the dHJ-ZMM interplay coordinates meiotic progression with crossover assurance”*

To: ***“We uncovered a reciprocal functional interplay between dHJs and ZMM proteins that is crucial for the maintenance of the SC structure. We propose that by supporting chromosome synapsis - and thereby contributing to the suppression of de novo DSB formation - this dHJ-ZMM interplay coordinates meiotic progression with crossover assurance.”***

These revisions preserve our core model while addressing the reviewer’s request.

Concluding remarks

Even though it may not have come across in our previous response, we appreciate the reviewer’s detailed and thoughtful critique, which has strengthened our manuscript. While we maintain that our interpretation is well-supported by the data and consistent with previous literature, we agree that alternative perspectives are possible, and we hope the reviewer finds the revised framing to be both scientifically rigorous and appropriately nuanced.